# Oncogene aberrations drive medulloblastoma progression, not initiation

Konstantin Okonechnikov[1,2,3,27], Piyush Joshi[1,2,3,4,27], Verena Körber[5,6,27], Anne Rademacher[7], Michele Bortolomeazzi[8], Jan-Philipp Mallm[8], Jan Vaillant[1,2,4,9,10], Patricia Benites Goncalves da Silva[1,2,4], Britta Statz[1,2,3], Mari Sepp[11], Ioannis Sarropoulos[11], Tetsuya Yamada[11], Andrea Wittmann[1,2,9], Kathrin Schramm[1,2,9], Mirjam Blattner-Johnson[1,2,9], Petra Fiesel[1,2,12], Barbara Jones[1,2,9,13], Natalie Jäger[1,2,3], Till Milde[1,2,13,14], Kristian W. Pajtler[1,2,3,13], Cornelis M. van Tilburg[1,2,13,14], Olaf Witt[1,2,13,14], Konrad Bochennek[15], Katharina Johanna Weber[16,17,18,19], Lisa Nonnenmacher[20], Christian Reimann[20], David R. Ghasemi[1,2,3,21,22], Ulrich Schüller[21,22,23], Martin Mynarek[21,24], Stefan Rutkowski[21], David T. W. Jones[1,2,9], Andrey Korshunov[1,2,12,25], Karsten Rippe[7], Frank Westermann[1,2,26], Supat Thongjuea[1,2,3], Thomas Höfer[5], Henrik Kaessmann[9], Lena M. Kutscher[1,2,4,28 ✉] & Stefan M. Pfister[1,2,3,13,28 ✉]

Despite recent advances in understanding disease biology, treatment of group 3/4 medulloblastoma remains a therapeutic challenge in paediatric neuro-oncology[1]. Bulk-omics approaches have identified considerable intertumoural heterogeneity in group 3/4 medulloblastoma, including the presence of clear single-gene oncogenic drivers in only a subset of cases, whereas in most cases, large-scale copy number aberrations prevail[2,3]. However, intratumoural heterogeneity, the role of oncogene aberrations, and broad copy number variation in tumour evolution and treatment resistance remain poorly understood. To dissect this interplay, we used single-cell technologies (single-nucleus RNA sequencing (snRNA-seq), single-nucleus assay for transposase-accessible chromatin with high-throughput sequencing (snATAC-seq) and spatial transcriptomics) on a cohort of group 3/4 medulloblastoma with known alterations in the oncogenes *MYC*, *MYCN* and *PRDM6*. We show that large-scale chromosomal aberrations are early tumour-initiating events, whereas the single-gene oncogenic events arise late and are typically subclonal, but *MYC* can become clonal upon disease progression to drive further tumour development and therapy resistance. Spatial transcriptomics shows that the subclones are mostly interspersed across tumour tissue, but clear segregation is also present. Using a population genetics model, we estimate medulloblastoma initiation in the cerebellar unipolar brush cell lineage starting from the first gestational trimester. Our findings demonstrate how single-cell technologies can be applied for early detection and diagnosis of this fatal disease.

Intratumoural heterogeneity, a hallmark of cancer, refers to the presence of diverse molecular and functional cell populations within a single tumour[4]. Intratumoural heterogeneity is driven by genetic mutations, transcriptomic or epigenomic plasticity and reprogramming of the microenvironment[5]. The malignant childhood tumour medulloblastoma is heterogeneous[2], especially in groups 3 and 4. This heterogeneity makes effective treatment of these tumours difficult and contributes to overall low survival rates[6]. Advanced DNA methylation profiling has classified group 3/4 tumours into eight distinct molecular subgroups[7]. In addition, single-cell transcriptomic profiling has unveiled the intricate regulatory activity of transcription factors and signalling pathways that orchestrate cellular diversity[8,9]. Despite these advances, the role of oncogenes in shaping intratumour heterogeneity remains unknown.

A minority of group 3/4 medulloblastoma tumours harbour single-gene oncogenic drivers, including *MYC*[10] and *MYCN*[11] amplifications as well as *PRDM6* overexpression owing to enhancer hijacking by means of a tandem duplication of the adjacent *SNCAIP* gene[2]. By contrast, most group 3/4 tumours display recurrent, large-scale copy number changes[2,12,13], including loss of chromosomes 8 and 11 and gain of chromosome 7 and isochromosome 17q. A fundamental question of which genetic events initiate and drive these tumours remains unanswered. Using single-cell multi-omics and spatial transcriptomic approaches, we determined the interplay between large-scale copy number variants (CNVs) and single-gene somatic events in driving medulloblastoma heterogeneity and evolution.

## Driver oncogenic events are subclonal

To understand the clonal genetic events in tumour initiation, evolution and progression, we molecularly profiled a specific tumour cohort with a known amplification of *MYCN* or *MYC* or overexpression of *PRDM6*

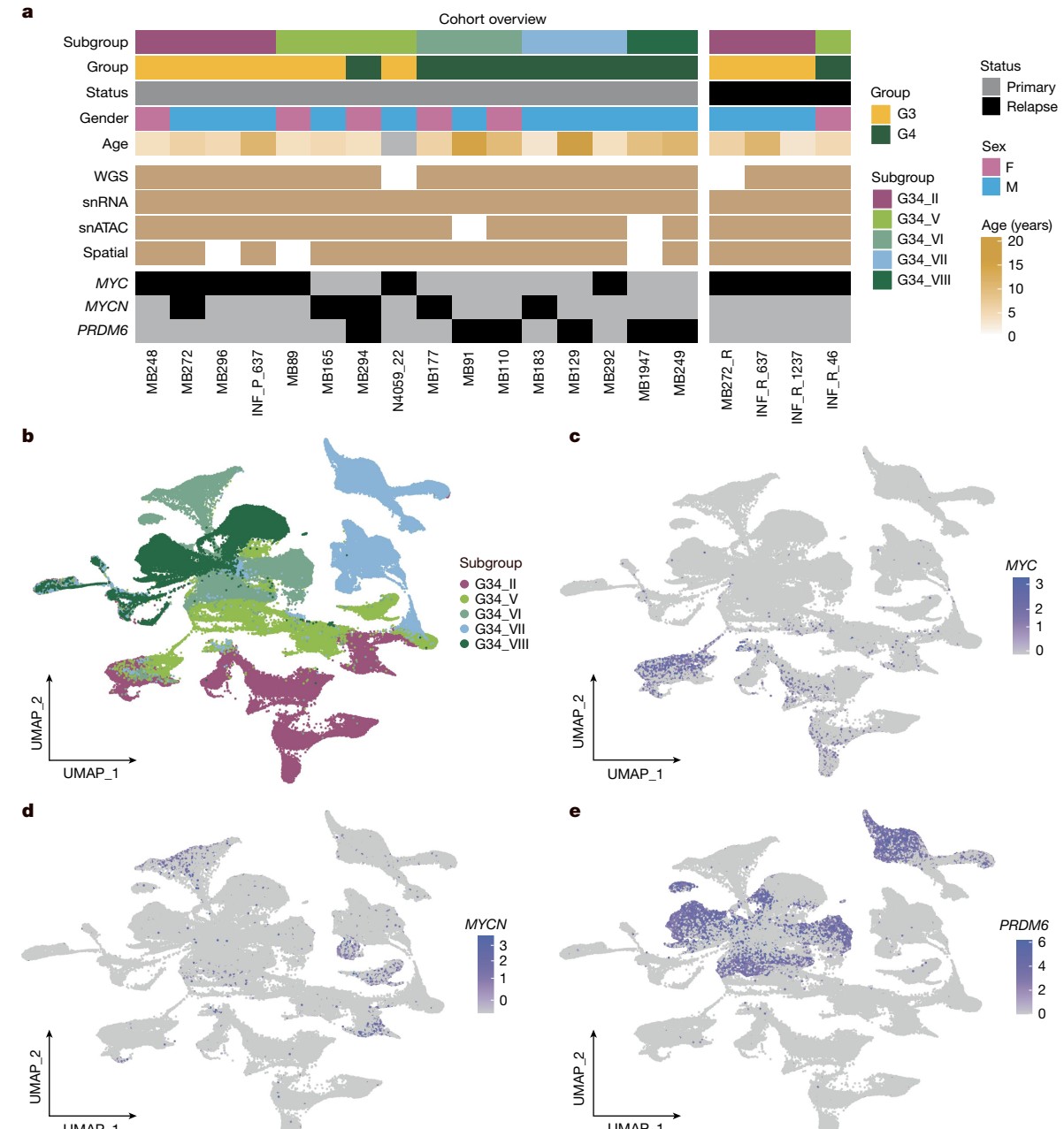

**Fig. 1 | Single-nucleus transcriptional profiling of 16 oncogene-associated group 3/4 medulloblastoma primary tumour samples. a**, Overview of target cohort with annotation. Two primary-relapse pairs (MB272/R, INF_P/R_637) are from the same patients. **b**, UMAP of snRNA-seq merged dataset, with medulloblastoma subgroups annotated. **c–e**, Feature plots showing *MYC* (**c**), *MYCN* (**d**) and *PRDM6* (**e**) expression within the UMAP of the merged snRNA-seq dataset. G3, group 3; G4, group 4; G34, group 3/4 medulloblastoma; F, female; M, male.

(*n* = 16 primary, *n* = 4 relapses; Fig. 1a and Supplementary Table 1). In larger datasets, amplification or activation of these oncogenes is present in approximately 30% of group 3/4 medulloblastoma cases[2], whereas approximately 70% of cases lack a single-gene somatic event. The presence of focal *MYC*/*MYCN* amplifications or the *SNCAIP* tandem duplication was verified using bulk molecular profiles or fluorescence in situ hybridization (FISH) for each sample in this cohort (Supplementary Table 1). We analysed single-nucleus profiles of our target cohort, examining both snRNA-seq (*n* = 20) and snATAC-seq (*n* = 16) in the same nuclei.

Uniform manifold approximation and projection (UMAP) visualization of the snRNA-seq data from primary tumours showed group 3/4 subgroup-specific clusters without batch effect adjustments (Fig. 1b);

mixed normal cell types arising from different samples clustered together as expected (Extended Data Fig. 1a). Expression of the oncogenes *MYC*, *MYCN* and *PRDM6* demonstrated clear sample specificity (Fig. 1c–e and Extended Data Fig. 1b). In addition, expression of known marker genes delineated non-tumour cell types, including *PTPRC* (microglia), *IGFBP7* (meningeal) and *AQP4* (astroglia) (Extended Data Fig. 1c–e). Non-tumour cell clusters were also verified from CNV profiling of the full combined snRNA-seq dataset (Extended Data Fig. 1f). This separation of structure was recapitulated with snATAC-seq, as visualized by means of UMAP (Extended Data Fig. 1g,h). For samples with multi-omics data, non-tumour cells in snATAC-seq were labelled on the basis of their associated non-tumour clusters from the snRNA-seq data (Extended Data Fig. 1g).

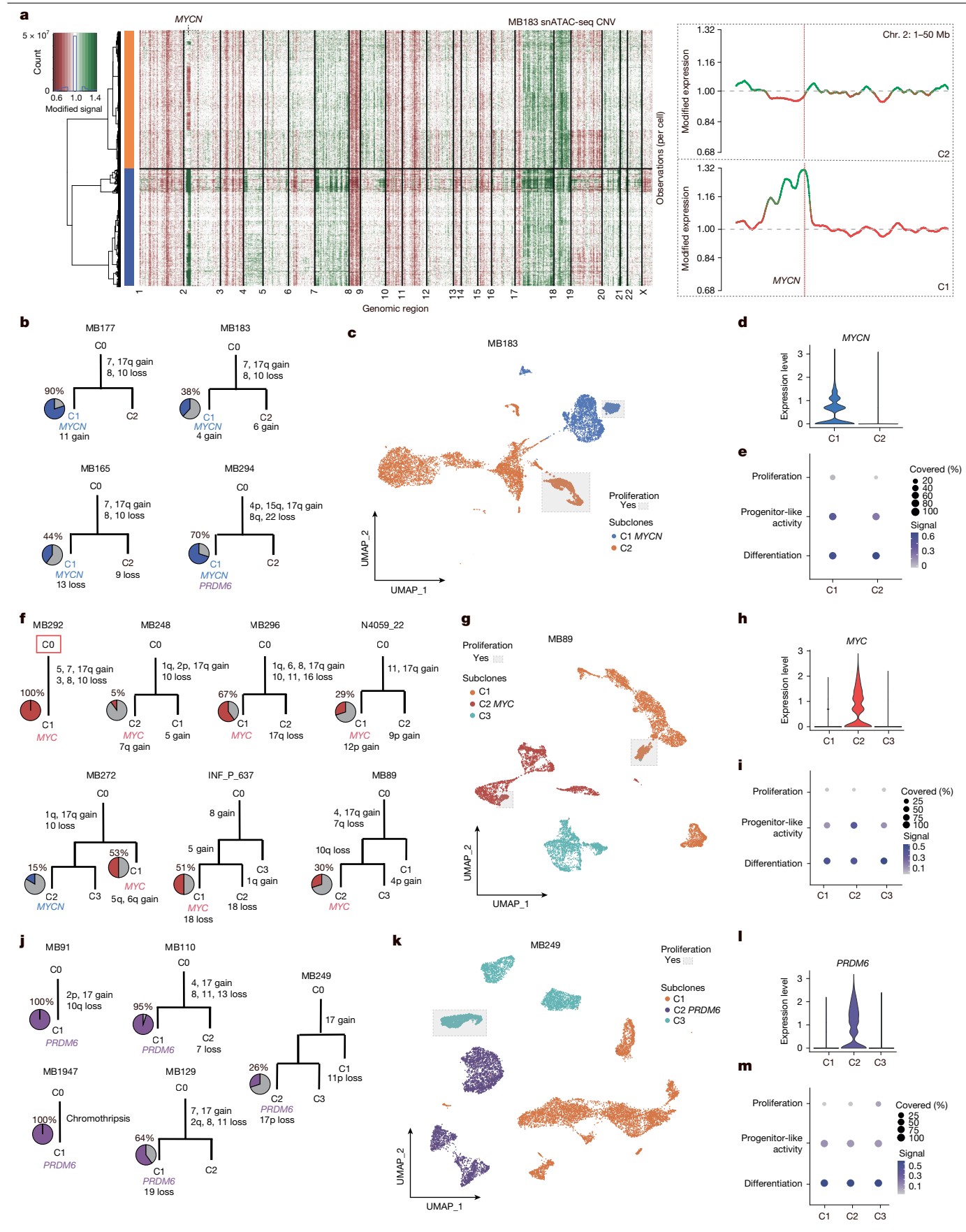

**Fig. 2 |** See next page for caption.

**Fig. 2 | Clonal proliferation and differentiation gradients are independent of oncogene expression. a**, Copy numbers derived from snATAC-seq data in *MYCN*-amplified sample MB183. Red, chromosome loss. Green, chromosome gain. Right side, inset of *MYCN* chromosome region. **b**, Somatic phylogeny trees for *MYCN* samples. Blue, proportion of *MYCN*-expressing cells. **c**, snRNA-seq UMAP of single *MYCN* sample MB183. Grey boxes, proliferating cell clusters with strong proliferation enrichment. Blue, C1 clone. Orange, C2 clone. **d**, *MYCN* expression in C1 and C2 clones. **e**, Per cell gene set variance analysis (GSVA) enrichments of proliferation, progenitor-like activity and differentiation in single sample shown in **c**. **f**, Somatic phylogeny trees for *MYC* samples. Red, proportion of *MYC*-expressing cells. Red square, cases MB292 and MB248 with somatic mutations in C0. **g**, snRNA-seq UMAP of single *MYC* sample MB89. Grey boxes, proliferating cell clusters with strong proliferation enrichment. Red, *MYC*-expressing C2 clone. Orange, C1 clone. Aquamarine, C3 clone. **h**, *MYC* expression in C1, C2 and C3 clones. **i**, Per cell GSVA enrichments of proliferation, progenitor-like activity and differentiation in single sample shown in **g**. **j**, Somatic phylogeny trees for *PRDM6* samples. Purple, proportion of *PRDM6*-expressing cells. Red square, case MB249 with somatic mutations outside CNV regions. **k**, snRNA-seq UMAP of single *PRDM6* sample MB249. Grey box, proliferating cell cluster with strong proliferation enrichment. Purple, *PRDM6*-expressing C2 clone. Orange, C1 clone. Aquamarine, differentiation signal enrichment in C3 clone. **l**, *PRDM6* expression in C1, C2 and C3 clones. **m**, Per cell GSVA enrichments of proliferation, progenitor-like activity and differentiation in single sample shown in **k**.

To investigate the clonal heterogeneity of *MYCN*-amplified tumour samples (subgroups V/VII), we adapted the inferCNV[14] approach (Methods) to infer CNV profiles of cell clusters per sample, using both snRNA-seq and snATAC-seq data. To verify the single-cell CNV calls, we calculated the correlation between pseudobulk from single-cell and bulk DNA methylation CNV profiles (Supplementary Table 1) across all cases, as shown in Extended Data Fig. 2a–c. A full cohort cross-comparison demonstrated that most of the snRNA-seq pseudobulk CNV profiles matched well, with the exception of $n = 3$ false positive cases (Extended Data Fig. 2d). CNV profiles from snATAC-seq data showed the highest correlation to the correct control bulk profile for all samples (Extended Data Fig. 2e), demonstrating the benefit of using this data type for CNV calling.

From inspection of the CNV results per sample, in most cases, we observed clusters with discordant CNVs, which we labelled as subclones. For example, in all *MYCN*-amplified tumours ($n = 4$), we identified two distinct subclones: with (C1) and without (C2) *MYCN* amplification, respectively (Fig. 2a and Extended Data Fig. 3a). Notably, in all cases, reconstruction of the putative phylogenetic trees showed that *MYCN* amplification was not the initiating event for the tumour. Instead, large-scale CNVs, such as loss of chromosomes 8 and 10 or gain of chromosome 7 or 17q, were already present in the presumptive founder clone (C0) (Fig. 2b). Moreover, further unique CNVs were found only within *MYCN*- and non-*MYCN*-amplified subclones. Detailed visualizations of single-cell CNV profiles per sample are available through the interactive online web application (Methods).

Next, we examined the differentiation, proliferation and aggressive progenitor-like activity states of individual cells within each subclone using snRNA-seq expression of established reference gene lists for these defined medulloblastoma cell states[8]. We identified that both *MYCN*-amplified and non-amplified subclones maintained separate proliferating and differentiated compartments (Fig. 2c,d and Extended Data Fig. 3b,c). The *MYCN* subclone was also uniquely enriched with a progenitor-like gene expression signature (Fig. 2e). As cells differentiated, the oncogene itself showed lower expression within the *MYCN*-amplified subclone (Pearson correlation = −0.23, $P < 2.2 \times 10^{-16}$; Extended Data Fig. 3d), demonstrating that *MYCN* is connected to the undifferentiated state[15]. Similar differentiation levels among subclones and a slight bias towards progenitor activity in *MYCN*-amplified subclones were observed in all four *MYCN*-amplified tumours (Extended Data Fig. 3e).

By inspecting differentially expressed genes specific for each subclone (Supplementary Table 2), we also identified a unique property of non-*MYCN* subclones with a stronger enrichment of genes expressed in unipolar brush cell progenitors ($P < 1.14 \times 10^{-6}$), the cell of origin of group 3/4 medulloblastoma[16,17], whereas *MYCN* subclone-associated genes were not enriched in these genes ($P > 0.05$). This observation was also confirmed using subclone-specific genes associated with *cis*-regulatory elements derived from the integration of snATAC-seq data (Supplementary Table 3).

Performing single-cell CNV analyses on *MYC*-amplified tumour samples (subgroups II/V), we identified a subclonal *MYC* amplification in six of seven samples (Fig. 2f and Extended Data Fig. 3f). Similar to *MYCN*-amplified tumours, the common and likely initiating events in the founder clone (C0) were large-scale chromosome 10 loss and/or chromosome 17q gain, with subclonal *MYC* amplification occurring later during tumour evolution. Remarkably, the clonal structure of *MYC*-amplified tumours was more complex ($n = 3$ of 7 cases), with the formation of three or more unique subclones (Fig. 2f, bottom). Typically, *MYC*-amplified subclones had their own proliferating and differentiating compartments (Fig. 2g,h and Extended Data Fig. 3g,h). Similar to the *MYCN*-amplified clones, only *MYC*-amplified clones demonstrated strong enrichment of progenitor-like activity compared with non-*MYC*-amplified compartments (Fig. 2i). Differentially expressed genes specific to *MYC*-amplified subclones were enriched in known *MYC* target genes[18] ($P < 1.11 \times 10^{-16}$; Supplementary Table 2) and *MYC* expression decreased as cells differentiated (Pearson correlation = −0.18, $P < 2.2 \times 10^{-16}$; Extended Data Fig. 3i), whereas expressed genes in non-*MYC* subclones demonstrated enrichment in unipolar brush cell-related genes ($P < 1.54 \times 10^{-11}$; Supplementary Table 2). The *MYC* subclone-specific genes were not found to be enriched in any corresponding subclone-specific CNVs. Across six samples with subclonal *MYC* amplifications, the differentiation level was similar among subclones; however, progenitor-like activity was significantly enriched in *MYC*-amplified subclones (Extended Data Fig. 3j).

Last, we examined tumours with enhanced *PRDM6* expression (subgroups VII/VIII), in which *SNCAIP* gene duplication leads to aberrant activation of *PRDM6* by means of enhancer hijacking[2]. In our cohort, we identified three of five samples in which *PRDM6* overexpression was subclonal (Fig. 2j and Extended Data Fig. 4a). Chromosome 17q gain was the most frequent CNV within the founder clone (C0). In contrast to *MYC* and *MYCN* clones, we could not identify a distinct proliferating compartment in *PRDM6*-specific clones (Fig. 2k,l). Instead, we found only an overall small proportion of cells (less than 5%) with the proliferation gene signature in *PRDM6* subclones (Extended Data Fig. 4b). We also did not identify enriched progenitor-like activity in the *PRDM6* subclones (Fig. 2m and Extended Data Fig. 4c), except in one specific case in which we detected an *MYCN*-amplified subclone that additionally harboured an *SNCAIP* duplication with associated *PRDM6* overexpression (Fig. 2b, bottom right).

To further verify the presence of subclones, we performed single-cell whole-genome DNA and RNA sequencing from the same cells, in a subset of *MYCN*- and *MYC*-amplified cases, $n = 6$ (Supplementary Table 1). In all tested samples, the presence of the corresponding specific subclones and their CNV profile matched between the snMultiomic data (combined snRNA-seq and snATAC-seq data from the same cell), on the basis of single-cell RNA projection (Extended Data Fig. 5a) and whole-genome sequencing (WGS) CNV analysis (Extended Data Fig. 5b–h).

Despite the low mutational burden in medulloblastoma[2], we also inspected the somatic single-nucleotide variants (SNVs) to confirm

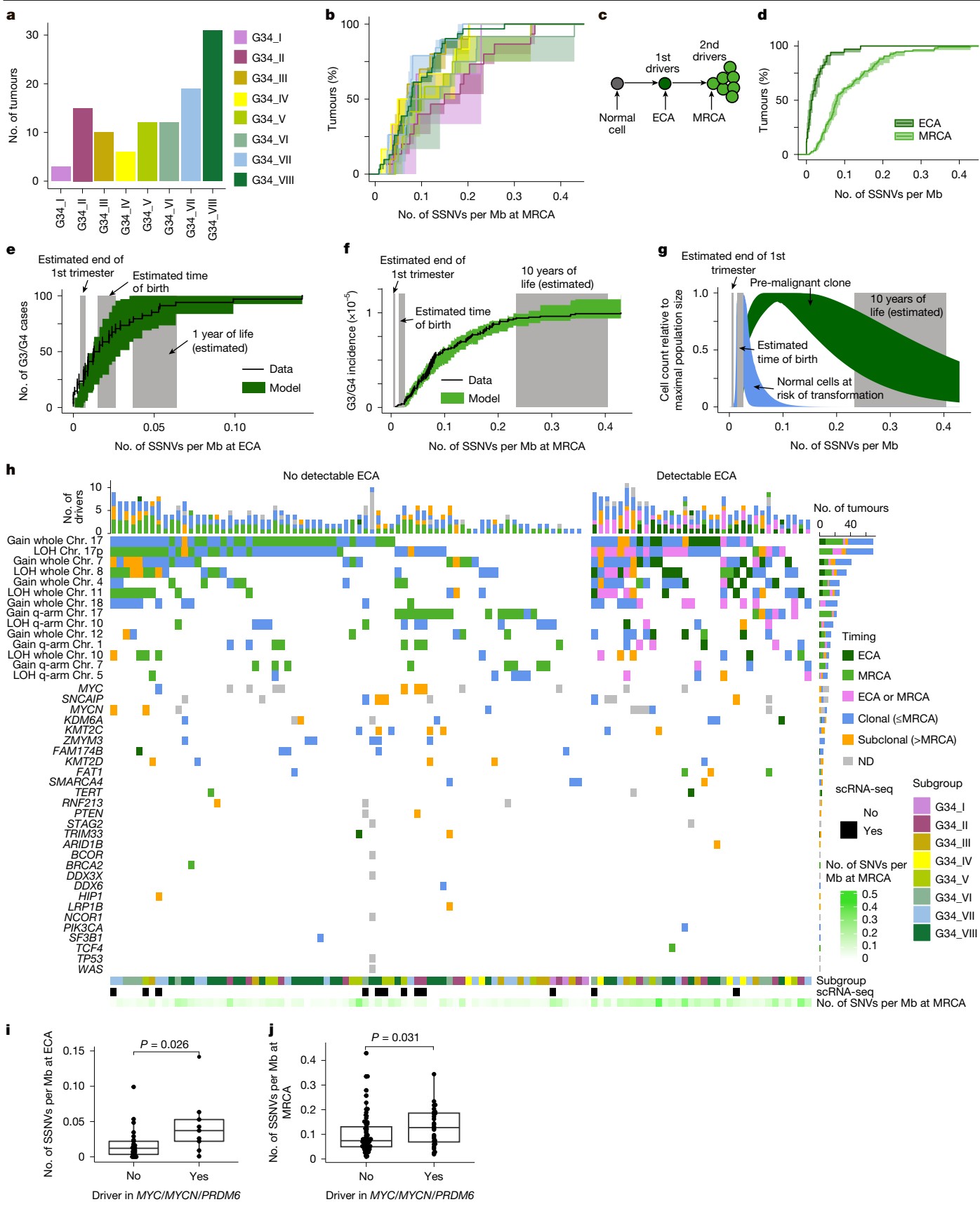

**Fig. 3 | See next page for caption.**

**Fig. 3 | Somatic mutation profiles and association with cell of origin.**
**a**, Group 3/4 medulloblastoma subgroups analysed by bulk WGS. **b**, SNV densities at MRCA per group 3/4 medulloblastoma subgroup (I, $n$ = 3; II, $n$ = 15; III, $n$ = 10; IV, $n$ = 6; V, $n$ = 12; VI, $n$ = 12; VII, $n$ = 19; VIII, $n$ = 31). Shown are mean and 95% CI (estimated by bootstrapping the genomic segments 1,000 times). **c**, Early medulloblastoma evolution. Driver mutation in an ECA spawns a pre-malignant lesion. Malignant transformation occurs upon further drivers in the tumour's MRCA. **d**, SNV densities at ECA and MRCA for group 3/4 medulloblastoma ($n$ = 108). Mean and 95% CI, estimated by bootstrapping the genomic segments 1,000 times. **e**, Model fit to SNV densities at ECA. Line, mean and standard deviation (estimated by bootstrapping the genomic segments 1,000 times) of the measured SNV densities; green and grey areas, 95% credible interval of the model fit, and of key time points. **f**, As in **e**, but for SNV densities at MRCA. **g**, 95% credible intervals of modelled tissue of origin (blue) and pre-malignant clone (green). Grey areas as in **f**. **h**, Mutation spectrum with timing information ('ECA', CNV uniquely timed to ECA; 'MRCA', CNV uniquely timed to MRCA; 'ECA or MRCA', CNV in agreement with both ECA and MRCA; 'clonal', CNV/small mutation was clonal, no further mapping to ECA/MRCA possible; 'subclonal', CNV/small mutation was subclonal; ND, no data). Subclonality information for amplification of *MYC/MYCN* and duplication of *SNCAIP* from single-cell data. **i**, SNV density at ECA in group 3/4 medulloblastoma with and without driver in *MYC/MYCN/PRDM6*. *P* value, unpaired Wilcoxon rank sum test ($n$ = 80 without, $n$ = 28 with driver). **j**, As in **i**, but for SNV density at MRCA. 95% CI, 95% confidence interval; scRNA-seq, single-cell RNA-seq; SSNV, somatic single-nucleotide variants.

the tumour phylogeny composition predicted by snRNA-seq and snATAC-seq data. Using WGS data, we examined mutations in CNV regions specific to the founder clone (C0) or not lying within CNVs. In 9 of 12 samples, we did not identify the presence of drivers or co-mutations (exceptions: two *MYC*-amplified cases, Fig. 2f, and one *PRDM6* case, Fig. 2j). We further investigated the mutational landscape at the single-cell level by increasing the sequencing coverage of the snATAC-seq data in three *MYC*-amplified cases. This approach allowed us to recover up to 40% (range, 20% to 60%) of somatic mutations per sample (Supplementary Table 4). The positive correlation (maximum *P* value: $7.4 \times 10^{-6}$) of their variant allele frequency (VAF) and the corresponding bulk WGS profiles verified the accuracy of the approach (Extended Data Fig. 4d–f). Importantly, in all three cases, it was possible to identify unique somatic mutations specific for the respective subclone, further supporting a common origin of these clones and secondary subclonal evolution (Extended Data Fig. 4g–i). Moreover, by comparing the extended somatic SNVs identified in the snATAC data with germline SNVs (see the Methods for details), the subclone-specific SNVs were observed to have approximately four times lower mean VAF in comparison with those SNVs that are common among subclones (Extended Data Fig. 4j–l).

Collectively, these findings nominate large-scale CNVs as likely tumour-initiating events in group 3/4 medulloblastoma, with focal oncogene aberrations occurring only during tumour evolution.

## Tumour onset from first trimester onwards

To investigate clonal dynamics during the initiation of group 3/4 medulloblastomas, we analysed WGS data from the medulloblastoma International Cancer Genome Consortium (ICGC) cohort[2] (Supplementary Table 5). Somatic tissues accumulate SNVs continuously over time[19–21], and hence SNV density in the tumour cell of origin (in population genetics, 'most recent common ancestor' (MRCA)) can be interpreted as a measure for the patient's age at tumour initiation[22,23]. To time the developmental origin of medulloblastoma with this approach, we quantified clonal SNV densities from the allele frequency distribution of somatic variants in 181 primary medulloblastomas of all subgroups (Extended Data Fig. 6a,b; comprising 108 group 3/4 medulloblastomas, 21 infant Sonic Hedgehog (SHH)-medulloblastomas, 35 childhood/adulthood SHH-medulloblastomas and 17 WNT-medulloblastomas). Overall, the clonal SNV densities across subgroups recapitulated the age-incidence distribution of the disease, with infant SHH-medulloblastoma having the lowest densities (0.02 ± 0.01 SNVs per megabase (Mb)), followed by group 3/4 medulloblastoma (0.1 ± 0.08 SNVs per Mb), WNT-medulloblastoma (0.28 ± 0.47 SNVs per Mb) and adult SHH-medulloblastoma (0.41 ± 0.46 SNVs per Mb; Extended Data Fig. 6c). Clonal SNV densities were also correlated with age at diagnosis (Spearman's $\rho$ = 0.73, $P < 2.2 \times 10^{-16}$; Extended Data Fig. 6d), collectively supporting our approach to infer the evolutionary dynamics at medulloblastoma onset from somatic SNVs.

To estimate age of tumour initiation in group 3/4 medulloblastomas, we analysed 109 tumour samples of this subgroup in more detail (Fig. 3a). As with the entire cohort, clonal SNV densities were likewise correlated with the age at diagnosis among group 3/4 medulloblastoma (Spearman's $\rho$ = 0.51, $P = 3.959 \times 10^{-8}$; Extended Data Fig. 6e). However, contrary to the clear temporal order in tumour initiation of the major medulloblastoma groups, clonal SNV densities were statistically indistinguishable between group 3/4 medulloblastoma subgroups I–VIII (Wilcoxon rank sum test, all adjusted *P* values > 0.05), indicating that growth of the final tumour mass begins around the same developmental time window in all group 3/4 medulloblastoma subgroups (Fig. 3b).

To refine our analysis, we timed the acquisition of clonal CNVs (copy number gains or loss of heterozygosity (LOH)) relative to the tumour's MRCA. To this end, we compared densities of clonal SNVs acquired before a chromosomal gain, and hence present on multiple copies of a chromosomal region, with the density of clonal SNVs overall (Methods). Similar to neuroblastoma[23] and other tumour entities[22], 34 of 109 group 3/4 medulloblastomas showed evidence of having acquired at least one copy number gain in an early common ancestor (ECA), antecedent to the tumour's MRCA (Fig. 3c,d). The number of such early CNVs varied between 1 and 16 per tumour (mean, 5.3), with no significant difference in SNV density between early CNVs within a tumour (Extended Data Fig. 6f). Hence, in tumours with early CNVs, all early CNVs probably arose in an ECA during a confined time window before the onset of tumour growth. In the remaining cases, where we identified no early CNVs, clonal chromosomal gains probably occurred concomitantly with, or shortly before, the onset of tumour growth. To corroborate these observations, we contrasted our approach[23] with an alternative computational tool, MutationTimeR[22], which yielded similar results (Extended Data Fig. 6g). Hence, our data suggest that at least some CNVs in group 3/4 medulloblastoma arise before the onset of tumour growth, in line with multiple rounds of mutation and selection at tumour initiation.

To date these events in actual time, we calibrated a population genetics model of mutation and selection during tumour initiation[23] with the measured SNV densities at ECA and MRCA, along with the patient age at diagnosis (see the Methods for details). Briefly, the model assumes that medulloblastoma initiation is driven by clonal selection for two consecutive drivers in the transient cell population of (differentiating) unipolar brush progenitor cells from the rhombic lip[16,17,24] (Extended Data Fig. 6h). Acquisition of the first driver defines a pre-malignant state, arising before a tumour's MRCA. Although in principle the first driver can be any type of mutation, small driver mutations are overall rare in group 3/4 medulloblastoma[2], and thus are probably not the major driving force of early tumour evolution. Hence, we focused on cases in which early CNVs defined an ECA, and associated the time point at which the first driver mutation was acquired with the mutation density in the ECA. We assumed that the second driver emerges subsequently in the pre-malignant clone, spawned by the ECA.

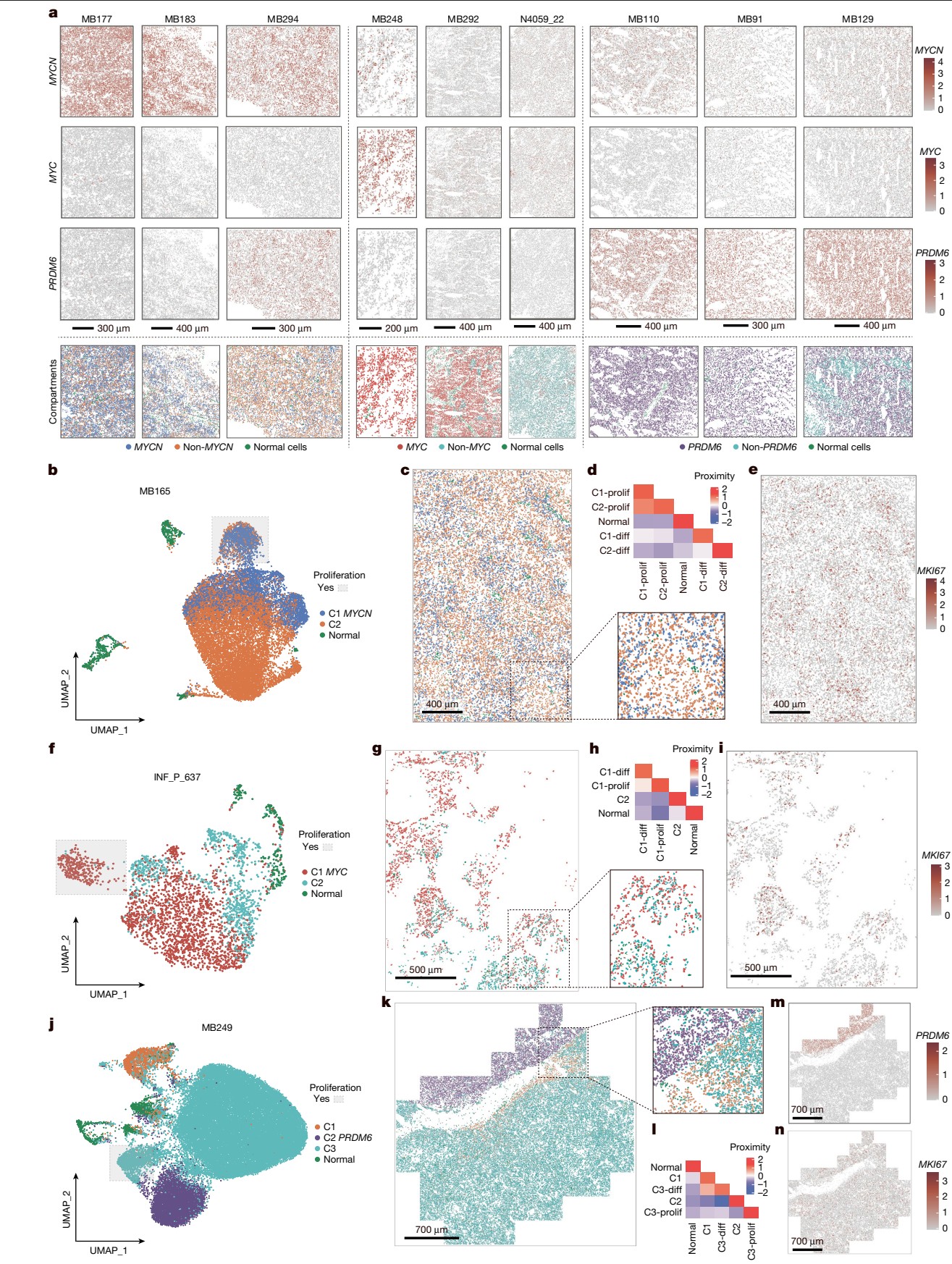

**Fig. 4 | See next page for caption.**

**Fig. 4 | Spatial heterogeneity across oncogene-associated group 3/4 medulloblastoma samples. a**, Spatial gene expression of *MYC*, *MYCN* and *PRDM6*. Last row, projection of clones derived from snRNA-seq. **b**, Spatial data UMAP of representative *MYCN* sample. **c**, Spatial visualization of clones of sample in **b**. Enlarged view of a fragment in the bottom right. **d**, Proximity of each compartment to each other of sample in **b**. **e**, *MKI67* spatial expression of sample in **b**. **f**, Spatial data UMAP of representative *MYC* sample. **g**, Spatial

Hence, we associated the mutation density in the MRCA with the acquisition of the second driver and the onset of tumour growth. Finally, upon malignant transformation of the tumour's MRCA, we assumed exponential growth to a tumour size of $10^9$ cells (corresponding to a few cubic centimetres) at the age of diagnosis[25] (see Methods for details).

Using the bespoke model, we estimated driver mutation rates and associated selective advantages from the clonal SNV densities at ECA and MRCA measured across group 3/4 medulloblastomas. Simultaneously, we estimated per-tumour doubling times using age and the subclonal VAFs of 35 tumours with sufficient data quality and information on age at diagnosis (Supplementary Table 5). Consistent with a higher activity of S-phase genes and MYC target genes and poorer overall survival[26] (Extended Data Fig. 6i–k), we estimated shorter tumour doubling times in medulloblastomas at the group 3 pole as compared with tumours at the group 4 pole (Extended Data Fig. 6j), confirming our modelling approach. We then used the per-tumour doubling times to translate SNV densities at ECA and MRCA into real-time, finding that the first oncogenic event (that is, the ECA) occurs within the first gestational trimester in 24% of cases, during late gestation in around 35% of cases and within the first year of life in 26% of cases (Fig. 3e and Extended Data Fig. 6l). The onset of tumour growth from its MRCA is placed considerably later, within the first decade of life (Fig. 3f), suggesting a long latency phase between pre-malignancy and the detection of a symptomatic tumour. Overall, the inferred dynamics of tumour initiation are consistent with a tumour origin in (differentiating) unipolar brush progenitor cells[24,27], sustaining a pre-malignant clone that outlives the cell state of origin for several years (Fig. 3g).

## Early acquisition of large-scale CNVs

To gain mechanistic insight into group 3/4 medulloblastoma initiation, we asked whether particular mutations occur predominantly early or late. To address this question, we first focused on CNVs that were found more frequently than expected by chance, and hence are likely drivers of malignancy. Combining the enrichment results obtained in our cohort (Extended Data Fig. 7a and Methods) with published data[28], we classified gains of chromosomes 1q, 4, 7, 12, 17/17q and 18, as well as LOH on chromosomes 5q, 8, 10/10q, 11 and 17p, as putative drivers of group 3/4 medulloblastoma initiation. Except for four cases, in which no ECA was identifiable, all group 3/4 medulloblastomas harboured at least one of these CNVs clonally (Fig. 3h). To search for putative drivers located on these regions, we analysed the expression of genes lying in commonly gained or lost regions. For this purpose, we contrasted gene expression between tumours with and without particular CNVs in a bulk RNA-seq cohort of group 3/4 medulloblastoma[3] (Supplementary Table 6a) and inspected differentially expressed genes specific for group 3/4 medulloblastoma in a global central nervous system tumour cohort[24] (Supplementary Table 6b). Among these identified genes, up to 10% have been described as known somatic drivers, and thus associated with medulloblastoma evolution[29].

Overall, gains of chromosome 17 or 17q were the most frequent aberrations in group 3/4 medulloblastoma, followed by gain of whole chromosome 7 and LOH of whole chromosome 8. The SNV densities at chromosomal gains were often smaller than the SNV densities at the

tumour's MRCA, in particular for gains of chromosomes 4, 7, 12 and 17 and LOH of chromosomes 8 and 11 (Extended Data Fig. 7b). Although we cannot rule out that small mutations or chromosomal losses preceded these gains, their consistent early timing suggests that chromosomal gains or losses might be among the earliest events during group 3/4 medulloblastoma initiation. In contrast to the high abundance of CNVs, focal events in known driver genes (SNVs, indels, focal amplifications/deletions or structural rearrangement) were overall rare (Fig. 3h). Among these, amplification of *MYC* or *MYCN* and duplication of *SNCAIP* leading to *PRDM6* overexpression were the most frequent alterations (Fig. 3h). However, the single-cell analysis (compare with Fig. 2b,f,j) showed that these mutations were mostly subclonal. Interestingly, group 3/4 medulloblastomas with amplification of *MYC* or *MYCN* or duplicated *SNCAIP* had significantly higher SNV densities at both ECA (Fig. 3i) and MRCA (Fig. 3j) than the remaining tumours, suggesting that later onset of (pre-)malignancy may predispose to the subsequent acquisition of these drivers. In general, the mutational landscape in group 3/4 medulloblastoma suggests a fundamental role of CNVs during tumour initiation, although further mutations acquired during disease progression seem to drive subclonal evolution in a subset of tumours only.

## Subclonal spatial heterogeneity

To better understand the spatial relationship of the tumour subclones and disclose further insight into their evolution, we performed spatial transcriptomics on samples with available material (*n* = 13 primary, *n* = 4 relapse). We used technology that applies multiplexed in situ hybridization of a selected gene set to achieve single-cell spatial resolution of a tumour sample (Supplementary Tables 7 and 8). UMAP visualization of merged spatial data from primary tumour samples reflected combined snMultiomic profiling, allowing us to distinguish subgroup-specific properties and identify non-tumour cell types (Extended Data Fig. 8a–e). The spatial locations of tumour cells were determined by the expression of *MYC*, *MYCN* and *PRDM6* (Fig. 4a), along with other genes associated with proliferation (for example, *MKI67*; Extended Data Fig. 8f). The tumour microenvironment, including glial, immune and meningeal cells, was characterized using cell type-specific markers (Extended Data Fig. 8f).

To determine the spatial distribution of the identified subclones, we projected the snRNA-seq data onto the spatial data (Fig. 4a, last row). Overall, we distinguished two major spatial localization patterns: interspersed, in which independent subclones mixed throughout the tumour sample, and segregated, in which a clear boundary between independent subclones could be delineated. In most cases, the subclones were interspersed, as observed from the spatial distribution of the corresponding marker gene expression.

In *MYCN*-amplified tumours, the observed clonal architecture derived from snRNA-seq data was also present in the spatial data (Fig. 4b). The subclones exhibited an interspersed spatial pattern (Fig. 4c), with pockets of *MYCN* and non-*MYCN* clones highlighted through neighbourhood enrichment within the tumour sample (Fig. 4c, inset). The proliferating compartments within these subclones were also interspersed across the tumour tissue, observed by *MKI67* expression (Fig. 4e and Extended Data Fig. 8f). On a smaller scale, however, neighbourhood enrichment analysis showed that proliferating cells of subclones clustered together,

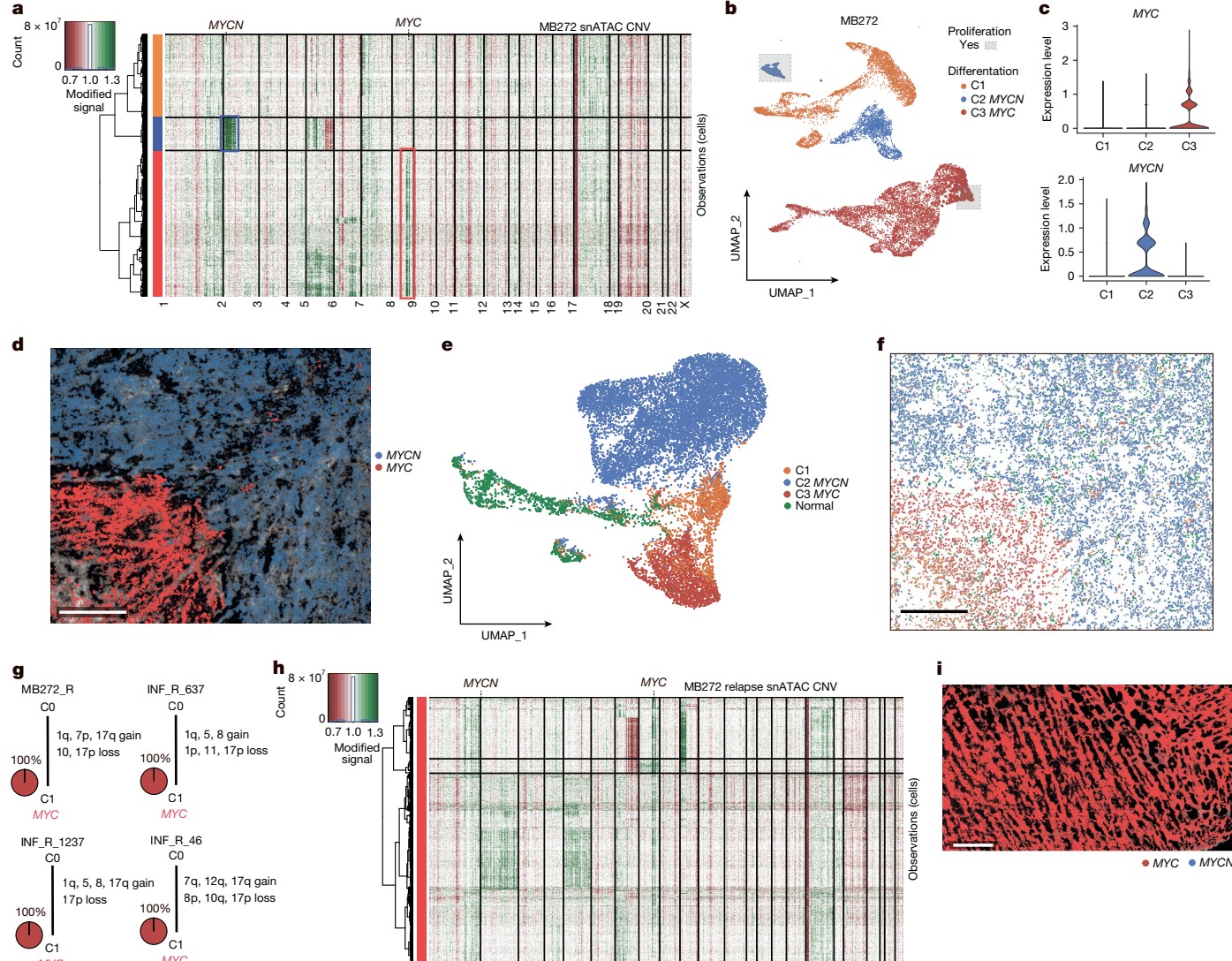

**Fig. 5 | Independent oncogene subclones may co-occur in one tumour, but subclones are lost at relapse. a**, Copy number profiles of snATAC-seq data from *MYC-MYCN* sample MB272. Red, chromosome loss. Green, chromosome gain. **b**, snRNA-seq UMAP of sample shown in **a**. Grey boxes, proliferating cell clusters with strong proliferation enrichment. Blue, *MYCN*-expressing C2 clone. Red, *MYC*-expressing C3 clone. Orange, C1 clone. **c**, *MYC* and *MYCN* expression in C1, C2 and C3 clones. **d**, Spatial gene expression of *MYC* (red) and *MYCN* (blue) from original signals. **e**, Spatial data UMAP of sample shown in **d. f**, Spatial visualization of clones of sample in **d. g**, Somatic phylogeny trees for *MYC* relapse samples. **h**, Copy number profiles of snATAC-seq data of relapse sample arising from primary sample shown in **a–f. i**, Spatial gene expression of *MYC* (red) and *MYCN* (blue) in spatial transcriptomic relapse sample of case shown in **a–f**. Scale bars, 400 μm (**d,f**), 300 μm (**i**).

away from the differentiating cells (Fig. 4d). The normal cells were mostly isolated from the tumour subcompartments.

A similarly interspersed spatial pattern of *MYC* and non-*MYC* subclones was present in *MYC*-amplified tumours (Fig. 4f,g), yet islands of segregated non-*MYC* subclones were also observed (Fig. 4g, inset). Proliferating cells (*MKI67*+) were interspersed throughout the tumour tissue (Fig. 4i and Extended Data Fig. 8f). The differentiated tumour cell compartment within the *MYC* subclone (C1-diff) was in closer proximity to the proliferating cell compartment within the same subclone (C1-prolif; Fig. 4h). Normal cells were largely isolated from the *MYC*-amplified subclone compartments.

We observed a segregated spatial separation of subclones in the *PRDM6* sample (Fig. 4j,k,m). This spatial segregation of clones was confirmed in another region of the same tumour specimen (Extended Data Fig. 8g,h). Although the subclones were segregated, the proliferating cell compartments within the subclones were interspersed within the spatial block (Fig. 4n and Extended Data Fig. 8i). As expected, the

neighbourhood enrichment analysis in this sample showed the separation of the *PRDM6* clone from other compartments (Fig. 4i). In another sample we also confirmed the dual-oncogene, the *PRDM6-MYCN* subclone (Fig. 2b, bottom right), in which cells with activity in both genes reside in the same spatial regions (Extended Data Fig. 8j,k).

Together, the observed spatial patterns suggest that clonal evolution does not lead to spatial compartmentalization within tumours. Instead, cellular migration may dictate communication among the clones, which can then drive competing or collaborative interactions among the co-existing tumour populations.

## *MYC* subclones take over at relapse

We identified a primary tumour sample with two distinct subclones (MB272), harbouring *MYC* (C3) and *MYCN* (C2) amplifications simultaneously (Figs. 2f and 5a–c). This subgroup II sample was originally characterized as *MYC*-amplified only on the basis of bulk methylation

profiling (Extended Data Fig. 9a), whereas the *MYCN*-amplified subclones were observed in the single-cell multiome and spatial transcriptomics analysis. This discrepancy most probably results from examining different fragments of the tumour tissue for each analysis. The *MYC* and *MYCN* subclones in this tumour sample had proliferating and differentiating compartments (Fig. 5b,c). The *MYC* subclone was strongly enriched with progenitor-like activity (Extended Data Fig. 9b). Remarkably, a clear spatial separation was observed between *MYC*- and *MYCN*-expressing cells (Fig. 5d and Extended Data Fig. 9c,d). This spatial segregation (Fig. 5e,f) reflected the phylogenetic tree of tumour evolution projected from the snRNA-seq CNV annotation (Fig. 2f). Further sets of subclone-specific genes also showed explicit spatial specificity (Extended Data Fig. 9e,f). As expected, low contact proximity was identified between *MYC* and *MYCN* subclones (Extended Data Fig. 9g).

According to current knowledge, *MYC* and *MYCN* amplifications are considered mutually exclusive events[30] in medulloblastoma (Extended Data Fig. 9h) and other tumour types[31]. Because of the unexpected occurrence of both amplifications in this case, we conducted a systematic analysis across a larger medulloblastoma cohort to identify further cases in which these oncogenes may co-occur. We identified six putative cases on the basis of DNA methylation CNV profiles (Extended Data Fig. 9i). Using immunohistochemistry, we identified another case in which both MYC and MYCN staining were seen in the same tumour sample (Extended Data Fig. 6j). During the preparation of this manuscript, a case study reported a primary tumour sample in which both *MYC*- and *MYCN*-amplified cells were present[32]. Together, these independent cases suggest the possibility that *MYC* and *MYCN* amplifications co-occur within the same tumour more frequently than originally thought. Nevertheless, it is noteworthy that individual cells within the tumour express only one of these oncogenes and are spatially segregated.

The presence of *MYC* and *MYCN* subclones cannot be distinguished using bulk profile techniques owing to the potential low presence of cells of a particular subclone in the obtained data. Therefore, we generated unique signatures of *MYC* and *MYCN* subclones derived from single-cell data to identify further samples harbouring two oncogene amplifications. We performed a deconvolution analysis of bulk transcriptome profiles, using the *MYC*/*MYCN* case as the reference control. Using this method, we detected further samples in which *MYC* and *MYCN* subclones may co-occur in the same sample (Extended Data Fig. 9k). We validated this finding using FISH on an identified sample with available material (Extended Data Fig. 6l).

We next checked whether this information could be exploited for diagnostic purposes. Therefore, we investigated whether the relative presence of *MYC* or *MYCN* subclones derived from the deconvolution analysis predicted patient outcomes. In subgroup V, 4 of 41 cases harboured a known *MYC* amplification, on the basis of CNV profiles, and correlated with a low probability of survival (Extended Data Fig. 9m). We identified 14 potential cases with an occurrence of an *MYC*-amplified subclone on the basis of deconvolution. These patients had a lower overall survival (Extended Data Fig. 9n). Therefore, the poor outcomes of subgroup V patients may be explained by an undiagnosed *MYC* subclone that potentially outcompetes other subclones to drive relapse.

To further test this possibility, we performed single-nucleus molecular profiling on four relapse *MYC*-amplified cases. In all relapse cases, new subclones arose, but all tumour cells harboured the *MYC* amplification (Fig. 5g). For example, in the matched relapse *MYC*/*MYCN* case, the *MYCN* subclone was lost at relapse (Fig. 5h). This loss of *MYCN* expression was confirmed using spatial transcriptomics (Fig. 5i and Extended Data Fig. 9o,p). These results suggest that the *MYC* subclone outcompetes other subclone(s) during tumour progression and hence the presence of subclonal *MYC* amplification at diagnosis may predict the probability of relapse.

## Discussion

Despite advances in understanding the cellular origin of group 3/4 medulloblastoma, the tumour-initiating and driving mechanisms remain elusive. In our study, we use single-cell multi-omics data analysis combined with spatial profiling to identify the somatic subclone properties within these tumours and demonstrate that the genetic aberrations that lead to overexpression of oncogenes are not the likely initiating events in group 3/4 medulloblastoma. Therefore, *MYC* and *MYCN* are probably not the primary 'drivers', but instead are acquired after malignant transformation and probably accelerate tumour growth. Instead, our results suggest that the initiating or 'driving' events in group 3/4 medulloblastoma are large-scale CNVs. This finding is in line with the hypothesis that tetraploidization is a frequent early event in medulloblastoma[12] and is associated with intermediate survival rates and high risk of relapse[33]. Using mutational clocks and mathematical modelling, we find that medulloblastoma initiation is probably a multi-step process. Specifically, our model suggests that early CNVs, acquired as early as in fetal development, drive a pre-malignant clone, in which malignant transformation occurs within the first decade of life. Although this process is similar to observations in mouse models, in which an initial hyperplastic state precedes medulloblastoma growth[34,35], our modelling approach has some limitations. We here assume that unipolar brush cell (UBC) progenitors, the likely cell of origin of group 3/4 medulloblastomas[16,17], divide at a fairly constant rate and that tumour growth can be approximated with exponential growth. Both assumptions may oversimplify the true biology, which may influence our timing estimates. Reassuringly, however, a disease origin that dates back to fetal development, or the first year of life, agrees with the detection of UBC progenitors in human brain samples from this time span[24]. Intriguingly, another paediatric tumour, neuroblastoma, has a similar order of genetic events: CNVs are the initiating event and occur early in the first trimester of pregnancy[23]. How exactly large-scale CNVs drive early tumourigenesis in different cell types and whether this knowledge can potentially be exploited for early cancer detection remain to be explored.

Group 3/4 medulloblastoma with *MYC*, *MYCN* or *PRDM6* alterations have complex subclonal structures, with each subclone having unique properties. Therefore, our results strongly argue against the dogmatic 'cancer stem cell' hierarchy for group 3/4 medulloblastoma, as these tumours maintain distinct subclones with separate proliferating and differentiating compartments, irrespective of the presence of recognized oncogenes. Although *MYC* or *MYCN* amplifications could arise from the formation of circular extrachromosomal DNA[36], our data show that all subclones with *MYC*/*MYCN* amplification demonstrated other large, specific CNV gains and losses within the oncogene-amplified subclone, suggesting these clones are not the founder clone.

Single-cell spatial data from group 3/4 tumours allowed us to inspect the composition of the subclones across tumour tissue fragments; however, owing to the fixed number of genes ($n = 100$) and limited image size (maximum 2 mm), some other spatial properties, such as formation of blood vessels, were not covered in our study. The application of new spatial techniques that overcome such limitations will be an important research direction in the future.

In addition, undetected *MYC*/*MYCN* subclones challenge the bulk analysis approach in the diagnostic space. These findings, along with support from others[37], suggest that single-cell analyses may be an important diagnostic tool in the future, especially for group 4 and subgroup V tumours. In addition, our data challenge the cutoffs used for a tumour to be called *MYC*/*MYCN*-amplified by FISH, as even the smallest *MYC* subclones, which initially have low abundance, have the potential to expand into the dominant clone during relapse. *MYC* amplification may drive disease progression and contribute to therapy resistance and relapse. Such a pattern of *MYC* dominance in subclonal evolution

has been observed in other tumours including gliomas[38], suggesting that our results may be relevant also to other tumour entities associated with *MYC* oncogenesis.

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

¹Hopp Children's Cancer Center (KiTZ), Heidelberg, Germany. ²National Center for Tumor Diseases (NCT) Heidelberg, a partnership between DKFZ and Heidelberg University Hospital, Heidelberg, Germany. ³Division of Pediatric Neurooncology, German Cancer Research Center (DKFZ) and German Cancer Consortium (DKTK), Heidelberg, Germany. ⁴Developmental Origins of Pediatric Cancer Junior Research Group, German Cancer Research Center (DKFZ), Heidelberg, Germany. ⁵Division of Theoretical Systems Biology, German Cancer Research Center (DKFZ), Heidelberg, Germany. ⁶MRC Molecular Haematology Unit, Weatherall Institute of Molecular Medicine, Radcliffe Department of Medicine, University of Oxford, Oxford, UK. ⁷Division of Chromatin Networks, German Cancer Research Center (DKFZ) and Bioquant, Heidelberg, Germany. ⁸Single-cell Open Lab, German Cancer Research Center (DKFZ), Heidelberg, Germany. ⁹Division of Pediatric Glioma Research, German Cancer Research Center (DKFZ), Heidelberg, Germany. ¹⁰Faculty of Biosciences, Heidelberg University, Heidelberg, Germany. ¹¹Center for Molecular Biology of Heidelberg University (ZMBH), DKFZ-ZMBH Alliance, Heidelberg, Germany. ¹²CCU Neuropathology, German Cancer Research Center (DKFZ) and German Cancer Consortium (DKTK), Heidelberg, Germany. ¹³Department of Pediatric Oncology, Hematology & Immunology, Heidelberg University Hospital, Heidelberg, Germany. ¹⁴CCU Pediatric Oncology, German Cancer Research Center (DKFZ) and German Cancer Consortium (DKTK), Heidelberg, Germany. ¹⁵Frankfurt University Hospital, Goethe University, Frankfurt, Germany. ¹⁶Goethe University Frankfurt, University Hospital, Neurological Institute (Edinger Institute), Frankfurt am Main, Germany. ¹⁷German Cancer Consortium (DKTK), Heidelberg, Germany and German Cancer Research Center (DKFZ), Heidelberg, Germany. ¹⁸Frankfurt Cancer Institute (FCI), Frankfurt am Main, Germany. ¹⁹Goethe University Frankfurt, University Hospital, University Cancer Center (UCT) Frankfurt, Frankfurt am Main, Germany. ²⁰University Children's Hospital, Ulm, Germany. ²¹Department of Pediatric Hematology and Oncology, University Medical Center Hamburg-Eppendorf, Hamburg, Germany. ²²Research Institute Children's Cancer Center Hamburg, Hamburg, Germany. ²³Department of Neuropathology, University Medical Center Hamburg-Eppendorf, Hamburg, Germany. ²⁴Mildred Scheel Cancer Career Center HaTriCS4, University Medical Center Hamburg-Eppendorf, Hamburg, Germany. ²⁵Department of Neuropathology, Institute of Pathology, Heidelberg University Hospital, Heidelberg, Germany. ²⁶Division of Neuroblastoma Genomics, German Cancer Research Center (DKFZ), Heidelberg, Germany. ²⁷These authors contributed equally: Konstantin Okonechnikov, Piyush Joshi, Verena Körber. ²⁸These authors jointly supervised this work: Lena M. Kutscher, Stefan M. Pfister. ✉e-mail: l.kutscher@kitz-heidelberg.de; s.pfister@kitz-heidelberg.de

# Methods

## Target cohort selection and verification

Target tumour tissue samples were collected from global medulloblastoma published materials (ICGC[2], fresh-frozen paraffin-embedded (FFPE)[13] and Individualized Therapy For Relapsed Malignancies in Childhood (INFORM)[39] cohorts). For each selected case, the copy number/structural variant profiles from methylation and/or WGS data were used to identify *MYC/MYCN* amplification and *SNCAIP* structural variant presence. Bulk gene expression RNA-seq profiles from these samples were used to inspect *MYC/MYCN/SNCAIP/PRDM6* expression as well. For some cases with sufficiently available FFPE material, further FISH experiments were performed to verify the selection (details in Supplementary Table 1). No statistical methods were used to predetermine sample size.

## Single-nucleus multi-omics sequencing

Flash-frozen tumour samples were processed to extract nuclei as described earlier[27]. Extracted nuclei were processed using Chromium Single Cell Multiome ATAC Gene expression kit and Chromium Controller instrument (10x Genomics) as per manufacturer's recommendations. One sample, MB248, was processed with Chromium Next GEM Single Cell 3′ reagent kit as per the manufacturer's recommendation. In total, 15,000–20,000 nuclei were loaded per channel along with the multiome gel bead. Libraries were quantified using Qubit Flurometer (Thermo Fisher Scientific) and profiled using Fragment Analyzer. snRNA-seq and snATAC-seq libraries were sequenced using a NextSeq2000 to the recommended lengths. If the snATAC-seq library was not of good quality, we still used the obtained snRNA-seq library if that was found to be appropriate on the basis of quality control parameters. snRNA-seq and snATAC-seq datasets were further analysed separately.

## snRNA-seq data analysis

De-multiplexed reads were aligned to human genome assembly GRCh38 (v. p13, release 37, gencodegenes.org). Comprehensive gene annotation (PRI) was customized by filtering to transcripts with the following biotype: protein coding, lncRNA, IG and TR gene and pseudogene as recommended for cellranger mkgtf wrapper. Reads were aligned using STARsolo with parameters: --soloType *CB_UMI_Simple* --soloFeatures *Gene GeneFull* --soloUMIfiltering *MultiGeneUMI* --soloCBmatchWLtype *1MM_multi_pseudocounts* --soloCellFilter *None* --outSAMmultNmax *1* --limitSjdbInsertNsj *1500000*. For overlapping genes for which intronic alignment recovered low counts, exonic alignment counts were used. Cells were separated from debris using the diem pipeline[40]. Cells with mitochondria fraction above 2%, number of detected genes above 6,600 and an intronic fraction (number of reads aligned to intron/total number of reads aligned to exon + intron) less than 25% were also filtered out. Filtered cells were corrected for background signature using the SoupX pipeline[41]. Finally, the scrublet[42] tool was used to remove putative doublets. Further, the gene expression matrices from all samples were merged together in the full matrix and processed by means of the Seurat package[43] to normalize, compute top principal complements ($n = 30$), find most highly variable genes ($n = 2,500$) and visualize by means of UMAP. After distinguishing non-tumour cells on the basis of corresponding markers and combined UMAPs, per sample processing was performed using the Seurat toolkit using the same settings combined with cell clustering. The enrichment of proliferation, differentiation and progenitor-like activity of medulloblastoma-specific markers per cell was performed using the single sample function from the GSVA R package[44] using two independent reference datasets[8,9]. Cell clusters enriched with proliferation signals were selected and marked on the basis of maximum GSVA signal enrichment from manual inspection per sample.

## snATAC-seq data analysis

ATAC-seq reads were aligned to GRCh38 using the Cellranger arc wrapper. The selected cells were processed using the Signac R package[45] to filter the doublets/outliers on the basis of signal per cell distribution analysis and to inspect the cell compartments by means of UMAP visualization after normalization and identification of the most highly variable regions. snRNA-seq data information was used to annotate the cells from corresponding processed data.

To identify differentially enriched *cis*-regulatory elements per sample per annotation, peaks were first called on the merged snATAC-seq data using the ArchR R package[46] by first creating pseudobulk replicates using addGroupCoverages (minCells=2000, maxCells=5000, minReplicates=2, maxReplicates=5, groupBy="Sample", maxFragments=100 * 10^6) and calling reproducible peaks using addReproduciblePeakSet( groupBy="Sample", maxPeaks=150000). Obtained peaks were further filtered on the basis of presence in at least 5% of cells in any sample. Marker peaks were then obtained using getMarkerFeatures() on the basis of subclonal annotation and filtering on the basis of area under the curve > 0.52 and false discovery rate (FDR) < 0.01.

Genes correlated to the peaks were identified using addPeak2GeneLinks() using the subset of cells with dual snRNA-seq and snATAC-seq profiles in the merged data. Correlated genes (more than 0.1) were intersected with peaks associated with genes identified on the basis of GREAT[47] annotation to identify the robust pairs of peaks to gene links.

## Single-cell CNV phylogeny reconstruction

Initially, CNV analysis was performed on a subset of samples ($n = 2$ for each oncogene) using the InferCNV tool[14] on the raw gene expression matrix with droplet protocol adjusted parameters (average read counts cutoff 0.5, smooth method runmeans, denoise active) and hierarchical clustering by means of the ward.D2 method to derive the clonal phylogeny. The main subclones obtained from this phylogeny demonstrated a close match to the initial Seurat clustering results (mean purity evaluation metrics across samples: 0.921). To further improve the visualization and increase computational efficiency, the CNV analysis was performed on the full cohort by transferring single cells into meta-cells, on the basis of the established method[48]. For this purpose, we computed the sum of gene expression counts across $n = 5$ cells combined within the clusters derived from Seurat processing. The meta-cell InferCNV calling was performed for each sample separately with read counts cutoff 0.5, and the phylogeny clustering results were visualized in UMAPs with $k$ as number of clusters, varying from 2 to 5. Differentially expressed genes for identified subclones per sample were computed by means of a Wilcoxon rank sum test. Significant *MYC/MYCN/PRDM6* differential expression and progenitor-like activity enrichment values were computed per cell to finalize the derived phylogeny cut limit for each case after manual inspection. The selected cut limit for the number of subclones in the phylogeny was verified by using random tree subcluster partition in the phylogeny reconstruction with a minimum $P < 0.05$.

All tumour samples ($n = 16$) were also merged together and CNV profiling was performed to verify the status of non-tumour cells. The annotation of verified normal cells was further used as reference control for each sample.

For snATAC-seq data, a matrix with all genomic regions as raw and read counts per column per sample was used to adjust for InferCNV input format. Further meta-cell formation and the same CNV calling procedure as for snRNA-seq were performed on the derived matrices. The subclone annotation derived from snRNA-seq data was used to assign corresponding cluster phylogeny per sample.

Interactive CNV visualization for the results from snRNA-seq and snATAC-seq data per sample is available through ShinyApp: http://kokonech.shinyapps.io/mbOncoAberrations.

## Mutation calling from snATAC-seq data

For $n = 3$ *MYC* cases the number of reads in snATAC-seq data was increased up to $3 \times 10^9$ per sample to have sufficient coverage for

mutation calling. The alignment of novel reads was performed using the same strategy as described above. Initial subclone annotation was used to classify the cells. Afterwards, the mutation calling was performed using the SComatic method[49] on the BAM files extracted for each subclone. Initial mutation filtering was performed on the basis of inspection of somatic mutations derived from WGS data[2] as the main control. Results were also confirmed using the bcftools method[50] on the full merged pseudobulk snATAC-seq data with standard settings. The VAF from the full merged pseudobulk mutation calling was used to compare with the VAF from bulk control results. The visualization of somatic mutation presence across subclones per sample was performed by means of the R package ComplexHeatmap. For more relaxed filtering control, blood bulk WGS control was applied to exclude non-somatic mutations, while mutation filtering limits were strengthened: min coverage 20× and minimum support 5×. Annotation of identified mutations was performed using Annovar toolkit[51] using gencode v.38 materials.

## Single-cell DNA and RNA sequencing procedure

Flash-frozen tumour samples were processed to extract nuclei as described previously[52]. Single nuclei were sorted into the wells of DNA LoBind 96-well plates using fluorescence activated cell sorting. Each plate was used for downstream genome and transcriptome isolation, reverse transcription of the transcriptomes and primary template-directed amplification using the ResolveOME Multiomic kit from Bioskryb, according to the manufacturer's protocol. Selected wells were used for quality control before library preparation to assess the quantity and quality of the extracted genome or transcriptome DNA, using the 4200 TapeStation System. Sequencing libraries were prepared using Illumina-compatible unique dual index adaptors, according to the manufacturer's protocol as provided in the ResolveOME kit. After amplification, barcoded libraries were pooled and purified using Ampure Beads to select 250–1,000-base pair (bp) fragments with an average peak size of 400–500 bp. Multiplex libraries were sequenced using one lane of the NovaSeq 6000 System each, with 100-bp paired-end sequencing for the genome libraries and 100-bp single-read sequencing for the transcriptome libraries.

## Single-cell DNA and RNA sequencing data analysis

DNA sequencing reads per cell were initially processed (quality control, alignment) by means of the BJ-DNA-QC Nextflow-based pipeline from BioSkryb using hg38 as the main reference. CNV profiling was performed with the Ginkgo tool[53] on the basis of resolution 100 kilobases (kb). Outlier signal adjustment was performed on the basis of cut for mean plus 2 s.d. Heatmap visualization was performed by means of the ComplexHeatmap R package using the clustering method ward. D2 with Euclidean distance.

RNA-seq reads per cell were aligned to the hg38 reference by means of the STAR tool[54]. Gene expression counts were computed by means of the FeatureCounts option from the Subread toolkit[55] using gencode v.38 reference. Combined gene expression matrix analysis was performed by means of the Seurat toolkit[43]. Non-tumour cells as well as subclone specificity were identified on the basis of the inspection of known gene markers and projection into snRNA-seq profiles using the transfer Seurat R toolkit function.

## Molecular cartography

The specific gene set (n = 100) covering group 3/4 known driver genes alongside marker genes of the developing cerebellum non-malignant cell types was selected for the protocol (Supplementary Table 7). The gene selection was on the basis of specific properties. First, the main selection was split into two blocks: tumour-specific genes (60%) and normal cell markers (40%). The tumour-specific genes had several blocks of selection: (1) known target markers of group 3/4 tumours

including *MYC*, *MYCN*, *SNCAIP* and *PRDM6*; (2) proliferation, differentiation and cell-cycle activity markers; (3) cells-of-origin-associated markers. The normal cell type markers were selected on the basis of knowledge about normal cell types of the cerebellum that could also be present in the tumour. For each cell type, at least two markers were selected. All selected genes were also verified on available snRNA-seq data as well as bulk profiles.

Optimal cutting temperature compound (OCT)-embedded samples were cryo-sectioned into 10-µm sections onto a molecular cartography slide. Fixation, permeabilization, hybridization and automated fluorescence microscopy imaging were performed according to the manufacturer's protocol (Molecular preparation of human brain (beta), Molecular colouring, workflow setup) as described previously[52].

## Spatial data analysis

The detection of cell boundaries was performed with CellPose[56]. Afterwards, gene expression counts were computed per cell and extracted using further custom Python scripts. Initial cell filtering was performed by assigning the minimum number of counts/genes per cell and size of the cells. Afterwards, the analysis of the formed gene expression matrix, including clustering and UMAP visualization, was executed using the Seurat toolkit[43]. Annotation of cell states and types was achieved through direct projection with the snRNA-seq data by means of transfer function and verified by visual inspection of marker genes. Spatial-specific analysis, the detection of closest cell connections, was conducted using the Giotto toolkit[57].

## Deconvolution analysis of *MYC*/*MYCN* cases

The assigned *MYC*/*MYCN* single-cell dataset (MB272) with annotation of compartments was used as reference control for the CIBERSORT method[58] to perform deconvolution on a set of bulk FFPE medulloblastoma RNA-seq profiles from *MYC*/*MYCN* samples[13]. For each case, *MYC*/*MYCN* status was derived from methylation copy number profile. The deconvolution values obtained from CIBERSORT results were visualized by using the ComplexHeatmap R package in order to demonstrate compartment enrichments per sample.

Survival analyses on the basis of expression of *MYC* as well as the computed deconvolution of *MYC* compartment proportion of multiple genes were performed using the Kaplan–Meyer algorithm with applied Bonferroni correction for multiple testing. The resulting plots were generated by means of the R2 Genomics Analysis and Visualization Platform (http://r2.amc.nl).

## Bulk RNA-seq data analysis

Bulk RNA-seq data from the ICGC cohort overlapping with WGS data were available for 85 group 3/4 medulloblastomas of our cohort[2]. We performed GSVA on reads per kilobase of transcript per million mapped read scores using the R package GSEABase. We used the following gene sets: MYC target genes: 'HALLMARK_MYC_TARGETS_V2', 'MYC_UP.V1_UP', 'DANG_MYC_TARGETS_UP'[26]; S-phase genes: 'REACTOME_S_PHASE', 'SA_G1_AND_S_PHASES'.

Extended gene expression and associated CNV profile metadata for 405 group 3/4 medulloblastoma samples were integrated for inspection of common CNV gains/losses[3]. The chromosomal arm gains or losses as provided in this corresponding study were binarized. DESeq2 was used to compute differentially expressed genes using lfcShrink(type="apeglm") and filtered using adjusted $P < 0.001$ and log fold change (lfc) > 0.5 criteria to obtain a list of statistically significant altered gene expression. We then used msigdbr and clusterProfiler R packages to identify chromosomal loci of the differentially expressed genes.

Additionally evident differentially expressed genes (adjusted $P < 0.05$) among the central nervous system tumours provided in the corresponding study[24] were inspected to identify those that are specific

for common group 3/4 gain (overexpressed, lfc > 0.5) and loss (low expressed, lfc < 0.5) regions.

## WGS data

Mutation calls (SNVs, indels, CNVs and structural variants (SVs)) of previously published WGS data from medulloblastomas of all subtypes were taken from the ICGC dataset[2]. Only samples from primary tumours with clear subtype annotation and clear ploidy status were included; see Supplementary Table 5 for an overview on these samples and associated clinical data.

## Driver mutations (SNVs and CNVs)

Non-synonymous SNVs, small indels and small structural rearrangements (amplifications, defined as copy number gains ≥ 10, homozygous deletions with less than 0.9 copy numbers and translocations with a minimal event score of 5) were classified as driver mutations if they targeted a splice-site or an exonic region of *PRDM6*, *MYC* or a gene listed as a putative driver of medulloblastoma in the cancer driver database intogen[59] (release date 31 May 2023). Moreover, we included *TERT* promoter mutations at hg19 positions 1295228 and 1295250 as drivers. High-level amplifications affecting *MYC* or *MYCN* (identified from methylation/WGS copy number profiles) and duplications of *SNCAIP*, leading to overexpression of *PRDM6* (identified from WGS SV calling), were additionally integrated from a previous global data analysis[2].

Large-scale CNVs were defined as CNVs spanning at least 1 Mb and with a coverage ratio less than 0.9 or a coverage ratio greater than 1.1, according to the output by ACEseq. Retained CNVs with a size of at least 25% the size of the p arm of a respective chromosome were further classified as affecting both arms if the CNV spanned the centromere, or else as affecting the p arm or the q arm. Among these CNVs, we tested for positive enrichment of particular chromosomes in the cohort using a binomial test with success probability 1/24 (that is, assuming that each chromosome has equal probability to be affected by the CNV). Chromosomes with an adjusted $P < 0.05$ (Holm's correction) were classified as likely drivers of medulloblastoma. This analysis was separately performed for gained and lost chromosomes. Among group 3/4 medulloblastomas, we identified gains of chromosomes 4, 7/7q, 12 and 17/17q and losses of chromosomes 8, 10/10q, 11 and 17p as significant. We augmented this list by gains of chromosomes 18 and 1q and loss of 5q, as was reported previously[60].

## Timing of CNVs, ECA and MRCA

Quantification of mutation densities at copy number gains was performed using the R package NBevolution v.0.0.0.9000, which is described in detail in the corresponding study[23]. In brief, we counted clonal mutations separately on each autosome, stratified by copy number state using the function *count.clonal.mutations()* with max.CN=4, excluding chromosomal segments with length less than $10^7$ bp. *count. clonal.mutations()* fits a binomial mixture model with success probabilities according to the expected mean values of the clonal VAF peaks, which, for an impure sample with tumour cell content $\rho$, are given by

$$\text{VAFs} \in \left\{ \frac{\rho}{\zeta}, \frac{(\text{CN} - b)\rho}{\zeta}, \frac{b\rho}{\zeta} \right\}, \qquad (1)$$

where CN denotes the copy number of a given segment, $b$ denotes the copy number of the minor allele (that is, the allele with the smallest number of copies) on this segment and

$$\zeta = \rho\text{CN} + 2(1 - \rho) \qquad (2)$$

is the average copy number of a given locus in the sample. Mutation densities (SSNVs per bp) at MRCA and ECA, denoted by $\widetilde{m}_{\text{MRCA}}$ and $\widetilde{m}_{\text{ECA}}$,

respectively, were computed using the function MRCA.ECA.quantification(). In brief, MRCA.ECA.quantification() first estimates $\widetilde{m}_{\text{MRCA}}$ from the number of all clonal mutations and the total size of the analysed genome, $g = \sum_l g_l$, where the index $l$ labels individual segments contributing to the analysis, yielding

$$\widetilde{m}_{\text{MRCA}} = \sum_l \frac{n_{1,l} + n_{\text{CN}-b,l}(\text{CN}_l - b_l) + n_{b,l}b_l}{g\,\text{CN}_l}, \qquad (3)$$

where $n_{k,l}$ denotes the number of clonal mutations present on $k$ copies of the $l$-th segment. Note that $\widetilde{m}_{\text{MRCA}}$ normalizes mutation densities per copy and hence can be interpreted as molecular time. The function MRCA.ECA.quantification() also computes lower and upper 95% confidence bounds for $\widetilde{m}_{\text{MRCA}}$ by bootstrapping the genomic segments 1,000 times. In the next step, MRCA.ECA.quantification() asks for evidence for an earlier common ancestor in the data. If a chromosomal gain occurs concomitantly with the onset of tumour growth, the density of amplified clonal mutations (that is, present on multiple copies of a gained allele) will, on average, be equal to the density of clonal mutations on non-amplified chromosomes or chromosomal regions. By contrast, if a chromosomal gain occurs earlier, the density of amplified clonal mutations on the gained allele will be smaller than the density of clonal mutations on non-amplified chromosomes or chromosomal regions. To distinguish these two cases, MRCA.ECA.quantification() tests for each gained segment whether or not the density of amplified clonal mutations agrees with the mutation density at MRCA, on the basis of a negative binomial distribution. If the mutation density of amplified clonal mutations is significantly smaller than the mutation density at MRCA, the segment is assigned to an earlier time point or else to the MRCA. MRCA.ECA.quantification() then asks whether segments assigned to earlier time points emerged in the same time window or during different time windows. We define as the null hypothesis that all CNVs emerged in an ECA. To test the null hypothesis, MRCA.ECA. quantification() computes the mutation densities at the ECA from the number of clonal mutations on the amplified minor allele, $n_{b,l}$ (if $b > 1$), and from the number of mutations on the amplified major allele, $n_{\text{CN}-b,l}$, as

$$\widetilde{m}_{\text{ECA}} = \frac{\sum_{l,P_{\text{adj},l} \le 0.01} n_{b,l} + n_{\text{CN}-b,l}}{\sum_{l,P_{\text{adj},l} \le 0.01} g_{l,b} + g_{l,\text{CN}-b}}, \qquad (4)$$

where $P_{\text{adj},l}$ is the adjusted $P$ value of segment $l$ belonging to the MRCA, and $g_{l,b}$ and $g_{l,\text{CN}-b}$ are the length of segment $l$, contributed by the amplified minor and major alleles, respectively. In analogy to $\widetilde{m}_{\text{MRCA}}$, MRCA.ECA.quantification() also estimates lower and upper 95% confidence bounds by bootstrapping. On the basis of a negative binomial distribution (and an FDR of 0.01), MRCA.ECA.quantification() then tests for each early segment whether or not its mutation density indeed conforms to a joined ECA, as defined by $\widetilde{m}_{\text{ECA}}$. Only segments with mutation densities conforming to $\widetilde{m}_{\text{ECA}}$ are assigned to the ECA, whereas all other segments are reported as conforming neither to the ECA nor to the MRCA.

Upon timing MRCA and ECA for each sample, we translated mutation densities into weeks post conception (p.c.) by inferring SSNV rates per diploid genome and embryonic day ($\mu\lambda$), using the measured VAF distributions and age at diagnosis as outlined below in section 'Real-time estimate of cell division rate'. As mutation calling was performed by comparing tumours against a matched blood control, mutation densities correlate with the time after gastrulation (at approximately 2 weeks after conception). Thus, the mutation density per haploid genome ($3.3 \times 10^9$ bp), $\widetilde{m}$, relates to the time p.c. according to $\widetilde{m}(t) = \frac{\mu\lambda}{\text{day}} \frac{1}{3.3 \times 10^9}(t - 14\,\text{d})$. The estimated time of birth was taken as 38 weeks after gastrulation (40 weeks p.c.).

## Timing of SNVs and small indels

We classified SNVs and small indels as subclonal or clonal on the basis of the number of variant reads, $n_{var}$, the number of reference reads, $n_{ref}$, tumour purity $\rho$ and copy number $k$. Specifically, mutations were classified as subclonal if the probability to sample at most $n_{var}$ variant reads out of $n_{var} + n_{ref}$ total reads according to a binomial distribution with success probability $\frac{\rho}{\rho k + 2(1-\rho)}$ was smaller than 5%. If a mutation was classified as clonal and fell on a region with $k = 3$, we moreover classified the mutation as early clonal (that is, acquired before the chromosomal gain on the gained chromosome and hence present on at least two copies) if the probability to sample at most $n_{var}$ variant reads out of $n_{var} + n_{ref}$ total reads was at least 5% according to a binomial distribution with success probability $\frac{2\rho}{\rho k + 2(1-\rho)}$, or else as late clonal.

## Modelling medulloblastoma initiation

We modelled medulloblastoma initiation and growth with a population genetics model originally developed for neuroblastoma, as described previously[23]. In brief, the model assumes that disease initiation is driven by two consecutive drivers in a transiently expanding tissue of origin, which for group 3/4 medulloblastoma is probably the population of unipolar brush cell progenitors (UBCPs[16,17,24]). The two driver events are associated with the ECA and the MRCA of the tumour, and spawn, respectively, a pre-malignant and the malignant tumour clone. We assumed that both drivers occur with small probabilities $\mu_1$ and $\mu_2$ during cell divisions, and confer a selective advantage ($r$ and $s$, respectively) that acts by reducing cell differentiation. Moreover, we assumed that UBCPs acquire on average $\mu$ neutral somatic variants per cell division, which we modelled with a Poisson process. The population of UBCPs has been experimentally described from week 9 p.c. until the time of birth[24,27]. To capture this trend, we modelled an initial phase of exponential growth at rate $\lambda_1 - \delta_1$, $\lambda_1 > \delta_1$ until time $T$, where $\lambda_1$ and $\delta_1$ denote the division and differentiation rate, respectively, and a subsequent phase of exponential decline at rate $\lambda_2 - \delta_2$, $\lambda_2 < \delta_2$.

Following refs. 23,61, we calculated the probability of the MRCA to occur at time $t$ according to

$$P_{MRCA} = \begin{cases} P_{MRCA,I}, & t < T \\ 1 - (1 - P_{MRCA,I})(1 - P_{MRCA,II})(1 - P_{MRCA,III}), & t \geq T' \end{cases} \quad (5)$$

with[40]

$$P_{MRCA,I}(t) = \sum_{x=1}^{N(t)-1} e^{-\mu_1(x-1)}(1 - e^{-\mu_1})\left(1 - \exp\left\{-\frac{\mu_1\mu_2\lambda_1 TN(t)F}{1 - \frac{\delta_1}{\lambda_1}}\right\}\right),$$

$$P_{MRCA,II}(t) = 1 - \exp\left(-\frac{\mu_1\mu_2\lambda_1\lambda_2 v_{2,D}T}{\lambda_2 - \delta_2/r}N(T)\{e^{(\lambda_2 - \delta_2/r)(t-T)} - 1\}\right),$$

$P_{MRCA,III}(t)$

$$= 1 - \exp\left(-\frac{\mu_1\mu_2\lambda_2^2 v_{2,D}N(T)}{\delta_2\left(\frac{1}{r} - 1\right)}\left\{\frac{e^{(\lambda_2 - \delta_2)(t-T)} - 1}{\lambda_2 - \delta_2} - \frac{e^{\left(\lambda_2 - \frac{\delta_2}{r}\right)(t-T)} - 1}{\lambda_2 - \frac{\delta_2}{r}}\right\}\right),$$

where $v_{2,D} = 1 - \frac{\delta_2}{s\lambda_2}$ is the survival probability of a cell undergoing the second oncogenic event while the population decays, $F = \int_0^1 v_{2,E}/(v_{2,E}z^\alpha)dz$, $v_{2,E} = 1 - \frac{\delta_1}{s\lambda_1}$ and $\alpha = \frac{\delta_1 - s\lambda_1}{\delta_1 - \lambda_1}$. Moreover, we calculated the probability of the ECA to occur at $t_1$, conditioned on the MRCA occurring at $t_2$, as described previously[23]

$$P(t_1|t_2) = \frac{t_1}{t_2}; t_1 < t_2 \leq T,$$

and

$$P(t_1|t_2) =$$
$$\frac{\Theta(T - t_1)\lambda_1\delta_2\left(1 - \frac{1}{r}\right)t_1 + \Theta(t_1 - T)[\lambda_2 + \lambda_1 T\delta_2(1 - 1/r) - \lambda_2 e^{\delta_2(1 - 1/r)(T-t_1)}]}{\lambda_2 + \lambda_1 T\delta_2(1 - 1/r) - \lambda_2 e^{\delta_2(1 - 1/r)(T-t_2)}}; \quad (6)$$
$$t_2 > T,$$

where $\Theta(\cdot)$ is the Heaviside step function and $0 \leq t_1 < t_2$.

To estimate the model parameters from the WGS data, we contrasted the probability of acquiring the first and second drivers with the measured distribution of SNV densities at ECA and MRCA in group 3/4 medulloblastomas using approximate Bayesian computation with sequential Monte-Carlo sampling (ABC-SMC), as implemented in pyABC[62]. We used a population size of 1,000 parameter sets and 25 SMC generations or $\varepsilon \leq 0.05$ as termination criteria. The model fit was performed in analogy to Körber et al.[23] (code and pseudo-code are available at https://github.com/kokonech/mbOncoAberrations). 95% posterior-probability bounds for the model fits were estimated by simulating the model with each sampled parameter set and cutting off 2.5% at each end of the simulated distribution.

## Modelling medulloblastoma growth

We modelled medulloblastoma growth from the MRCA as exponential growth with rate $\lambda_T - \delta_T$, where $\lambda_T$ denotes the division rate and $\delta_T$ the loss rate (owing to differentiation or death) in the tumour. Denoting the number of tumour cells with $N_T(t)$ and assuming that neutral mutations are on average acquired at a rate $\mu\lambda_T N_T(t)$ per haploid genome during tumour growth, the site frequency spectrum of neutral variants at $t_{end}$ is, on average[63],

$$S_k(i, \mu) = \int_0^{t_{end}} P_{1,i}(\lambda_T, \delta_T, t_{end} - t)\mu k\lambda_T N_T(t)dt, \quad (7)$$

where $k$ is the chromosomal copy number, and $P_{1,i}(\lambda_T, \delta_T, t_{end} - t)$ is the probability to grow from a single cell to a clone of size $i$ within a time span $t_{end} - t$, according to a supercritical linear birth–death process (for example, see ref. 64).

To estimate $\mu$ and $\delta_T/\lambda_T$ from the WGS data, we followed the strategy described previously[23] to compare $M_k(a, \mu)$, the cumulative allele frequency histogram of variants present in at least $a$ cells in regions with copy number $k$, given a mutation rate $\mu$, between model and data. Here,

$$M_k(a, \mu) = \sum_{i=a}^{b} S_k(i, \mu)$$
$$\approx \int_0^{t_{end}} \mu k\lambda_T N_T(t)\frac{P_{1,b}(\lambda_T, \delta_T, t_{end} - t) - P_{1,a}(\lambda_T, \delta_T, t_{end} - t)}{\log\beta(t_{end} - t)}dt, \quad (8)$$

$$\beta(t) = \frac{\lambda_T(e^{(\lambda_T - \delta_T)t} - 1)}{\lambda_T e^{(\lambda_T - \delta_T)t} - \delta_T}.$$

To this end, we used ABC-SMC[62] with a population size of 1,000 parameter sets and 25 generations or $\varepsilon \leq 0.05$ as termination criteria. To learn the dynamics of tumour growth with confidence, we included tumours with well-defined subclonal tails and no evidence for subclonal selection. Tumours were selected (Supplementary Table 5) on the basis of visual inspection of the VAF histograms, to remove cases with poor subclonal resolution. In addition, we removed cases without age information and those with evidence for subclonal selection as suggested by the evolutionary model implemented in Mobster[65] (setting auto-setup = 'FAST'), which we ran on autosomes and upon computing pseudo-heterozygous VAFs, $\widehat{VAF}$, defined as 50% of the mutant sample fraction, SF (hence, $\widehat{VAF} = \frac{\zeta}{2k}VAF$, where $k$ is the number of alleles carrying the mutation and VAF is the actual VAF). For the 35 retained group 3/4 medulloblastomas, we followed the strategy outlined by

Körber et al.[23] to estimate the model parameters from the measured VAF distribution (code and pseudo-code are available at https://github.com/kokonech/mbOncoAberrations).

## Real-time estimate of cell division rate

From the model fits of medulloblastoma initiation and growth to WGS data, we estimated differentiation/loss rates and mutation rates relative to the rate of cell divisions. To convert these estimates to real-time, we used the age distribution at diagnosis for calibration. In a first step, we estimated the cell division rate of UBCPs, $\lambda$, from the number of generations between gastrulation and MRCA plus the number of generations between MRCA and diagnosis ($t_D$), which can be inferred from the mutational burden in the tumour[23]. Recall the selective advantage of the growing tumour, $s$. With $\lambda_T = s\lambda$, this yields

$$\lambda = \frac{1}{t_D}\left(\frac{\widetilde{m}_{MRCA}}{\mu} + \frac{\log N_T(t_D)}{1 - \frac{\delta_T}{s\lambda}}\right) = \frac{1}{t_D}\left(\frac{\widetilde{m}_{MRCA}}{\mu} + \frac{\log N_T(t_D)}{\mu}\mu_{eff}\right), \qquad (9)$$

where we used the estimate for $\mu$ from the parameter inference for medulloblastoma initiation and the estimate for the effective mutation rate, $\mu_{eff} = \frac{\mu}{1 - \frac{\delta_T}{s\lambda}}$, from the parameter inference for medulloblastoma growth. Assuming a tumour mass in the order of a few cubic centimetres and hence $N_T(t_D) = 10^9$ cells, and defining $t_D$ as the age at diagnosis, $A$, plus, on average, 250 d of embryogenesis after gastrulation, we obtained for each tumour (labelled with index $i$) an estimate for the division rate with mean, $\langle\lambda_i\rangle = \frac{1}{2\langle\mu\rangle(A_i + 250\,\text{d})}(\langle 2\widetilde{m}_{MRCA,i}\rangle + \log 10^9\langle\mu_{eff,i}\rangle)$, and standard deviation,

$\sigma(\lambda_i) = \frac{1}{2\langle\mu\rangle(A_i + 250\,\text{d})}\left(\frac{2\langle\widetilde{m}_{MRCA,i}\rangle + \log 10^9\langle\mu_{eff,i}\rangle}{2\langle\mu\rangle}\sigma\langle 2\mu\rangle + \sigma(2\widetilde{m}_{MRCA,i}) + \sigma(\mu_{eff,i})\right)$, in

actual time (where the factor 2 accounts for the fact that $\langle\mu\rangle$ and $\widetilde{m}_{MRCA,i}$ measure the mutation rate and the mutation density, respectively, per haploid genome).

Finally, we computed the mutation rate per day during tumour initiation, by computing $\mu\lambda_{T,i}$, with associated uncertainty $\mu\Delta\lambda_{T,i} + \lambda_{T,i}\sigma(\mu)$, which relates the molecular clock to real-time. For this purpose, we averaged across the inferences from all tumours that went into the analysis.

## Reporting summary

Further information on research design is available in the Nature Portfolio Reporting Summary linked to this article.

## Data availability

The DNA whole-genome sequencing mutation results were integrated from the corresponding medulloblastoma molecular landscape study[2] deposited at the European Genome-Phenome Archive under accession number EGAS00001001953. Single-nucleus RNA and ATAC data are available at the GEO database under accession numbers GSE253557 and GSE253573. All raw images and processed data after cell segmentation from spatial transcriptomics experiments are available at the GEO database and can be accessed under accession number GSE252090. Variant calls from WGS data are available from Mendeley Data (https://doi.org/10.17632/g4r22w9pp8.1). scRNA/ATAC and WGS data analysis results can be visually inspected on the interactive web application http://kokonech.shinyapps.io/mbOncoAberrations.

## Code availability

All custom Python and R scripts as well as details about the external software environment applied during the data analysis are available at GitHub (https://github.com/kokonech/mbOncoAberrations) and at Zenodo (https://doi.org/10.5281/zenodo.15083518)[66].

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

**Acknowledgements** This project has received funding from the European Research Council (ERC) under the European Union's Horizon 2020 research and innovation programme (grant agreement no. 819894) and Seventh Framework Programme (FP7-2007-2013) (grant agreement no. 615253). The INFORM programme is financially supported by the German Cancer Research Center (DKFZ), several German health insurance companies, the German Cancer Consortium (DKTK), the German Federal Ministry of Education and Research (BMBF), the German Federal Ministry of Health (BMG), the Ministry of Science, Research and the Arts of the State of Baden-Württemberg (MWK BW), the German Cancer Aid (DKH), the German Childhood Cancer Foundation (DKS), RTL television and the aid organization BILD hilft e.V. (Ein Herz für Kinder), and a generous private donation of the Scheu family. The study was in parts funded by the European Horizon 2020 Programme—'iPC - individualizedPaediatricCure' (grant no. 826121) to S.M.P. and by The Medulloblastoma Initiative to L.M.K. and S.M.P. V.K. acknowledges funding from the Deutsche Forschungsgemeinschaft (DFG, German Research Foundation, project no. 526169089). K.J.W. was funded by the Mildred Scheel Career Center Frankfurt (Deutsche Krebshilfe). We also thank C. Maus, E. Wang, L. Weiser and G. Warsow for their highly dedicated support in data management and processing, and R. Autry, G. Balasubramanian, C. Previti and R. Kabbe for their sincere and dedicated contribution to the bioinformatics analyses.

**Author contributions** K.O., P.J. and V.K. performed the main data processing and multi-level bioinformatics analysis. A.R., M.B. and J.-P.M. contributed with spatial data generation and analysis. P.B.G.S., B.S., J.V., D.R.G., A.W. and L.M.K. prepared tumour materials. M.S., I.S., T.Y. and H.K. contributed to the project from an evolutionary biology perspective. K.S., M.B.-J., P.F., B.J., N.J., T.M., K.W.P., C.M.v.T., O.W., K.B., K.J.W., L.N., C.R., U.S., M.M., S.R. and D.T.W.J. provided

tumour tissue samples and feedback for the manuscript. A.K., K.R., F.W., S.T. and T.H. contributed to the design of the study. K.O., P.J., V.K., L.M.K. and S.M.P. prepared the figures and wrote the manuscript on the basis of feedback from all authors. L.M.K. and S.M.P. co-led the study.

**Funding** Open access funding provided by Deutsches Krebsforschungszentrum (DKFZ).

**Competing interests** C.M.v.T. participates on advisory boards for Alexion, Bayer and Novartis. The other authors declare no competing interests.

**Additional information**

**Correspondence and requests for materials** should be addressed to Lena M. Kutscher or Stefan M. Pfister.

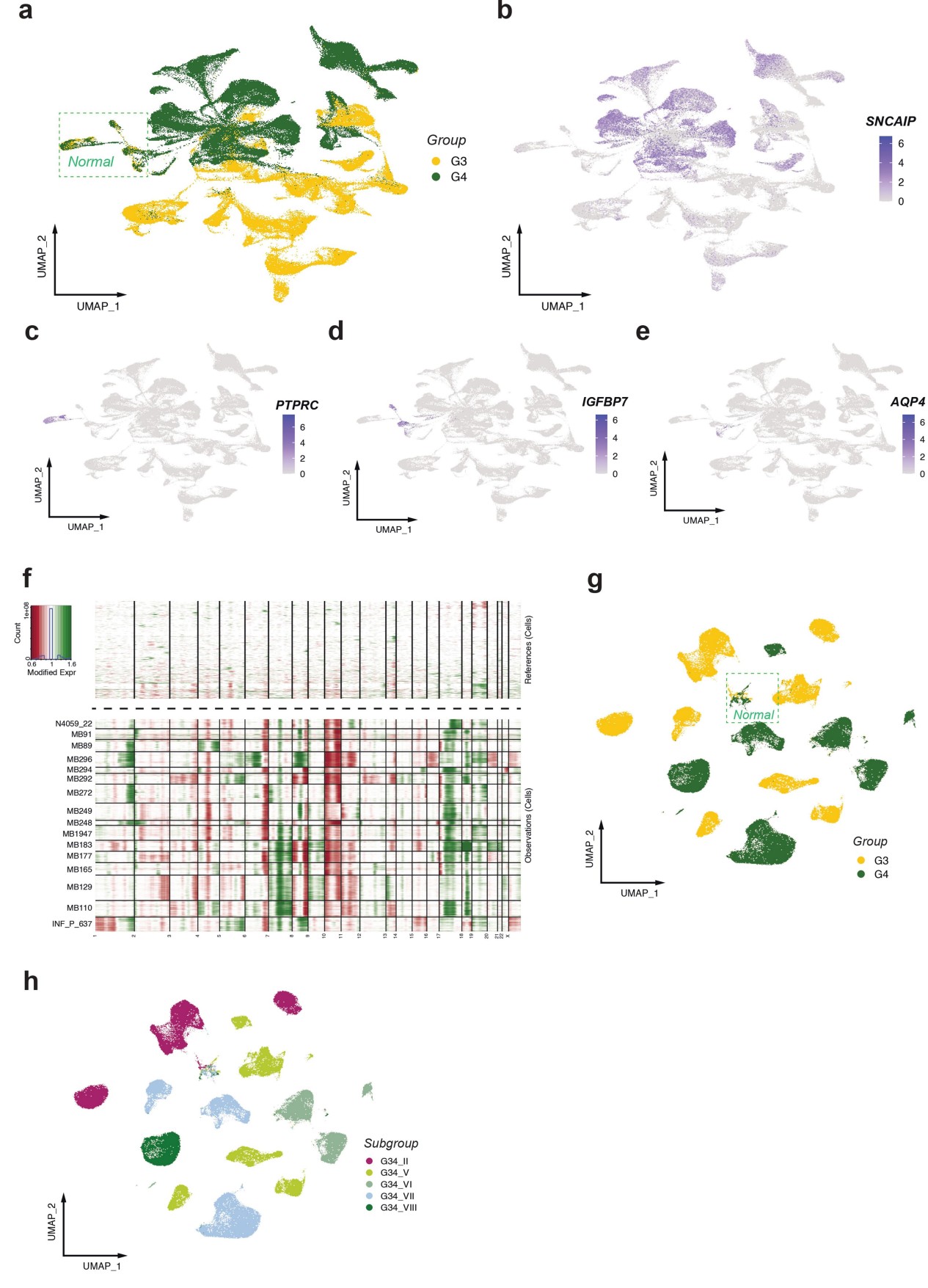

**Extended Data Fig. 1** | See next page for caption.

**Extended Data Fig. 1 | Group 3/4 medulloblastoma single-nuclei RNA and ATAC data properties.** a) UMAP of snRNA-seq merged dataset with medulloblastoma groups annotated. Non-tumor cells marked by dotted box. Feature plots showing b) *SNCAIP*, c) *PTPRC*, d) *IGFBP7* and e) *AQP4* expression within UMAP of merged snRNA-seq dataset. f) Global CNV profiles derived from snRNA-seq data. Top fragment: non-tumor cells. G) UMAP of snATAC-seq merged dataset with medulloblastoma groups annotated. Green box, normal cells. h) UMAP of snATAC-seq merged dataset with medulloblastoma subgroups annotated.

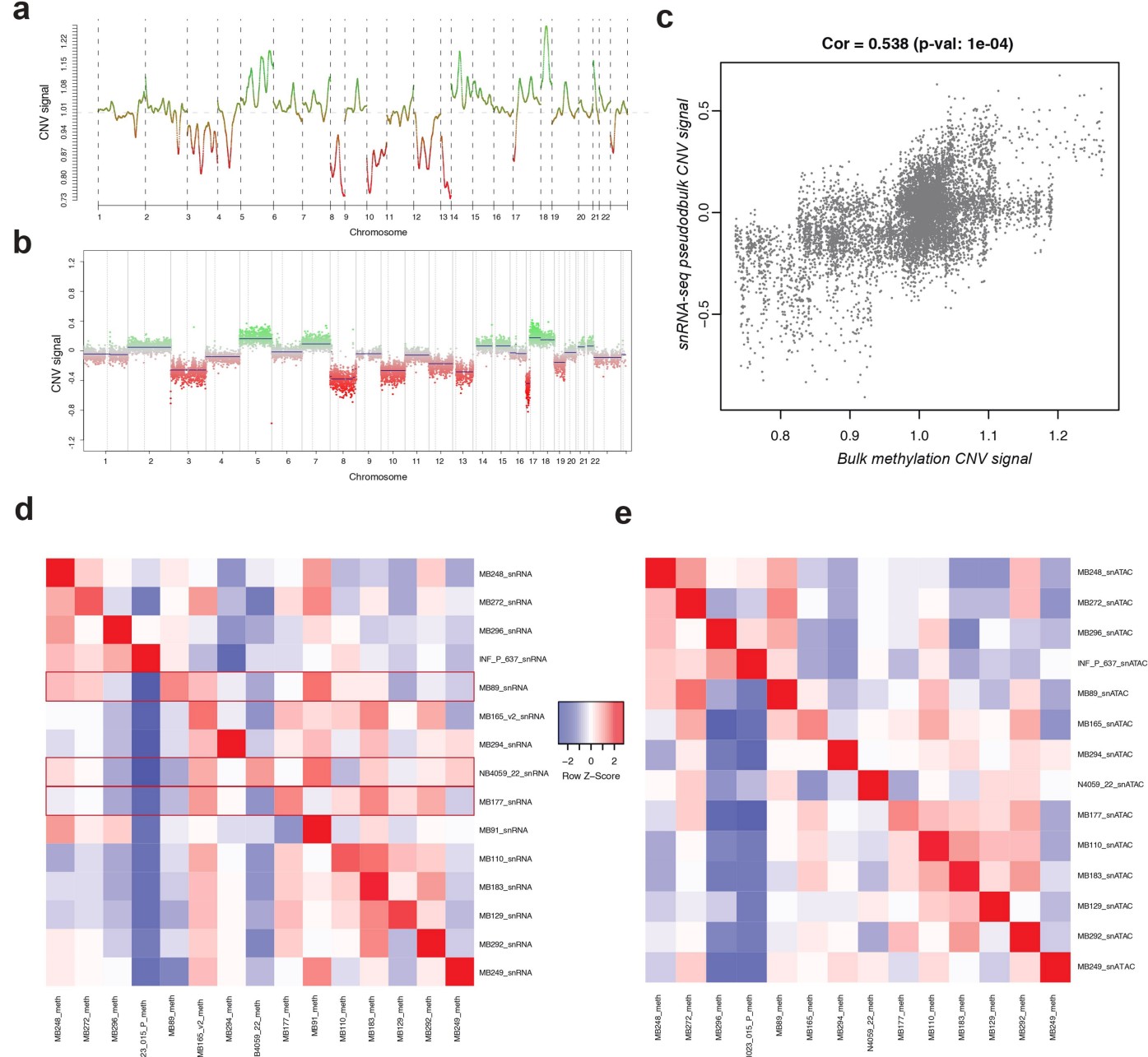

**Extended Data Fig. 2 | Validation of copy number profiling using single-nuclei RNA and ATAC data.** a) Merged pseudo-bulk CNV profile of snRNA-seq data from sample MB292 b) Methylation data-derived CNV profiles from sample MB292. c) Correlation plot of CNV values across 500 Kbp bins between snRNA-seq pseudo-bulk and methylation bulk profiles from sample MB292. d) Cross-comparison of snRNA-seq CNV profiles against bulk profiles. Red boxes, 3 cases where the highest correlation does not correspond to the same sample. e) Cross-comparison of snATAC-seq CNV profiles against bulk profiles.

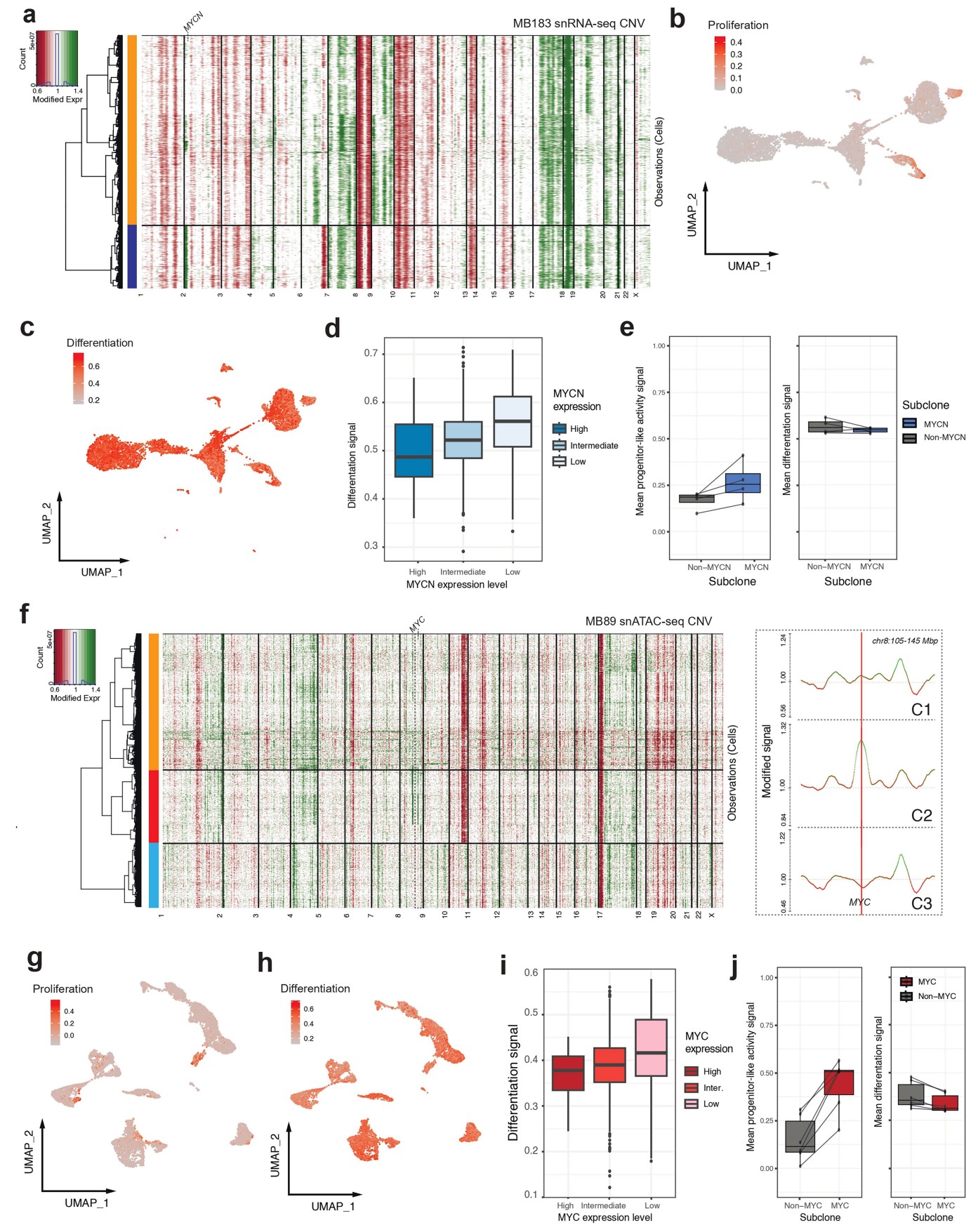

**Extended Data Fig. 3** | See next page for caption.

**Extended Data Fig. 3 | Copy number profiling of single-nuclei profiles from Group 3/4 *MYC*- and *MYCN*-amplified samples.** a) Copy number profile of snRNA-seq data from *MYCN* samples MB183. b,c) Per cell GSVA enrichment of proliferation (b) and differentiation (c) markers within UMAP of MB183 *MYCN*-amplified sample. d) Differentiation signal compared to ranked *MYCN* expression within *MYCN*-amplified subclone in sample MB183. *MYCN* normalized expression cutoffs: low = zero, intermediate > 0 and <2, high > 2. e) Boxplots showing difference in mean signal of progenitor-like activity (left) and differentiation (right) between *MYCN*-amplified and non-*MYCN*-amplified subclones in *n* = 4 tumor cases. f) Copy number profiles of snATAC-seq data from *MYC* sample MB89. Right side: zoom-in on MYC region. g,h) Per cell GSVA enrichment of proliferation (g) and differentiation (h) markers within UMAP of MB89 *MYC*-amplified sample. i) Differentiation signal compared to ranked *MYC* expression within *MYC*-specific sublclone in sample MB89. *MYC* normalized expression cutoffs: low = zero, intermediate > 0 and <2, high > 2. j) Boxplots showing difference in mean signal of progenitor-like activity (left side, t-test p-val: 0.003) and differentiation (right side) between *MYC*-amplified and non-*MYC*-amplified subclones in *n* = 6 tumor cases.

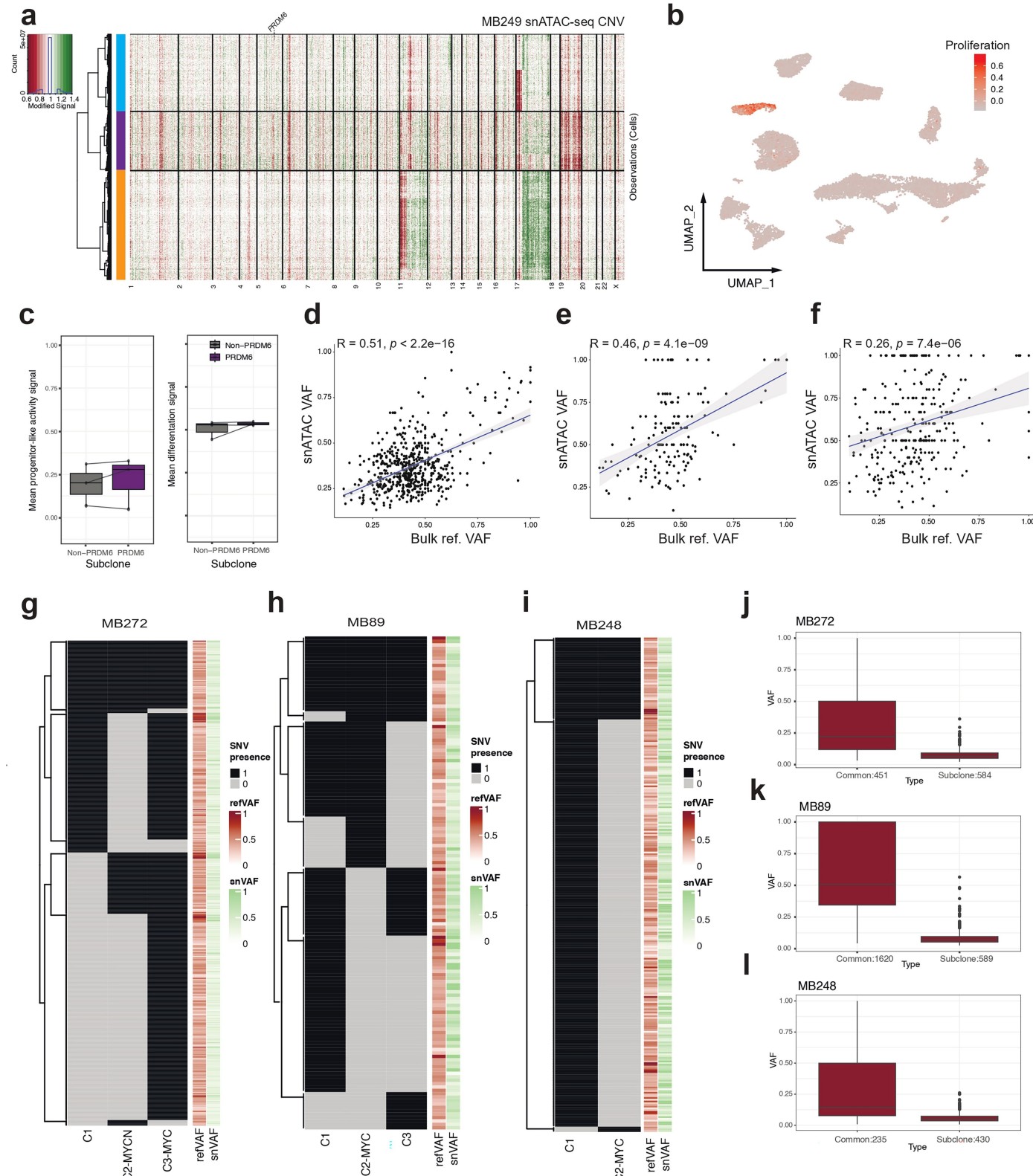

**Extended Data Fig. 4 | Copy number profiling of single nuclei profiles from Group 3/4 *PRDM6* samples.** a) Copy number profiles of snATAC-seq from *SCNAIP-PRDM6* sample MB249. b) Per cell GSVA enrichment of proliferation markers within UMAP of MB249 *PRDM6* sample. c) Boxplots showing the difference in the mean signal of progenitor-like activity (left) and differentiation (right) between *PRDM6-* and non-*PRDM6* subclones in *n* = 3 tumor cases. d-e) Correlation of VAF between mutations called from snATAC and bulk WGS data from *n* = 3 *MYC* cases: MB272 (d), MB89 (e), MB248 (f). g-i) Mutation heatmaps obtained via snATAC-seq data across subclones f from *n* = 3 *MYC* cases: MB272 (g), MB89(h), MB248 (i). j-l) Boxplots of comparison for number of subclone-specific *vs.* common mutations across snATAC profiles from *n* = 3 *MYC* cases: MB272 (j), MB89(k), MB248 (l) SNVs were filtered using bulk germline control and additional filtering parameters (see Methods for details).

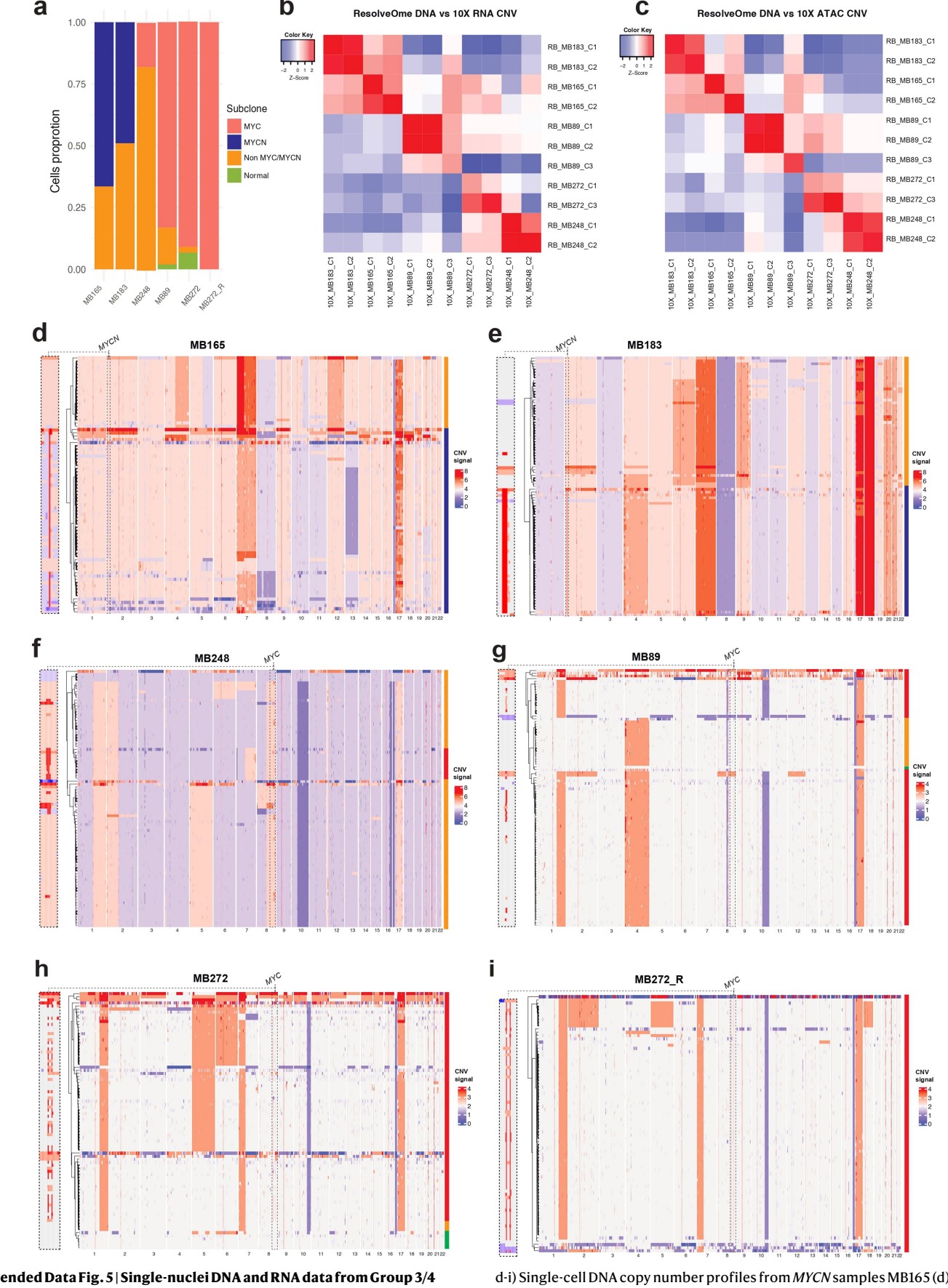

**Extended Data Fig. 5 | Single-nuclei DNA and RNA data from Group 3/4 samples confirm CNV subclones.** a) Projection of RNA data into annotation of subclones. b-c) Cross-comparison of snRNA-seq (b) and snATAC-seq (c) subclonal CNV pseudo-bulk profiles against scDNA subclonal CNV profiles.

d-i) Single-cell DNA copy number profiles from *MYCN* samples MB165 (d), MB183 (e), and *MYC* samples MB248 (f), MB89 (g), MB272 primary (h) and MB272 relapse (i). Zoom-in for target region (*MYC* or *MYCN*) is provided on the right side.

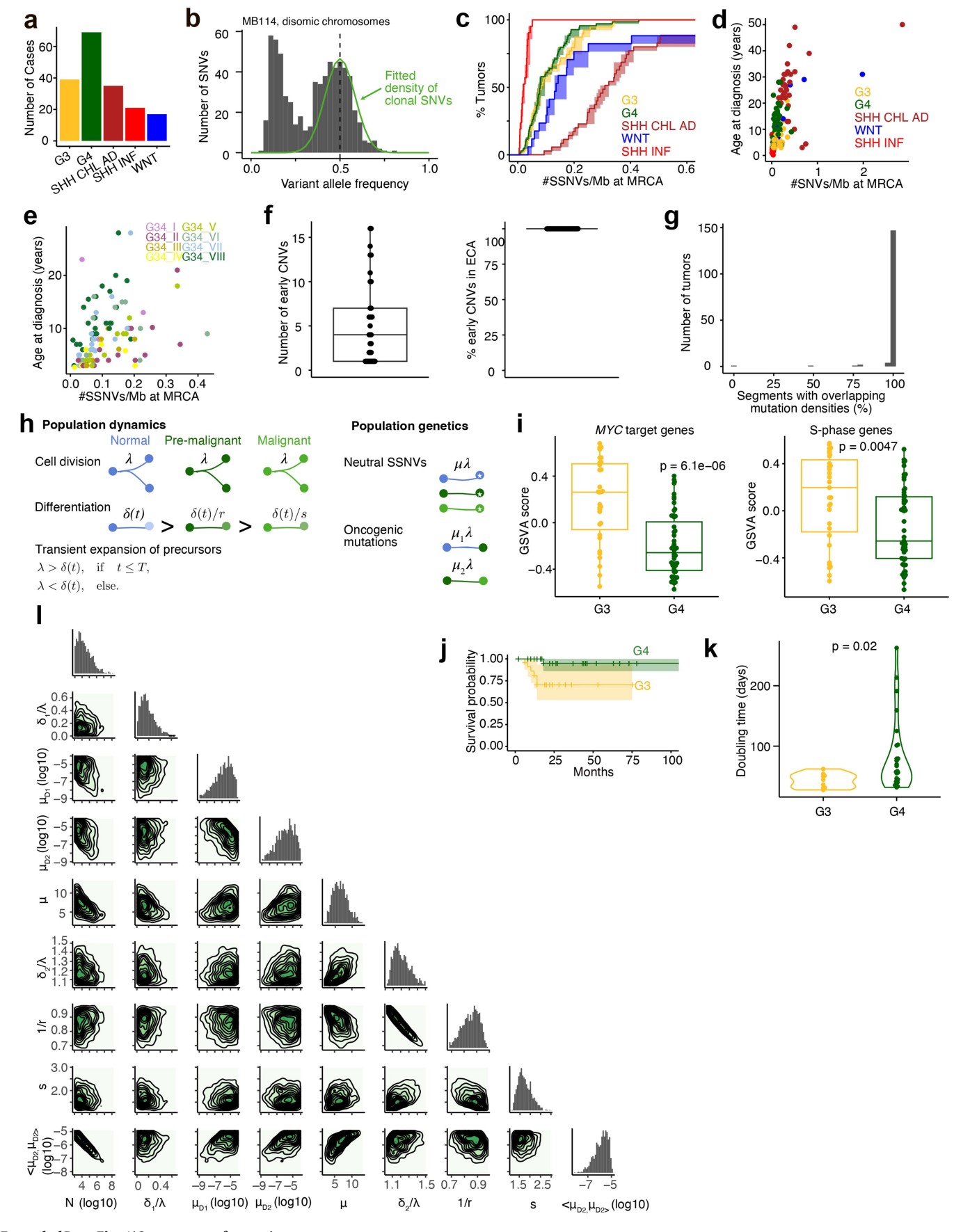

**Extended Data Fig. 6** | See next page for caption.

**Extended Data Fig. 6 | Early evolution of Group 3/4 medulloblastoma.**
a) Medulloblastoma samples analyzed by bulk WGS. b) SNV variant allele frequencies on disomic chromosomes for sample MB104 (G34_VIII). Green line, fitted clonal SNV density; dashed line, true clonal VAF estimated with ACEseq. c), SNV densities at MRCA (39 MB G3, 69 MB G4, 21 MB SHH INF, 35 MB SHH CHL/AD, and 17 MB WNT; 4 MB SHH CHL/AD and 2 MB WNT had clonal densities between 0.7 and 2.9 SNVs/Mb and are not shown). Mean and 95% CI (estimated by bootstrapping genomic segments 1,000 times). d) Mean SNV densities at MRCA versus age at diagnosis (n = 173 cases with age information). e) As in d but for G3/4 subgroups (n = 105 cases with age information). f) Left panel, number of early CNVs per tumor. Right panel, percentage of early CNVs with SNV densities agreeing with a single ECA. Data are from G3/4 medulloblastomas with evidence for an ECA. g), Comparison between mutation density estimates obtained in this paper and with MutationTimeR[22]. Estimates at gains/LOH were computed relative to the MRCA using both methods. Shown is the percentage of CNVs per tumor with overlapping 95% CIs. Data are from 38 MB G3, 66 MB G4, 15 MB SHH INF, 28 SHH CHL/AD, and 9 MB WNT with clonal gains/LOH at copy number ≤4 and at least $10^7$ bp length. h) Population genetics model of tumors initiation in two steps. i) GSVA scores for *MYC* target genes and S-phase genes (86 G3/4 medulloblastomas with RNAseq data). j) Overall survival of 23 Group 3 and 36 Group 4 medulloblastomas with available data. k) Doubling times estimated from 35 G3/4 medulloblastomas using the population-genetics model outlined in h). l) Posterior probabilities for the model fit to all G3/4 MBs (n = 108). $<\mu_1, \mu_1>$, geometric mean of the driver mutation rate.

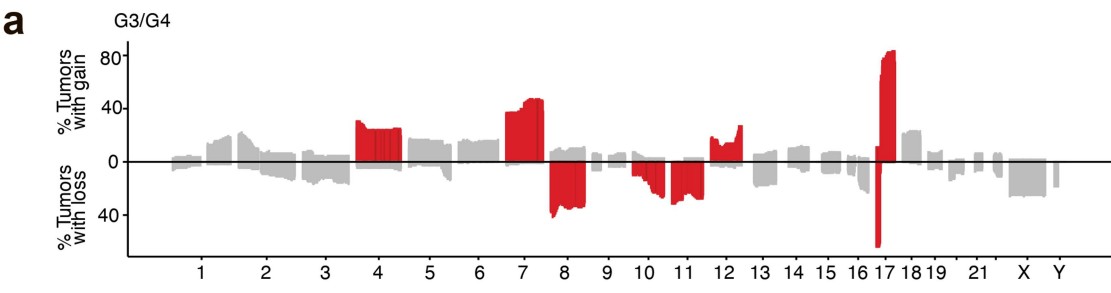

**a**

G3/G4

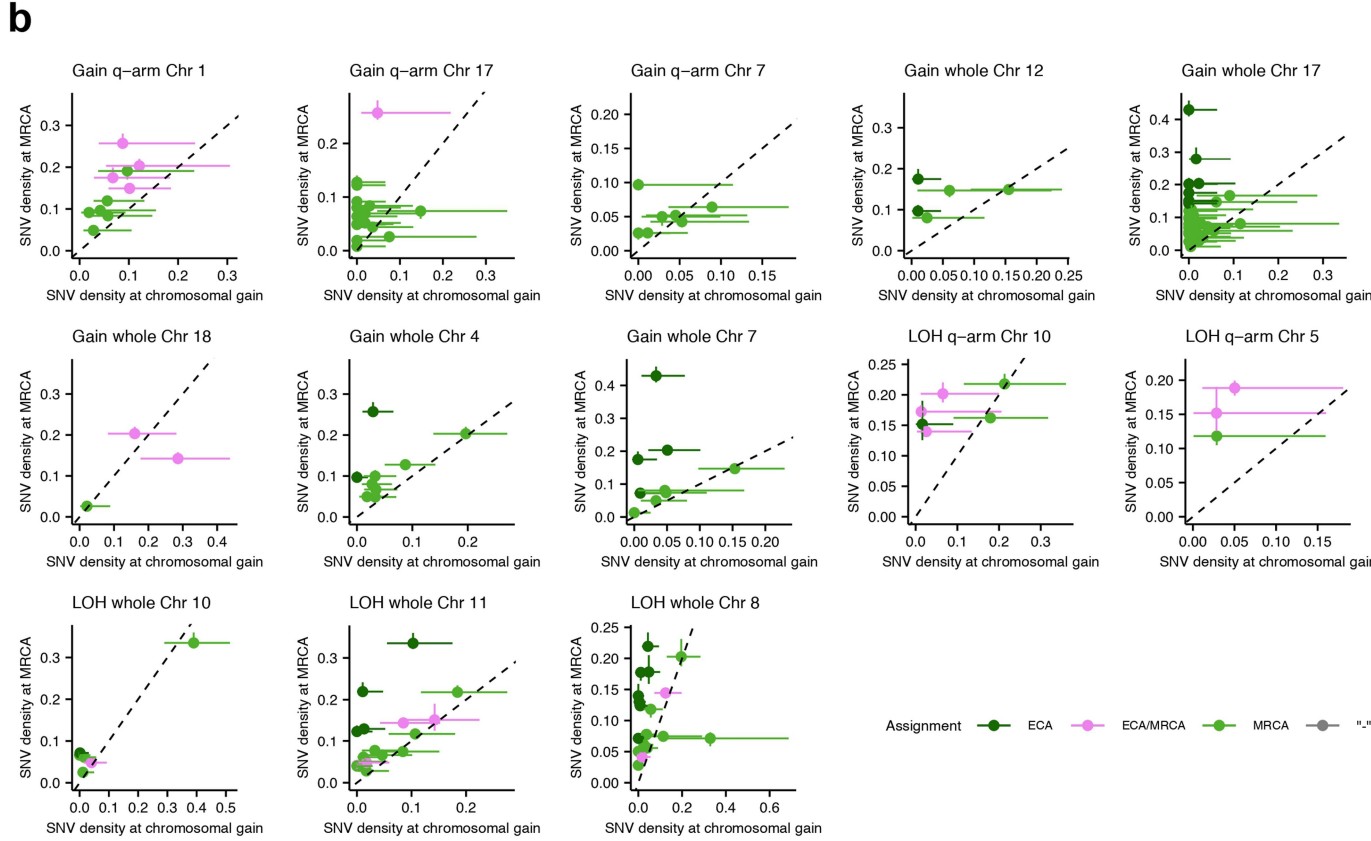

**Extended Data Fig. 7 | Clonal copy number changes in Group 3/4 medulloblastoma.** a) Percentage of tumors with copy number gains and losses ≥1 Mb along the genome. Red, regions where CNVs were significantly more frequent than expected, according to a Binomial test with $p_{adj} < 0.01$; Holm correction for multiple sampling. Shown are data from 109 Group 3/4 medulloblastomas. **b**, SNV densities at clonal chromosomal gains and at MRCA. Shown are mean and 95% confidence intervals (confidence intervals for SNV densities at chromosomal gains/LOH were estimated according to a Poisson distribution; confidence intervals for SNV densities at MRCA were estimated by bootstrapping genomic segments 1,000 times).

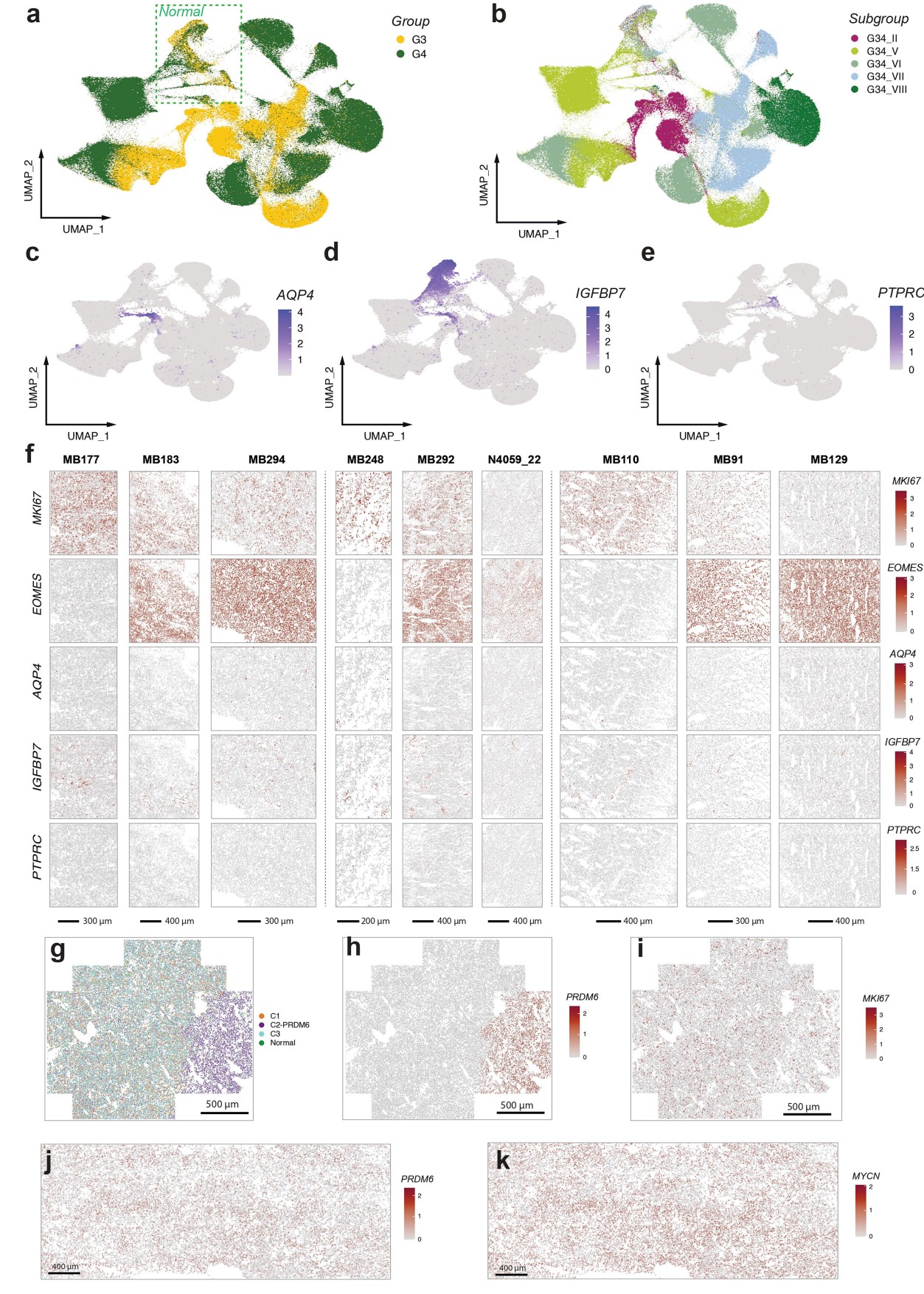

**Extended Data Fig. 8** | See next page for caption.

**Extended Data Fig. 8 | Spatial resolution of sub-clonal tumor populations.**
a) UMAP of spatial merged dataset with medulloblastoma groups annotation.
Normal cells marked. b) UMAP of spatial merged dataset with medulloblastoma
subgroups annotation. c-e) Feature plots showing c) *AQP4*, d) *IGFBP7* and
e) *PTPRC* expression within UMAP of merged spatial dataset. f) Spatial gene
expression of *MKI67, EOMES, AQP4, IGFBP7* and *PTPRC* across samples. g) Spatial
visualization of clones of PRDM6 sample in 2nd image fragment. h) *PRDM6* and
i) *MKI67* spatial expression of sample in g. j) *PRDM6* and k) *MYCN* spatial gene
expression in image fragment of sample MB292.

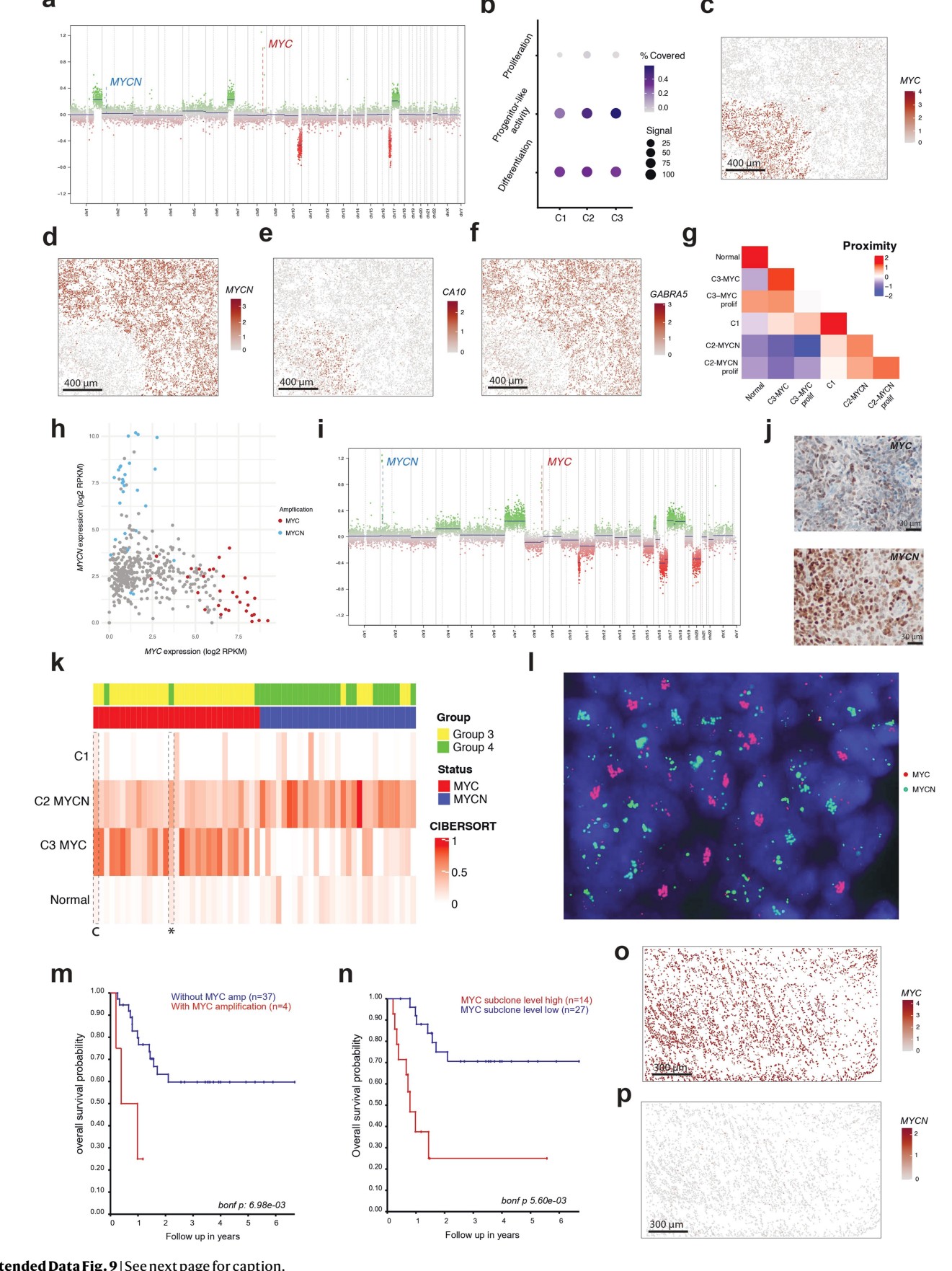

**Extended Data Fig. 9 |** See next page for caption.

**Extended Data Fig. 9 | Independent oncogene clones may co-occur in one tumor.** a) CNV profile of MB272 cases bulk methylation data. b) Per cell GSVA enrichments of proliferating, progenitor-like and differentiation in single sample MB272. Spatial expression of c) *MYC*, d) *MYCN*, e) *CA10* and f) *GABRA5* in sample MB272. g) Proximity of each compartment to each other in sample MB272 spatial data. h) Negative correlation (R = −0.287, *P* = 1.08e-09) between *MYC* and *MYCN* expression within medulloblastoma FFPE bulk RNA-seq cohort (*n* = 435). i) CNV profile of bulk methylation data from a Group 3/4 tumor with amplifications of *MYC* and *MYCN*. j) Identification of MYC (left) and MYCN (right) signals in the same sample using immunohistochemistry (IHC). k) CIBERSORT deconvolution results across subset of *MYC/MYCN* cases from medulloblastoma bulk FFPE RNA-seq cohort. MB272 single cell data with subclones annotation used as a reference, the data from control case is marked with c, target sample marked with asterisk. l) Identification of *MYC* (red) and *MYCN* (green) signals in the highlighted target Group 3/4 sample described in panel (k) using FISH. m) Kaplan–Meyer overall survival probability curves for medulloblastoma Subgroup V tumors with (red) and without (blue) *MYC* amplification as identified from bulk data CNV profiling. n) Kaplan–Meyer overall survival probability curves for medulloblastoma Subgroup V tumors with high (red) and low (blue) *MYC* subclone level enrichment. o) *MYC* and p) *MYCN* expression in spatial transcriptomic relapse sample.

Stefan M. Pfister

# Reporting Summary

## Statistics

For all statistical analyses, confirm that the following items are present in the figure legend, table legend, main text, or Methods section.

| n/a | Confirmed | |
|---|---|---|
| ☐ | ☒ | The exact sample size (*n*) for each experimental group/condition, given as a discrete number and unit of measurement |
| ☒ | ☐ | A statement on whether measurements were taken from distinct samples or whether the same sample was measured repeatedly |
| ☐ | ☒ | The statistical test(s) used AND whether they are one- or two-sided<br>*Only common tests should be described solely by name; describe more complex techniques in the Methods section.* |
| ☒ | ☐ | A description of all covariates tested |
| ☐ | ☒ | A description of any assumptions or corrections, such as tests of normality and adjustment for multiple comparisons |
| ☐ | ☒ | A full description of the statistical parameters including central tendency (e.g. means) or other basic estimates (e.g. regression coefficient) AND variation (e.g. standard deviation) or associated estimates of uncertainty (e.g. confidence intervals) |
| ☐ | ☒ | For null hypothesis testing, the test statistic (e.g. *F*, *t*, *r*) with confidence intervals, effect sizes, degrees of freedom and *P* value noted<br>*Give P values as exact values whenever suitable.* |
| ☒ | ☐ | For Bayesian analysis, information on the choice of priors and Markov chain Monte Carlo settings |
| ☒ | ☐ | For hierarchical and complex designs, identification of the appropriate level for tests and full reporting of outcomes |
| ☐ | ☒ | Estimates of effect sizes (e.g. Cohen's *d*, Pearson's *r*), indicating how they were calculated |

*Our web collection on statistics for biologists contains articles on many of the points above.*

## Software and code

Policy information about availability of computer code

| Data collection | Not applicable |
|---|---|

| Data analysis | Single nucleus RNA-sequencing and spatial data post analysis was performed in R 4.2 environment with the following packages: Seurat v4.1.3, dplyr v1.1.0, GSVA 1.46.0, Signac 1.10.0, InferCNV 1.10.1, scales_1.2.1 RImageJROI_0.1.2, data.table 1.14.6, stringr 1.5.0, Giotto v1.1.1., QuPath v0.3.2, ggplot2 3.4.0. |
| --- | --- |

Whole genome sequencig data analysis was performed in R v4.2 with the following packages:
ggplot2 v3.5.1, bedr v1.0.7, openxslx v4.2.5, NBevolution v0.0.0.9000, ggpubr v0.4.0, ggsci v2.9, pammtools v0.5.8, tidyverse v1.3.2, mixtools v1.2.0, RColorBrewer v1.1.3, ggbeeswarm v0.6.0, survminer v0.4.9, survival v3.4.0, GSA v1.3.3, GSVA v1.44.5, msigdb v1.4.0, ExperimentHub v2.4.0, GSEABase v1.58.0, ComplexHeatmap v2.13.1, wesanderson v0.3.6, GenomicRanges v1.48.0, ggbio v1.44.1, MutationTimeR v1.0.2, VariantAnnotation v1.42.1, cdata v1.2.0, cowplot v1.1.1, mobster v1.0.0, CNAqc v1.0.0, HDInterval v0.2.2, moments v0.14.1,

Approximate Bayesian Computation was run in python v3.10.1 using pyABC v0.12.6

Cell border detection for Resolve Bioscience spatial images was performed with CellPose 2.0.5. Gene expression counts were extracted with custom Python scripts in python3.2 environment.

The source code materials for data analysis are shared via the repository:
github.com/kokonech/mbOncoAberrations

For manuscripts utilizing custom algorithms or software that are central to the research but not yet described in published literature, software must be made available to editors and reviewers. We strongly encourage code deposition in a community repository (e.g. GitHub). See the Nature Portfolio guidelines for submitting code & software for further information.

# Data

Policy information about availability of data

All manuscripts must include a data availability statement. This statement should provide the following information, where applicable:
- Accession codes, unique identifiers, or web links for publicly available datasets
- A description of any restrictions on data availability
- For clinical datasets or third party data, please ensure that the statement adheres to our policy

The DNA whole genome sequencing mutation results were integrated from the corresponding medulloblastoma molecular landscape study deposited at European Genome-Phenome Archive under accession number EGAS00001001953. Single nuclei RNA and ATAC data available at GEO database under the accession numbers GSE253557 and GSE253573 accordingly. All raw images and processed data after cell segmentation from spatial transcriptomics experiments available at GEO database and can be accessed under the accession number GSE252090.

# Research involving human participants, their data, or biological material

Policy information about studies with human participants or human data. See also policy information about sex, gender (identity/presentation), and sexual orientation and race, ethnicity and racism.

| Reporting on sex and gender | Term sex is used for patients description, provided in annotation heatmap describing the target cohort. It is not used in the research. |
| --- | --- |
| Reporting on race, ethnicity, or other socially relevant groupings | The samples were collected from archives such as based on material availability, we do not have information on race or ethnicitiy for this cohort. |
| Population characteristics | The study focuses on medulloblastoma brain tumors, mean age : 5 years |
| Recruitment | Patients were included retrospectively based on the availability of fresh frozen tumour material. |
| Ethics oversight | This study was performed following the regulations of the ethics committee of the Medical Faculty of Heidelberg University. |

Note that full information on the approval of the study protocol must also be provided in the manuscript.

# Field-specific reporting

Please select the one below that is the best fit for your research. If you are not sure, read the appropriate sections before making your selection.

☒ Life sciences        ☐ Behavioural & social sciences        ☐ Ecological, evolutionary & environmental sciences

For a reference copy of the document with all sections, see nature.com/documents/nr-reporting-summary-flat.pdf

# Life sciences study design

All studies must disclose on these points even when the disclosure is negative.

| Sample size | n = 20 |
| --- | --- |

| | |
|---|---|
| Sample size | Sample size is based on the availability of fresh frozen material of previously diagnosed MB G34 tumours with specific focus on cases with MYC,MYCN and SNCAIP/PRDM6 somatic alternations. |
| Data exclusions | No data was excluded in this study except of specific subsets not fitting quality control limits (e.g. low quality cells from single cell RNA/ATAC/spatial data) |
| Replication | For each target somatic change (MYC/MYCN/PRDM6) at least n=3 samples were included. |
| Randomization | Not applicable since the analyses were performed on collected available tumour material. |
| Blinding | All analyses were performed on retrospectively collected tumour material. No analyses involving living organisms were performed. |

# Reporting for specific materials, systems and methods

We require information from authors about some types of materials, experimental systems and methods used in many studies. Here, indicate whether each material, system or method listed is relevant to your study. If you are not sure if a list item applies to your research, read the appropriate section before selecting a response.

## Materials & experimental systems

| n/a | Involved in the study |
|---|---|
| ☒ | ☐ Antibodies |
| ☒ | ☐ Eukaryotic cell lines |
| ☒ | ☐ Palaeontology and archaeology |
| ☒ | ☐ Animals and other organisms |
| ☒ | ☐ Clinical data |
| ☒ | ☐ Dual use research of concern |
| ☒ | ☐ Plants |

## Methods

| n/a | Involved in the study |
|---|---|
| ☒ | ☐ ChIP-seq |
| ☒ | ☐ Flow cytometry |
| ☒ | ☐ MRI-based neuroimaging |

## Plants

| | |
|---|---|
| Seed stocks | n/a |
| Novel plant genotypes | n/a |
| Authentication | n/a |

