## [Peer Review File · Nature]

Oncogene aberrations drive medulloblastoma progression, not initiation

Corresponding Author: Professor Stefan Pfister

Version 0:

Reviewer comments:

Referee #1

(Remarks to the Author)

Okonechnikov and colleagues profile 20 Group 3/4 medulloblastomas with single nucleus RNA-seq and ATAC-seq and (100 gene) spatial transcriptomics. They also reanalyze 183 ICGC medulloblastoma whole genomes.

From these analyses, they conclude that MYC and MYCN alterations are frequently subclonal, in contrast to chromosomal aberrations which are early, initiating events. Notably focal amplification (MYC or MYCN) positive are often spatially intermixed with their amplification-negative counterparts, and some tumors have both MYC and MYCN amplified in neighboring cells. They suggest that MYC amplification drives relapses, as primary tumors with subclonal MYC amplifications have poor survival, and 4 out of 4 MYC-amplified relapses have a clonal MYC amplifications.

In a second part of the paper, they re-analyze ICGC group 3/4 medulloblastoma WGS profiles to time clonal copy number gains and identify early clonal / amplified SNVs. Applying a two hit (ECA -> MRCA) model of medulloblastoma development, they identify a subset of tumors where they can associate one or more gains with an ECA. Integrating age, SNV density, and SNV copy number they estimate that some medulloblastomas were initiated as early as the first trimester of gestation.

Overall, the paper presents an interesting single cell resource on medulloblastoma. However, as it stands, the findings are not very conclusive and do not seem to substantially advance the field beyond the team's (considerable) previous body of work. Notably, many of the key claims in the abstract / title / headings are not supported by data, largely because correlative or anecdotal observations are over-interpreted (large scale copy number variation drives tumor growth, medulloblastoma oncogenes are essential for disease progression and therapy resistance, single-cell technologies enabling early detection and diagnosis etc).

The finding of MYC heterogeneity on its own seems not that surprising, given previous demonstrations using older technology (eg Qin Neuro Oncol 2022, Minasi Virchows 2023). The claim that some medulloblastomas arise during gestation or <1 year of life while plausible is not fully developed. Along these lines, it seems like the multi-omic dataset is not fully taken advantage of, for example the single nucleus RNA and ATAC could have been used to draw links between tumor heterogeneity and gene regulation. This could seem particularly important for gaining some insight into the group 3/4 medulloblastomas that do not focally amplify any oncogene, especially in the tumors where MYC / MYCN amplified and non-amplified cancer subclones co-exist.

In addition, I also have substantial technical concerns about both single cell and WGS analyses. The study also reads like at least two papers, as the WGS analysis is disjointed from the single cell part of the paper. Here, the analysis of mutation timing seems technically flawed or based on premises that are not fully justified or substantiated.

Major concerns:

- It has been long been known that many Group 3/4 medulloblastomas lack any focally amplified driver. What is driving the growth of these cancer cells?

The authors seem to propose that "large chromosomal copy changes" is the missing driver, implying that the simultaneous

loss or gain of hundreds or thousands of genes is responsible. But this seems implausible and perhaps less than satisfying. It would seem that the multi-omic data that the authors have assembled should provide an avenue towards addressing this exact question.

Specifically it would seem that the chromatin and expression patterns should help pinpoint a driver in the large regions that are gained or lost. The multi-clonal tumors in particular where MYC is amplified in only some of the cells could allow a near isogenic comparison. In the multi-clonal tumors, how do the cancer cells that do not amplify MYC compete? Are there noncoding mutations? Purely epigenetic alterations? These are the kinds of analyses I would expect with this kind of data.

In addition to analysis of their snRNA and snATAC data, the authors may want to try the BISCUT method (Taylor et al Nature 2023) to pinpoint genes.

- Timing of initiation estimates are interesting but hard to verify, contingent on modeling assumptions that while reasonable are also speculative and very (apropos the "A" in the ABC acronym) approximate. Are there any biologically important differences in tumors that arise during gestation? in terms of drivers? transcriptional states? Otherwise this just seems like an unfalsifiable thought exercise. It seems like there would be ways to orthogonally validate these predictions - for example are tumors with a late ECA or MRCA and also an early diagnosis, fast growing and aggressive? Presenting these data would make the data more convincing and potentially add to the clinical applicability.

- The premise of the ECA analysis seems questionable. First, this assumes that the first driver "hit" coincided with a gain. This would be true if the first hit was a gain or happened at the time of a gain. But not clear this would hold. First, the first hit could be SNV, indel, or loss. Second, gains are likely happening randomly during evolution, some early and some late. The authors do not seem to consider or (more importantly) rule out this possibility. In addition, to time the ECA, they puzzlingly average (early clonal) mutation density across all segments when computing the m_ECA (4). This additionally assumes that all gains happened at the same time.

Stepping back, it just seems difficult to estimate mutation timing with precision in such a low TMB tumor. It seems that ECA vs non-ECA patterns highlighted in Figure 3 are just noise from an underpowered analysis. Indeed most of the detectable ECA tumors seem to have higher TMB.

Technical concerns:

- I have major technical concerns about the phylogeny construction, on which some of the paper's main claims (early CNV, late MYC rest). The phylogeny construction is not described in detail in the methods. It also does not seem to be a phylogeny reconstruction per se but a hierarchical clustering of CNV matrices derived from snRNA and snATAC. Key details of this procedure are omitted, including bin sizes for RNA and ATAC, tree cutoffs, and annotations of tree leaves and nodes. Also the phylogenies are presented without showing the supporting RNA / ATAC data either in main figures or the supplement. It is thus hard to interpret or judge the quality of these findings.

- Related to above, how were tumor cells identified in snRNA and snATAC? Text mentions marker genes but this seems insufficient. The standard approach would be to identify cells with copy changes, but this is not described.

- Related to above, one explanation for MYC or NMYC heterogeneity could be ecDNA. In other words, MYC could be an initiating event but then its copy number fluctuates across the tumor because of non-Mendelian segregation. This could even lead to subclones where MYC is lost. How do the authors rule this out? Perhaps FISH could inform this.

- Also how were copy number altered samples integrated across snRNA and snATAC. Seurat, to the reviewer's knowledge does not account for aneuploidy. Do snRNA cells that map to snATAC clones have concordant copy profiles with those snATAC clones?

- What is the blue in Figure 2a ? hard to see the copy number variation and what is signal vs. noise, and there is quite a bit of noise, maybe zooming in on the MYCN region would help. The phylogeny should be overlaid.

- The ECA vs MRCA classification seems like a bespoke version of previous mutation timing approaches. The rationale of mutation timing analyses is that segments gained early in cancer evolution will have a lower density of amplified or early clonal mutations. It is unclear why the authors apply a bespoke approach to this analysis rather than using previous methods (such as Gerstung 2020 PCAWG)

- The bespoke approach applied by the authors categorizes segments as ECA vs MRCA, and imply that these gains coincide with the timing of a first vs. second driver, respectively. Without hard data to support this (not currently provided) it does not seem that this binary categorization is appropriate. It is more likely that gains occur asynchronously along a time continuum, each gain at a different time point, rather than at discrete moments coinciding with the acquisition of drivers. If the authors strongly feel that their data points to a different conclusion, it should be rigorously substantiated.

For example, do the authors assert that specific gains (such as 7 or 11 as suggested by the snRNA data) are the first / early driver? In that case the timing of these gains (not gains overall) should be analyzed. However, it is not clear what rules out chromosomal loss or LOH or a mutation as an early driver.

- It is unclear what is the power for timing copy gains in a tumor type like medulloblastoma with very few somatic SNV. Such

an analysis is only really powered for tumor types with high mutation burden, such as adult tumors that evolve across decades. The power to infer timing in a very low-TMB tumor like medulloblastoma seems very low. The authors should apply previously validated approaches for mutation timing and assess the power of these approaches and/or their approach in this low-TMB context.

- Related to above the highlighted differences between ECA vs non ECA tumors

- Figure 3k and l analyses don't make sense. Comparing the burden of CNVs to "focal mutations" seems like comparing apples to oranges - the biological significance is unclear. This analysis would likely give significant values for any tumor type. Also odd to use box plots for a discrete y axis (driver counts).

- How do growth estimates from site frequency spectrum analysis correlate with Ki67 or other (eg transcriptional) measures of grade?

- Text and captions does not clearly describe the data, how it was generated or analyzed. For example, it is unclear what is the source of phylogenies in Figs 1,2, and 5, presumably snATAC but unclear?

- According to the methods, a segment is assigned to ECA if the density of amplified clonal mutations is significantly less than the m_{MRCA} . Before getting into details of implementation, this does not make sense - the density of amplified mutations will always be less than the density of total mutations. Amplified clonal mutations are a subset of total clonal mutations.

- The methods are unclear which is partly due to imprecise or poorly defined jargon and/or mathematical notation and left out details. For example, "the number of B alleles on this segment (line 782). A B-allele is generally non-reference allele. The authors likely mean the germline homolog with the smaller copy number, often called the minor allele. However, whatever term the authors use, it should be defined, especially if it is non-standard usage. The reader has to also guess what is g_l and $g_l, CN-b$ (equation 4). There are many other such examples.

(Remarks on code availability)

The code is not runnable, partly because it points to file paths on a private file system.

for example:

/b06x-isilon/b06x-m/mbCSF/results/humanTumor/mbSpatial/perSampleV2

Referee #2

(Remarks to the Author)

Okonechnikov and colleagues report an interesting and important study of Group 3 and 4 medulloblastoma. These enigmatic tumors include the most aggressive subtypes of the disease. As indicated by the authors, we are beginning to understand the origins of these tumors (that to date have been less clear than other disease subtypes) but we know little about how the molecular genetic alterations in these tumors drive the pathological phenotype. The authors used single-cell technologies including snRNAseq, snATAC-seq and spatial transcriptomics to dissect cell populations among medulloblastomas with known alterations in established medulloblastoma oncogenes including MYC, MYCN, and PRDM6. There are a number of important and novel findings from this study that will be of broad interest. First, the study shows that large-scale chromosomal aberrations are early tumor initiating events, while single-gene oncogenic events are late events and typically sub-clonal. This is important because single gene events are often considered as therapeutic targets, but this may not be the case in these tumors. Understanding the timing of these alterations is key to distinguishing truncal and subclonal events. The authors show also that MYC alterations are clonal in certain instances of disease progression and may drive tumor development and treatment resistance. A particularly attractive aspect of the study is the use of spatial transcriptomics to define the topographical distribution of subclones that reveals clear segregation of the cells. Finally, by integrating the genetic findings with a population genetics model, the authors estimate that medulloblastoma initiation in cerebellar unipolar brush cell-lineages starts from the first gestational trimester. This report is likely to be of broad interest to the community and the data in general support the conclusions drawn. There are some questions that, if addressed, might further strengthen the manuscript.

→→→The authors discuss (and present in Fig. 2) the differentiation, proliferation and progenitor-like activity states of cells using snDNAseq. These data are interesting, but the presentation confusing. What precisely are the authors trying to say here? Since proliferation and differentiation subclones are seen in both the MYCN amplified and non-amplified subclones, then what is the relevance of MYCN to this phenotype? These clearly are different compartments, but what are these differences? No real discussion of this seems to be presented. There may be a decrease in the MYCN expression within the differentiation compartment but, in the context of a large set of genes that will be changing, what does this mean? Since the non-MYCN amplified subclone also displays differentiation, then what is the relevance of MYCN? Can this be better described and related to other changes in the transcriptomes. The authors use UMAPs for the various proliferation and differentiation states of cells but then (e.g., in Fig.2e) individual scale-dot heat graphs. Relating these two is not straightforward. It is worth noting that given the lack of apparent variance in the expression scores of C1 and C2 in the UMAP, these are not that informative.

The finding that large clonal CNVs are likely founding events is also interesting. But little discussion is made of what genes might be the focus of such events. If these large CNVs are driver events, then what genes are upregulated in their expression as a consequence? and how, if at all, do these genes relate to the progenitor, differentiation and other signatures studied. While the authors note MYC-amplified clones demonstrated strong enrichment of progenitor-like activity compared to non-MYC-amplified compartments and that differentially expressed genes specific to MYC-amplified subclones were enriched in known MYC target genes, it would be helpful to know if these are topographically enriched (or not) on chromosomal locations subject to CNV.

The data presented in Fig.3 and ext data Fig.4 are very interesting. To date, our understanding of the origins of medulloblastoma have largely come from extrapolating observations in mouse models. Therefore, timing disease onset in humans is important. As a side note, the authors may want to use nomenclature other than subgroup I-VIII in Fig.3h given that subgroup is also applied to Group 3 and 4 medulloblastoma. The authors note that they make the assumption that upon malignant transformation the tumor adopts exponential growth until age of diagnosis. This is not unreasonable, but it is also not inline with what is observed in mouse brain tumour models. In a number of these models, pinpointing transformation is difficult. Susceptible tissues initially undergo hyperplastic expansion and may remain in this state until progression to malignant state occurs. Some recognition of this and how it may affect assumptions is worth consideration. Does the overlap in timing of origin support a common lineage for Group 3 and 4 tumors?

Similar to my comment above, the authors classified gains of Chromosome 1q, 4, 7, 12, 17/17q and 18, as well as LOH on 5q, 8, 10/10q, 11 and 17p as putative drivers of Group 3 and 4 medulloblastoma initiation. However, no discussion is made of what genes on these CNVs might be clonally selecting this driver event. This is an important issue that warrants some discussion.

The spatial data presented in the manuscript are also very interesting; but other than a brief mention of proliferation e.g., Ki67 or differentiation there is a lack of clarity around what genes were selected for study and precisely why. The patterns of clones are also interesting, but the authors chose to restrict this analysis simply a description of clonal compartmentalisation. There are other important issues at play here. For example, how do proliferation and differentiation compartments relate to other microanatomical structure such as vessels that might provide a niche?

(Remarks on code availability)

Referee #3

(Remarks to the Author)

Okonechnikov, Joshi, Körber and colleagues present a manuscript focused on oncogene-amplified Group 3/4 medulloblastoma. Medulloblastoma is one of the more common types of childhood brain tumors, which are overall rare but pose significant clinical challenges. About one third of children with medulloblastoma will die from their disease. The authors present results on three sets of experiments: First, they describes the use of single cell RNA and ATAC-seq datasets of a relatively large cohort of 20 patient tumor samples. The authors infer copy number variations to define genetic subclones and putative phylogenetic trees. Amplifications of medulloblastoma oncogenes MYC, MYCN, and PRDM6 are often subclonal and occur late in tumor evolution. The authors conclude that oncogene amplifications drive disease progression and therapy resistance, but are likely not involved in the early steps of tumorigenesis. The second part describes the use of deep whole genome sequencing data from 183 medulloblastoma samples to utilize somatic variants as a molecular clock to define the time of onset for different medulloblastoma subgroups. The authors find that the earliest common ancestor of individual tumors likely formed between the first trimester of pregnancy and the first year of life. Finally, the authors apply spatial transcriptomics to a cohort of 17 patient samples to map the spatial relationship of molecular subclones defined in the first set of experiments and describe two major localization patterns in which subclones are either interspersed or segregated. Finally, rare mosaic medulloblastoma tumors that show amplifications of both MYC and MYCN in separate subclones are reported.

Overall, I find this study to be interesting as it offers a fresh look on medulloblastoma genome evolution from early initiating events to tumor relapse. The study is entirely based on data generated from patient samples, which in itself might be a valuable resource to the scientific community. The most intriguing part of the study are the molecular clock analyses which define a putative time of tumor onset. This is a clever approach that uses the density of somatic passenger mutations in genomic regions that subsequently undergo broad copy-number changes (which define the first oncogenic event/early common ancestor). The authors recently published this approach in a study focused on neuroblastoma (Körber et al, Nature Genetics 2023). The developmental origins of Group 3/4 medulloblastoma have recently been identified by correlative analyses to cell types in the developing human cerebellum, and an estimation of the time of tumor onset adds an important layer of information. That said, I have some concerns about the quality of some of the other analyses and the lack of technical validation as outlined in the specific comments below. The study is of interest to the field of medulloblastoma research and perhaps other pediatric cancers, but offers few insights for researchers outside this area of research.

1 - Single cell-derived CNV profiles appear very noisy and it is difficult to visually discern the variations that are being described. Very little information is provided in the methods on how inference of CNVs is performed. It is stated that five single cells are combined into a meta-cell based on their similarity (in the transcriptome space?) prior to inferCNV analysis to "improve the specificity". This approach is unorthodox and requires clarification and critical assessment.

2 - Bulk copy-number profiles (WGS or methylation array-based) should be placed adjacent to the single cell-derived profiles to allow for a better interpretation of potential artifacts of the approach. Are the same patients included in Figure 3h? Do clonal and subclonal CNVs match between the WGS and single cell datasets? Also it is generally difficult to appreciate the amplification status of said oncogenes. Other means of illustration should be explored. Limitations of the approach (inferring CNVs from RNA expression or open chromatin) also need to be more clearly discussed.

3 - It is unclear how phylogenetic trees were derived (the methods mention hierarchical clustering). More details need to be provided, and examples need to be shown. The presented results look sometimes arbitrary. For example, the first tumor presented in Figure 2a (MB183) has almost equal fractions of C1 and C2. Figure 1b shows 27% for C1. In the second tumor (MB177), I do not see on which basis C1 and C2 were separated, there seems to be a more obvious split. How robust and generalizable is the approach? How large (frequency/number of cells) does a subclone need to be in order to be detected? Are the same subclones detected if the data is randomly split into subsets? What is the minimum size of a CNV in order to be confidently distinguished between subclones?

4 - The UMAP clusterings provide strong evidence for the existence of major subclones in the presented samples. Based on these clusterings it appears that the number of subclones might be even higher than estimated based on inferred CNVs (e.g., in Figure 2g/k)? Comparing the differentiation state to tumor subclones is interesting, however a different scaling of values/colors would be appropriate.

5 - Are the oncogenes in question located on double minutes/ecDNA? The authors need to consider the possibility that double minutes are eliminated in cellular subsets because they are unequally distributed over daughter cells during cell division. This applies especially to the inferred phylogenies and the spatial analyses presented in Figure 4.

6 - The authors suggest that cellular migration may lead to interspersed cellular localization patterns of subclones, such as shown in Figure 4c. These results are based on RNA scope analysis of 100 target genes provided in Table 4. How were target genes selected, and how do they enable detection of genetic subclones? Or do patterns merely reflect the expression of MYCN? Have orthogonal assays for technical validation been considered? Please also state more clearly which instrument/technology has been used.

(Remarks on code availability)

Version 1:

Reviewer comments:

Referee #1

(Remarks to the Author)

The authors have adequately addressed my concerns, including technical issues. I congratulate them on an interesting dataset and story.

(Remarks on code availability)

Referee #2

(Remarks to the Author)

Okonechnikov and colleagues have made a thorough and careful attempt to answer the questions raised by the reviewers. I have no further issues that would improve the quality or support the conclusions of their manuscript significantly.

(Remarks on code availability)

Referee #3

(Remarks to the Author)

Okonechnikov and colleagues provide a revised manuscript describing their study of the subclonal genomic architecture of medulloblastoma. Their response has clarified some of my previous questions regarding the experimental and computational methods, which has allowed me to better judge the quality of the analyses. While I believe the study is interesting, I still have major concerns. I am not convinced that (i) the main data types used in the study (10x Genomics single cell RNA-seq and ATAC-seq) are suitable for inferring CNV profiles of sufficient quality, that (ii) some of the computational approaches used to analyze these datasets are appropriate, and that (iii) the critical assessment and validation of results is sufficient.

1 - The meta-cell approach, which is central to many of the analyses of the study, remains poorly justified and tested. A single sentence in the methods is not sufficient to describe the approach, and a single citation does not make it an

“established method”. Moreover, the cited publication does not study genetic subclones within the same tumor but uses the approach to separate all malignant from non-malignant cells. I understand that the code is available on GitHub, however my main issue is with the underlying assumptions and lack of critical assessment. My concern is that artifacts, which are inevitable when inferring CNV profiles from single cell RNA-seq and ATAC-seq datasets, are amplified using this approach, which in turn leads to wrongful definition of subclones.

2 - The sample that is showcased in Reviewer Figure 3.2 (sample MB272) is a good example of why I question if the data is suitable for inferring CNV profiles that are of sufficient quality for reliably defining subclones and phylogenies. The RNA-based analysis shows a loss of chromosome 10 (as annotated in the phylogenetic tree), which is not observed in the ATAC-based analysis (Figure 5a). The RNA-based analysis shows a gain of chromosome 19, whereas the ATAC-based analysis shows a loss of the same chromosome. There are many such examples in other samples. This puts the validity of the approach into question, especially when calling CNVs in smaller genomic regions or in subsets of cells. The novel analysis provided in Reviewer Figure 3.3 and in EDF 2 also indicates that inferred CNV profiles are associated with a high level of technical noise. In comparison, the newly added single cell DNA-based dataset is of higher quality, however in many cases CNVs and phylogenies do not match well with previous results and new results are not properly integrated into the manuscript.

3/4 - The approach to define genetic subclones and infer phylogenies as described in the revised manuscript is overly simplistic and relies heavily on manual inspection. This is an issue if CNV profiles are very noisy and subclone definitions are not obvious. How comparable are results obtained for the same sample from RNA and ATAC datasets? How do results compare to the newly generated DNA-based dataset? A systematic assessment is required.

5 - For this example (MB183) I cannot make out the co-occurring gain of chromosome 4 in the RNA-based analysis or the ATAC-based analysis. The described phylogeny appears to be more in line with the new DNA-based analysis. In any case, the question if described oncogenes are located on ecDNA remains important and other technical methods should be explored.

6 - My issue is with the finding of the interspersed cellular localization pattern of genetic subclones, as a contrast to the separated pattern. No reliable evidence is provided that the interspersed localization pattern (for example in MB165, Figure 4b-e) indeed represents different genetic subclones. Given the current evidence, it cannot be ruled out that cells of the same genetic subclone express varying levels of MYCN or “differentiation markers” that result in different genetic subclone annotations using the TransferAnchors approach.

(Remarks on code availability)

The authors state that code has been uploaded to GitHub.

Version 2:

Reviewer comments:

Referee #3

(Remarks to the Author)

Okonechnikov and colleagues have provided a rebuttal to my second round of comments that includes several additional figures and have made further clarifications within the manuscript. While some concerns regarding the inference of subclones from single-cell RNA and ATAC-seq profiles remain (see below), I believe that these can be addressed and that main findings of the study are sufficiently validated using orthogonal approaches.

My issue with the meta-cell approach for CNV analysis is that it assumes that cells within the same Seurat cluster (which is based on single-cell RNA or ATAC-seq profiles) share a similar CNV profile and hence can be combined into meta-cells. While this might be the case, it is also possible that cells from different subclones fall into the same cluster and are erroneously combined, thereby blurring out their signal. Looking at the results, I do not think that this is an issue that led to wrongful conclusions in the end. However, this is an unconventional approach that should be clearly laid out and justified to the reader (e.g., are major subclones likely to form separate clusters, how many RNA/ATAC clusters are detected per sample). Unless I have misunderstood the approach, in which case it needs to be explained more clearly, this is an important part of the analysis that is not “only applied to increase the computational speed and to improve the visualization”, as stated in the rebuttal. The definition of subclones and trees are dependent on it.

The added single-cell DNA sequencing analysis is an important validation of initial findings that highlights the validity and limitations of the RNA/ATAC-based approach. I agree that for most analyzed samples the inferred subclones match between both analyses. For MB272, the case that is characterized by an amplification of both MYC and MYCN, it appears that the amplification of MYCN on chromosome 2p is not detected in the DNA-based analysis or in the microarray-based CNV analysis of bulk DNA. Was a different piece of tumor tissue used for these analyses? The authors should clarify.

(Remarks on code availability)

Thank you for overall positive assessment of our manuscript, and for taking the time to read and provide insightful feedback. We have now addressed all concerns, and our responses are detailed below in blue.

Referee #1 (Remarks to the Author):

Okonechnikov and colleagues profile 20 Group 3/4 medulloblastomas with single nucleus RNA-seq and ATAC-seq and (100 gene) spatial transcriptomics. They also reanalyze 183 ICGC medulloblastoma whole genomes.

From these analyses, they conclude that MYC and MYCN alterations are frequently subclonal, in contrast to chromosomal aberrations which are early, initiating events. Notably focal amplification (MYC or MYCN) positive are often spatially intermixed with their amplification-negative counterparts, and some tumors have both MYC and MYCN amplified in neighboring cells. They suggest that MYC amplification drives relapses, as primary tumors with subclonal MYC amplifications have poor survival, and 4 out of 4 MYC-amplified relapses have a clonal MYC amplifications.

In a second part of the paper, they re-analyze ICGC group 3/4 medulloblastoma WGS profiles to time clonal copy number gains and identify early clonal / amplified SNVs. Applying a two hit (ECA -> MRCA) model of medulloblastoma development, they identify a subset of tumors where they can associate one or more gains with an ECA. Integrating age, SNV density, and SNV copy number they estimate that some medulloblastomas were initiated as early as the first trimester of gestation.

Overall, the paper presents an interesting single cell resource on medulloblastoma. However, as it stands, the findings are not very conclusive and do not seem to substantially advance the field beyond the team's (considerable) previous body of work. Notably, many of the key claims in the abstract / title / headings are not supported by data, largely because correlative or anecdotal observations are over-interpreted (large scale copy number variation drives tumor growth, medulloblastoma oncogenes are essential for disease progression and therapy resistance, single-cell technologies enabling early detection and diagnosis etc).

Reply: We thank the reviewer for their thorough feedback on our manuscript and appreciate the opportunity to clarify the novel findings of our study. Our present study utilizes for the first time single-cell multiomic sequencing and spatial transcriptomic data for Group 3/4 medulloblastoma to analyze the somatic subclonal architecture. In addition, we reconstruct the mutational evolution of these tumors including the spatial segregation of these subclonal populations. These findings are novel and supported by multiple lines of evidence. We have amended the discussion to highlight the novelty more clearly on pg. 17:

In our study we use single cell-multiomic data analysis combined with spatial profiling to identify the somatic subclone properties within these tumors and demonstrate that the genetic aberrations that lead to overexpression of oncogenes are not the initiating events in Group 3/4 medulloblastoma.

To account for the uncertainty with respect to causality, we have changed the title of the section "Large-scale copy number variation drivers early tumor growth" to "Large-scale copy number variation acquired early during tumor evolution".

More specifically, we have added significant orthogonal validation and alternative analytical approaches. First, we integrated single-nucleotide variant (SNV) calling on the single-nuclei multi-omics data to analyze the distribution of SNVs across subclones (details in updated Methods). For this purpose, we used known SNVs obtained from bulk whole genome sequencing (WGS) data as the reference (**Reviewer Table 1**). It is well established that

snRNA-seq using the 10X technology has limitations in detecting small mutations (Gupta et al, Cells, 2020), and indeed we observed that around only 4% (range 2% to 6%) of somatic mutations were detected across samples (**Reviewer Table 1**). However, snATAC-seq data recovered a much higher mean proportion of somatic SNVs, i.e., around 13% (range 11% to 22%). To further improve this sensitivity, we sequenced n=3 MYC-amplified cases at increased depth, from an average of 44.000 reads per cell originally to now 160.000 reads per cell, to obtain deeper coverage for improved mutation calling. This effort increased the detected SNVs to 40% (range 20% to 60%).

Reviewer Table 1. Summary of somatic mutation calling from single nuclei RNA, ATAC and deep coverage ATAC data in n=3 MYC cases. Initially mutations were obtained from bulk WGS data.

Sample	Bulk WGS	snRNA-seq	snATAC-seq	snATAC-seq deep coverage
MB272	919	56	206	561
MB89	757	28	86	178
MB248	1594	35	193	319

We also validated the variant allele frequency (VAF) of these snATAC-seq-detected somatic SNV events by comparing it to the observed VAF in the bulk WGS data for the same tumor sample. For the set of overlapping somatic SNV events, we observed a clear positive correlation in VAFs (**Reviewer Figure 1.1a-c**). Furthermore, we analyzed the distribution of these somatic SNV events across the identified subclones (**Reviewer Figure 1.1d-e**). We detected the presence of subclone-specific mutations in all cases, thus providing additional support for subclonal evolution.

Reviewer Figure 1.1 a-c) Correlation of VAF between mutations called from snATAC and bulk WGS data from $n=3$ MYC cases MB272 (a), MB89(b), and M248 (c). d-f) Mutation heatmaps obtained via snATAC data across subclones from $n=3$ MYC cases MB272 (d), MB89(e), and MB248 (f).

We have included these figures in the manuscript accordingly, on pg. 7, line 247:

We further investigated the mutational landscape at the single-cell level by increasing the coverage of the snATAC-seq data in three MYC-amplified cases. This approach allowed us to recover up to 40% (range 20% to 60%) of somatic mutations per sample (Extended Data Table 4). The positive correlation (max p -val: $7.4e-06$) of their variant allele frequency (VAF) and the corresponding bulk WGS profiles verified the accuracy of the approach (Extended Data Figure 4d-f). Importantly, in all three cases, it was possible to identify unique somatic mutations specific for the respective subclone, thus further supporting an initial common origin of these clones and secondary subclonal evolution (Extended Data Figure 4g-i).

Next, we applied an orthogonal experimental method capable of detecting chromosomal CNVs and RNA expression from the same cells using the ResolveOme technology in 4 MYC- and 2 MYCN-amplified cases (**Figure for reviewers 1.2a**, details in Methods), to verify the presence of subclones.

Reviewer Figure 1.2. a) Cohort of ResolveOme DNA and RNA protocol b) Projection of RNA data into annotation of subclones. c-h) Single cell DNA copy number profiles from MYCN samples MB165 (c), MB183 (d), and MYC samples MB248 (e), MB89 (f), MB272 primary (g) and MB272 relapse (h). Zoom-in for target region (MYC or MYCN) is provided on the right side.

Single-cell WGS profiles obtained from the experiment were used to perform CNV profiling, while RNA profiles were used to assign tumor and non-tumor cells based on marker gene expression, using the 10X Chromium snRNA-seq data as the reference. This projection of RNA data clearly confirmed the presence of the respective subclones (**Reviewer Figure 1.2b**).

Further, WGS CNV profiling verified the sub-clonality of *MYCN*- and *MYC*-amplified cells : in all primary tumors, we detected the presence of target-amplified and non-amplified subclones (**Reviewer Figure 1.2d-e**). However, in the MB272 primary sample, we detected the *MYC*-positive and negative subclones, but failed to detect the *MYCN*-positive subclone (**Reviewer Figure 1.2c**). This lack of detection is likely due to the spatial location of the tumor segment used for the experiment and relatively small number of cells (n=96). Nevertheless, the subclonal CNV profiles obtained from these experiments matched those obtained using the 10x Chromium multiomics data. For example, the subclonal gains in chr5 and chr6 that we observed in snRNA and snATAC data in MB272 *MYC*-subclone were verified (**Main Figure 5a,c**). Also, the analysis of MB272 relapse single-cell DNA CNVs fully reflected the presence of the *MYC*-amplification across all tumor cells (**Reviewer Figure 1.2d**), as was observed with the 10x multiomic data (**Figure 5i**).

We have included these new results in the manuscript on pg. 7, line 234:

To further verify the presence of subclones, we performed single-cell whole-genome DNA-sequencing and RNA-sequencing from the same cells, in a subset of MYCN- and MYC-amplified cases (Extended Data Figure 5a). In all tested samples, the presence of the corresponding specific subclones, and their CNV profile matched between the snMultiomic data based on single-cell RNA projection (Extended Data Fig. 5b) and WGS CNV analysis (Extended Data Fig. 5c-e).

The finding of *MYC* heterogeneity on its own seems not that surprising, given previous demonstrations using older technology (eg Qin Neuro Oncol 2022, Minasi Virchows 2023). The claim that some medulloblastomas arise during gestation or <1 year of life while plausible is not fully developed. Along these lines, it seems like the multi-omic dataset is not fully taken advantage of, for example the single nucleus RNA and ATAC could have been used to draw links between tumor heterogeneity and gene regulation. This could seem particularly important for gaining some insight into the group 3/4 medulloblastomas that do not focally amplify any oncogene, especially in the tumors where *MYC* / *MYCN* amplified and non-amplified cancer subclones co-exist.

Reply: Indeed, the presence of heterogeneity within *MYC*-amplified tumors was reported previously, and we have cited these papers in our manuscript accordingly. However, the origin of this heterogeneity in patient tumors was not previously investigated. Our analysis now provides evidence that *MYC* heterogeneity is evolutionary in nature. The novelty in this study is the absence of *MYC* amplifications in the initiating clones and the clonality of *MYC* at relapse in all cases investigated. Moreover, our study demonstrates how single-cell RNA and ATAC data can be used to distinguish subclones based on their inferred CNV profiles.

In parallel to this study, we investigated transcription-factor driven gene regulatory networks in Group 3/4 medulloblastoma, thus fully integrating single nuclei RNA and ATAC data (*bioRxiv*: 2024.02.09.579680). This manuscript is undergoing peer-review in parallel. Nevertheless, as suggested, we have performed additional in-depth analyses of the cis-regulatory elements associated with *MYC*/*MYCN* positive and negative clones and their association with subclonal marker genes, as explained in detail in our further responses below.

In addition, I also have substantial technical concerns about both single cell and WGS analyses. The study also reads like at least two papers, as the WGS analysis is disjointed from the single cell part of the paper. Here, the analysis of mutation timing seems technically flawed or based on premises that are not fully justified or substantiated.

Reply: The focus of our integrative analysis was to address the question of tumor origin, its evolutionary path and the role of oncogenes in tumor initiation, aggressiveness, and relapse. We used CNV profiles obtained from bulk and single-cell data as the core of the evidence for our conclusion – thus, both approaches share an underlying theme and complement each other.

To build an additional connection between the data, we further integrated the bulk and single-cell analyses using an additional data modality, namely by incorporating SNVs calling using snATAC-seq data (as described above). Moreover, we investigated the timing of mutations based on the inspection of VAFs between the subclones. Due to high noise associated with SNV calls from snATAC-seq data, we used more strict filtering settings for mutation calling to avoid false positives (details in Methods) and in parallel used germline WGS data from the same patient to remove non-somatic polymorphisms from the snATAC-seq results. With these adjustments, we recovered approximately 4x the amount of somatic mutations, and further inspected the VAF difference between mutations common among subclones or specific for an individual subclone. We observed that the VAFs in subclones (mean: 0.078) are approximately 4x lower than VAFs with common mutations (mean: 0.338) as shown in **Reviewer Figure 1.3**.

Reviewer Figure 1.3. Boxplots of comparison for variant allele frequency (VAF) between subclone-specific and common mutations across snATAC profiles from n=3 MYC cases MB272 (a), MB89(b), MB248 (c).

This observation confirms our hypothesis that these subclones arise later in tumor evolution, and establishes an additional connection between the different parts of our study. We updated this result in the manuscript accordingly on pg. 8, line 255.

Moreover, by comparing the extended somatic SNVs identified in the snATAC data with germline SNVs (see Methods for details), the subclone-specific SNVs were observed to have approximately 4x lower mean VAF in comparison to those SNVs that are common among subclones (Extended Data Fig. 4k-m).

Major concerns:

- It has been long been known that many Group 3/4 medulloblastomas lack any focally amplified driver. What is driving the growth of these cancer cells? The authors seem to propose that "large chromosomal copy changes" is the missing driver, implying that the simultaneous loss or gain of hundreds or thousands of genes is responsible. But this seems implausible and perhaps less than satisfying. It would seem that the multi-omic

data that the authors have assembled should provide an avenue towards addressing this exact question.

Reply: Indeed, the question of tumor initiation, in the absence of oncogenic drivers, is a perplexing one. Genomic analysis at the bulk level has previously shown that many medulloblastoma samples lack previously characterized alterations in known driver oncogenes, but instead seem to harbor large-scale chromosomal aberrations only (Jones et al, Nature, 2012). However, our comprehensive analysis has now shown that even in the case of tumors associated with a driver oncogene, the initiator events are likely to be large-scale CNVs and not the alteration in the oncogene driver itself. There is no evidence so far for any gene-specific aberrations to initiate these tumors, making the hypothesis of large CNVs initiating these tumors indeed more and more plausible.

We now also demonstrate that the specific copy-number alterations are non-random, and that associated gene expression changes are also consistent across multiple samples. We analyzed published bulk transcriptomic data (Cavalli et al, Cancer Cell, 2017) of 405 Group 3/4 tumor samples with CNV metadata. In this analysis, we identified genes up (or down) regulated concurrent to gain (or loss) of chromosomal segments belonging to chromosomes 1, 7, 10 and 17, across hundreds of samples. We identified that around ~10% of the differentially up (or down) regulated genes were located on the gained (or lost) chromosomal arm, highlighting the contribution of the CNV event to driving gene expression changes within the tumor. Genes that were upregulated due to chromosomal segment gain often included known oncogenes, such as *EZH2*, *MET*, *HIP1* (associated with chr7q gain) and *STAT3*, *STATB5* (chr17q gain). Conversely, loss of chromosomal segments also lead to the downregulation of tumor suppressor genes, including, *GAS7*, *NCOR1* and *TP53* (loss of chr17p) and *MGMT* and *SUFU* (loss of chr10q). We now provide this analysis as additional **Extended Data Table 6a**.

Importantly, on global level we clearly observed statistically significant enrichments of the noted CNV changes as we also state in the manuscript (**Extended Data Figure 7a**). In summary, we can conclude that chromosomal CNV events could drive the molecular signature of tumors through a predominant contribution to differentially expressed genes.

Specifically it would seem that the chromatin and expression patterns should help pinpoint a driver in the large regions that are gained or lost. The multi-clonal tumors in particular where *MYC* is amplified in only some of the cells could allow a near isogenic comparison. In the multi-clonal tumors, how do the cancer cells that do not amplify *MYC* compete? Are there noncoding mutations? Purely epigenetic alterations? These are the kinds of analyses I would expect with this kind of data.

Reply: In order to integrate the single-nuclei multi-omics data in our study, we focused on the *MYC/MYCN/PRDM6* cases within this manuscript. We used the ArchR method (*Granja et al., Nat Genet., 2021*) to combine gene expression and gene regulatory normalized signals to find cis-regulatory elements (CREs) as well as genes correlated with them for each tumor sample. Afterwards, we performed a stricter selection of the subclone-specific signals by stating a minimum number of tumor samples (n=3) to represent the corresponding groups (*MYC/MYCN/PRDM6*-subclones as well as oncogene-negative subclones). The resulting data are included into our manuscript as **Extended Data Table 3** and the results are discussed further below.

Based on your suggestions, we first investigated differentially expressed gene signatures that drive the identity of the *MYC/MYCN* compartments as well as oncogene-negative subclones. In particular, we systematically identified subclone-specific differentially expressed genes (DEGs) that are supported at least by n=2 cases (**Extended Data Table 2**). We found a clear

enrichment of neuroinflammation activity in *MYCN*-subclones, while rRNA and ribosome biogenesis were strongly enriched in *MYC*-specific subclones known to be associated with these somatic factors. *PRDM6* subclones demonstrated an association with forebrain development-related genes. The non-*MYCN*, non-*MYC* and non-*PRDM6* subclone DEGs showed enrichment in nervous system development. We further checked the enrichment of normal cell types of the cerebellum (using snAtlas from Sepp, et al, Nature, 2023) within subclones without oncogene activation. We found that these subclones are clearly enriched in the development of unipolar brush cells (nonMYC $p < 1e-12$, nonMYCN $p < 1e-06$), the proposed cell-of-origin for Group 3/4 tumors (Hendrikse et al, Nature, 2022); we did not observe this enrichment in MYC/MYC subclones ($p > 0.05$).

We also investigated the genomic locations of the identified subclone-specific CREs but did not find any enrichment of somatic mutations specific for non-MYC and non-MYCN subclones. We have also checked genes associated to CREs in oncogene positive and negative compartments (see Methods, **Extended Data Table 3**). Interestingly, non-MYC/N as well as PRDM6/non-PRDM6 subclone-specific CRE associated genes mostly showed enrichment in neurogenesis-related genes. We also performed the investigation of association of non-MYC/MYCN subclones to the normal cell types activity by using the cerebellum cell type markers as the main reference. We observed again, that non-MYC/MYCN subclones were mostly enriched in the similarity to differentiating unipolar brush cells (enrichment p-values nonMYC: $3.217e-07$, nonMYCN: $1.068e-07$). At the same time MYC/MYCN subclones did not show this effect (enrichment p-values > 0.05). Therefore, the critical unique property of non-MYC/MYCN subclones could be closer correspondence to the cells-of-origin. We state this new result in the manuscript on pg. 5, line 177.

By inspecting differentially expressed genes specific for each subclone (Extended Data Table 2) we also identified a unique property of non-MYCN subclones with stronger enrichment of markers of unipolar brush cells (p -val: $1.14e-06$), the cell-of-origin of medulloblastoma Group 3/4, while MYCN subclone-associated genes did not show this property (p -val > 0.05). This observation was also confirmed using subclone-specific genes associated with cis-regulatory elements derived from the integration of snATAC-seq data (Extended Data Table 3).

In the absence of any putative driver mutations, the only plausible cause seems to be the gain/loss of chromosomal segments where unknown drivers lie. Experimental validation will be required to identify what combinatorial set of gain/loss induces tumor initiation in unipolar brush cell progenitors and understand the dynamics of their interactions in terms of outcompeting or getting outcompeted by oncogene driver events. Both follow-up validations are entire studies within themselves, and are outside the scope of our current investigation.

In addition to analysis of their snRNA and snATAC data, the authors may want to try the BISCUT method (Taylor et al Nature 2023) to pinpoint genes.

Reply: We are thankful to the reviewer for this suggestion. The BISCUT method was designed to identify critical genes based on the inspection of tumor-specific telomere- or centromere-bound regions from bulk CNV profiles. However, the tool itself did not work properly on our input data (CNV profiles from ICGC MB cases) due to an error resulting in the program fail and unfortunately we did not get any replies from the tool developers regarding this challenge (<https://github.com/beroukhim-lab/BISCUT-py3/issues/7>). However, we inspected in detail

their full TCGA shared results listing critical tumor genes (n=1688). In particular, we performed an additional inspection of genes that are specific for common gains and losses detected from our global CNV analysis (**Extended Data Figure 7a**). For this task, we focused on the corresponding regions and identified differentially expressed genes within these regions that are specific for Group 3/4 medulloblastoma compared with other central nervous system tumors, using a bulk gene expression dataset covering 43 central nervous system tumor classes in n=2482 samples (Okonechnikov et al, Neurooncology, 2023). In particular, it was possible to identify 394 genes overexpressed and located within common regions of copy-number gains specific for Group 3/4 medulloblastoma as well as 261 genes inactive and located in deleted regions specific for Group 3/4 medulloblastoma. We further investigated the known biological properties of these genes and identified an overlap of n=29 (~5%) of genes with BISCUT-derived driver genes. This new result is now included in the corresponding **Extended Data Table 6b** as an additional column and we noted this result in the manuscript together with the previously mentioned **Extended Data Table 6a** on pg. 12, line 389:

To search for putative drivers located on these regions, we analyzed the expression of genes lying in commonly gained or lost regions. For this purpose we contrasted gene expression between tumors with and without particular CNVs in a bulk RNA-seq cohort of Group 3/4 medulloblastoma (Cavalli et al, Cancer Cell, 2017; Extended Data Table 6a) and inspected differentially expressed genes specific for Group 3/4 medulloblastoma in global CNS tumor cohort (Okonechnikov et al, Neurooncology, 2023; Extended Data Table 6b). Among these identified genes, up to 10% have been described as known somatic drivers (Sondka, Z. et al., Nature Reviews Cancer, 2018), thus associated with medulloblastoma evolution.

- Timing of initiation estimates are interesting but hard to verify, contingent on modeling assumptions that while reasonable are also speculative and very (apropos the "A" in the ABC acronym) approximate. Are there any biologically important differences in tumors that arise during gestation? in terms of drivers? transcriptional states? Otherwise this just seems like an unfalsifiable thought exercise. It seems like there would be ways to orthogonally validate these predictions - for example are tumors with a late ECA or MRCA and also an early diagnosis, fast growing and aggressive? Presenting these data would make the data more convincing and potentially add to the clinical applicability.

Reply: We thank the reviewer for their thought-provoking feedback. Our timing estimates rely on the mutation densities at a tumor's MRCA/ECA, and on their conversion into actual time. To gain further confidence in our timing estimates, we performed additional analyses to validate the estimated mutation densities at ECA/MRCA with an orthogonal method, and to validate the dynamic parameter estimates underlying our real-time estimates of tumor onset with transcriptome data.

To validate the estimated mutation densities at ECA/MRCA, we compared our estimated mutation densities to results obtained with MutationTimeR, a computational method that times copy number changes relative to a tumor's MRCA (Gerstung et al., Nature, 2020). While similar to our approach in principle, MutationTimeR uses the ratio between non-amplified and amplified clonal SNVs to time clonal copy number gains relative to the tumor's MRCA. By contrast, our method computes absolute mutation counts at chromosomal gains and at a tumor's MRCA from the SNV counts and then compares these estimates to each other. Assignment of a chromosomal gain to the MRCA or to an earlier time point is tested by comparing the number of non-amplified and amplified clonal SNVs with a negative binomial distribution (chosen to account for changes in mutation rate along the genome). Hence, our method takes the uncertainty in mutation densities arising from absolute mutation counts and the length of the respective chromosomal gain explicitly into account. This is particularly crucial for small chromosomal gains or small mutation densities, where sampling errors are large. Of note, this is not accounted for when taking the ratio between amplified and non-amplified

SNVs. A subtler difference between our approach and MutationTimeR is that we quantify the peak of non-amplified clonal mutations on its upper half only (e.g., from SNVs with VAFs \geq the tumor purity for a diploid tumor). We chose to implement this in order to avoid overestimation of clonal mutations due to subclonal variants that may be measured at VAFs comparable to clonal VAFs on the lower half of the clonal mutation peak. However, when comparing our timing estimates to those obtained with MutationTimeR, we found good agreement between the two methods: on average, the mutation density at a chromosomal gain relative to the tumor's MRCA overlapped between both methods for 98.5% of the timed chromosomal gains. We show this comparison in the new **Extended Data Fig. 6g**. Moreover, we have now developed an interactive ShinyApp that contains full ECA/MRCA estimation results for each sample on the page "Whole Genome Sequencing", including a chromosome-wise comparison between MutationTimeR and our approach. The tool is available online (<https://kokonech.shinyapps.io/mbOncoAberrations/>). Below we show an example full screenshot from the ShinyApp focusing on the results obtained with our method (**Reviewer Figure 1.4a**) as well as an example comparison of our method with MutationTimeR (**Reviewer Figure 1.4b**).

a

b

Reviewer Figure 1.4. a) Screenshot of the interactive ShinyApp showing WGS data analysis. Top left, meta data (molecular group, group3/4 subgroup, age at diagnosis, gender); top right; allele-specific copy number profile; middle left, the distribution of SNVs explained by particular mutational signatures; middle right, the VAF distribution of SNVs stratified by copy number and the binomial density distribution fitted to the clonal VAF peaks; bottom left and center, densities of non-amplified and amplified clonal SNVs per genomic segment; middle right, densities of amplified clonal SNVs per gained genomic segment along with SNV densities at ECA and MRCA (mean and 95% confidence interval). b) Left, comparison of mutation densities (relative to MRCA) as estimated with our method and with MutationTimeR; right, mutation time relative to MRCA estimated with MutationTimeR (mean and 95% confidence interval).

In a second step, we converted the mutation densities at ECA/MRCA into actual time, based on a population-genetics model of mutation and selection during tumor initiation and growth. The population-genetics model is parametrized with the cohort-wide distribution of mutation densities, the overall incidence of medulloblastoma and the age at tumor diagnosis. The critical rate that determines the conversion between mutation density and real time is the tumor doubling time. To give intuition, our model explains the time span between conception and diagnosis as the sum between tumor initiation and tumor growth. For a given age at diagnosis,

long durations of tumor growth (and hence long doubling times) correspond to an early MRCA and vice versa. To validate the inferred tumor doubling times, we performed transcriptomic analysis for a subset of tumors, where RNAseq was available. In total, we have tumor doubling time estimates for 35 group 3/4 medulloblastomas (for the remaining cases, data quality was insufficient confidently analyze subclonal SNVs with confidence, which form the basis to infer tumor growth dynamics). Of these, we had RNAseq data available for 30 out of 35 cases. Reassuringly, we observed a negative correlation between *KI67* expression and tumor doubling-time at the level of individual cases (**Reviewer Figure 1.5**).

Reviewer Figure 1.5. Tumor doubling time(days) versus *KI67* expression (RPKM). Doubling times were estimated using the population genetics model for 30 Group3/4 medulloblastomas with available RNAseq data. Line and shaded area, loess-regression with 95% confidence interval. Spearman's correlation coefficient is shown in the plot.

However, due to the limited sample size of 30 cases, for which both doubling-time estimates and gene expression data was available, we decided to compare tumors on a group-based level for a more robust analysis. This allowed us to include doubling times from all 35 tumors and gene expression data from 86 tumors, hence strengthening statistical power. At a group level, we found significantly longer doubling times in group 4 as compared to group 3 medulloblastomas (new **Extended Data Fig. 6k**). This finding is consistent with a higher activity of *MYC* target genes and S-phase genes as well as an inferior overall survival in group 3 as compared to group 4 medulloblastoma (Williamson et al., Cell Reports, 2022), which we recapitulated in our data (new **Extended Data Fig. 6i,j**). Thus, patient survival and transcription of cell cycle-related genes agree with our inferred doubling times, which, together with mutation densities at ECA and MRCA, forms the basis for timing the tumors in real-time. We hope that these additional analyses make our timing estimates more convincing.

We added to the text (p. 11):

*Using the bespoke model, we estimated driver mutation rates and associated selective advantages from the clonal SNV densities at ECA and MRCA measured across Group 3/4 medulloblastomas. Simultaneously, we estimated per-tumor doubling times using age and the subclonal variant allele frequencies of 35 tumors with sufficient data quality and information on age at diagnosis (Extended Data Table 5). Consistent with a higher activity of S-phase genes, *MYC* target genes, and poorer overall survival (Extended Data Fig. 4i-k; Williamson et al., 2022), we estimated shorter tumor doubling times in*

medulloblastomas at the Group 3 pole as compared to tumors at the Group 4 pole (Extended Data Fig. 4j), confirming our modeling approach.

Finally, while our model certainly simplifies the true biology, our inferred onset of tumorigenesis during gestation agrees with the detection of unipolar brush cells (UBC) in this time span, which transcriptionally resemble Group 3/4 medulloblastomas in the human brain (Okonechnikov et al., 2023; Smith et al., 2022; Hendrikse et al., 2022). We added limitations of our approach and their contextualization with observations obtained from transcriptional comparison of tumor and normal tissue at the single cell level to the discussion on pg. 17:

This finding is in line with the hypothesis that tetraploidization is a frequent early event in medulloblastoma (Jones et al., Nature, 2012) and is associated with survival rates and high risk of relapse (Mynarek et al., Acta Neuropathologica, 2023). Using mutational clocks and mathematical modeling, we find that medulloblastoma initiation is likely a multi-step process. Specifically, our model suggests that early CNVs acquired during fetal development drive a pre-malignant clone, in which malignant transformation occurs within the first decade of life. While this process is similar to observations in mouse models, where an initial hyperplastic state precedes medulloblastoma growth (Pei et al., Cancer Cell, 2012; Gibson et al., Nature, 2010; Oliver et al., Development, 2005), our modeling approach has some limitations. We here assume that unipolar brush cells (UBCs), the likely cell of origin of Group 3/4 medulloblastomas (Hendrikse et al., Nature, 2022; Smith et al., Nature, 2022), divide at a fairly constant rate and that tumor growth can be approximated with exponential growth. Both assumptions may oversimplify the true biology, which may influence our timing estimates. Reassuringly, however, a disease origin that dates back to fetal development or the first year of life, agrees with the detection of UBCs in human brain samples from this time span (Okonechnikov, Neurooncology, 2023).

- The premise of the ECA analysis seems questionable. First, this assumes that the first driver "hit" coincided with a gain. This would be true if the first hit was a gain or happened at the time of a gain. But not clear this would hold. First, the first hit could be SNV, indel, or loss. Second, gains are likely happening randomly during evolution, some early and some late. The authors do not seem to consider or (more importantly) rule out this possibility. In addition, to time the ECA, they puzzlingly average (early clonal) mutation density across all segments when computing the m_ECA (4). This additionally assumes that all gains happened at the same time.

Reply: We thank the reviewer for their comment on our ECA analysis. We agree that the first driver could be an SNV, indel, or loss; however, we do not detect this within these tumors. Nonetheless, we do not assume that the first driver coincides with a chromosomal gain. The definition of the ECA as an early common ancestor does not imply that this is the earliest common ancestor of the tumor, rather, it is defined as a putatively pre-malignant cell that is ancestral to all tumor cells and that acquired one or several chromosomal gains prior to the tumor's MRCA. We indeed make the null hypothesis that all chromosomal gains occurred in one event. However, we test for each chromosomal gain separately whether the mutation density timing this chromosomal gain conforms to this null hypothesis. Only then, we assign the gain to the ECA. In our data, the null hypothesis was not rejected in any of the Group 3/4 medulloblastomas, and differences in mutation densities between chromosomal gains were too subtle to be informative. Hence, while we cannot rule out that individual gains were acquired rapidly one after another on a time-scale beyond our resolution, our data suggest that all early chromosomal gains were acquired at least within a constrained time window.

That said, the reviewer is correct that we use mutation densities at ECA to calibrate our population genetics model of tumor initiation. While other mutations like SNVs and indels or losses might be the first driver, in our data, only 21 out of 109 Group 3/4 medulloblastomas

show evidence for a clonal SNV/indel in a known driver gene (**Figure 3h**), even after very careful investigation, suggesting that small mutations are most likely not the dominant tumor initiating mode in these tumors.

To make these points clearer, we added text to the following sections:

We added to the Methods section (p. 33):

If a chromosomal gain occurs concomitantly with the onset of tumor growth, the density of amplified clonal mutations (i.e., present on multiple copies of a gained allele) will, on average, be equal to the density of clonal mutations on non-amplified chromosomes or chromosomal regions. By contrast, if a chromosomal gain occurs earlier, the density of amplified clonal mutations on the gained allele will be smaller than the density of clonal mutations on non-amplified chromosomes or chromosomal regions. To distinguish these two cases, MRCA.ECA.quantification() tests for each gained segment whether the density of amplified clonal mutations agrees with the mutation density at MRCA, based on a negative binomial distribution. If the mutation density of amplified clonal mutations is significantly smaller than the mutation density at MRCA, the segment is assigned to an earlier time point and else to the MRCA. MRCA.ECA.quantification() then asks whether segments assigned to earlier time points emerged in the same time window or during different time windows. We define as the null hypothesis that all CNVs emerged in an early common ancestor (ECA). To test the null hypothesis, MRCA.ECA.quantification() computes the mutation densities at the ECA from the number of clonal mutations on the amplified minor allele, $n_{b,l}$ (if $b>1$) and from the number of mutations on the amplified major allele, $n_{CN-b,l}$, as

$$\tilde{m}_{ECA} = \frac{\sum_{l, p_{adj,l} \leq 0.01} n_{b,l} + n_{CN-b,l}}{\sum_{l, p_{adj,l} \leq 0.01} g_{l,b} + g_{l,CN-b}} \quad (4)$$

where $p_{adj,l}$ is the adjusted p value of segment l belonging to the MRCA, and $g_{l,b}$ and $g_{l,CN-b}$ are the length of segment l, contributed by the amplified minor and major allele, respectively. In analogy to \tilde{m}_{MRCA} , MRCA.ECA.quantification() also estimates lower and upper 95%-confidence bounds by bootstrapping. Based on a negative binomial distribution (and an FDR of 0.01), MRCA.ECA.quantification() then tests for each early segment whether its mutation density indeed conforms to a joined ECA, as defined by \tilde{m}_{ECA} . Only segments with mutation densities conforming to \tilde{m}_{ECA} are assigned to the ECA, while all other segments are reported as conforming neither to the ECA nor to the MRCA.

Moreover, we added to the main text (p. 11):

To date these events in actual time, we calibrated a population-genetics model of mutation and selection during tumor initiation (Körber, V. et al., Nature Genetics, 2023) with the measured SNV densities at ECA and MRCA, along with the patient age at diagnosis (see Methods for details). Briefly, the model assumes that medulloblastoma initiation is driven by clonal selection for two consecutive drivers in the transient cell population of (differentiating) unipolar brush progenitor cells from the rhombic lip (Hendrikse et al., Nature, 2022; Smith et al, Nature, 2022) (Extended Data Fig. 6h). Acquisition of the first driver defines a pre-malignant state, arising prior to a tumor's MRCA. While, in principle, the first driver can be any type of mutation, small driver mutations are overall rare in Group 3/4 medulloblastoma (Northcott et al., Nature, 2017), and thus are likely not the major driving force of early tumor evolution. Hence, we focused on cases where early CNVs defined an ECA, and associated the time point at

which the first driver mutation was acquired with the mutation density in the ECA. We assumed that the second driver emerges subsequently in the premalignant clone, spawned by the ECA. Hence, we associated the mutation density in the MRCA with the acquisition of the second driver and the onset of tumor growth. Finally, upon malignant transformation of the tumor's MRCA, we assumed exponential growth to a tumor size of 10^9 cells (corresponding to a few cubic centimeters) at the age of diagnosis (see Methods for details).

Stepping back, it just seems difficult to estimate mutation timing with precision in such a low TMB tumor. It seems that ECA vs non-ECA patterns highlighted in Figure 3 are just noise from an underpowered analysis. Indeed most of the detectable ECA tumors seem to have higher TMB.

Reply: As correctly noted by the reviewer, the uncertainty in timing ECA and MRCA increases as the mutation count decreases. Our approach explicitly models uncertainties in SNV counts based on a negative binomial distribution, hence accounting for an unequal distribution of SNVs along the genome. This is different from alternative timing methods, such as MutationTimeR, which use ratios between non-amplified and amplified SNVs and are hence more sensitive towards uncertainties in mutation densities due to a low tumor mutational burden. Our approach identifies an ECA if and only if there is significant evidence that mutations/chromosomal gains preceded the tumor's MRCA. Of course, we cannot (and we also do not) rule out the existence of an ECA in the remaining cases. To clarify this point, we added an explanation to the main text (page 8). Moreover, we now show inference and uncertainties for all SNV densities at chromosomal gains, ECA and MRCA for each tumor individually in the accompanying ShinyApp (**Reviewer Figure 1.4**). We added to page 10:

The number of such early CNVs varied between 1 and 16 per tumor (mean, 5.3), with no significant difference in SNV density between early CNVs within a tumor (Extended Data Fig. 6f). Hence in tumors with early CNVs, all early CNVs likely arose in an early common ancestor (ECA) during a confined time window prior to the onset of tumor growth. In the remaining cases, where we identified no early CNVs, clonal chromosomal gains likely occurred concomitantly with, or shortly before the onset of tumor growth.

Technical concerns:

- I have major technical concerns about the phylogeny construction, on which some of the paper's main claims (early CNV, late MYC rest). The phylogeny construction is not described in detail in the methods. It also does not seem to be a phylogeny reconstruction per se but a hierarchical clustering of CNV matrices derived from snRNA and snATAC. Key details of this procedure are omitted, including bin sizes for RNA and ATAC, tree cutoffs, and annotations of tree leaves and nodes. Also the phylogenies are presented without showing the supporting RNA / ATAC data either in main figures or the supplement. It is thus hard to interpret or judge the quality of these findings.

Reply: The phylogeny reconstruction was performed via hierarchical clustering via the ward.D2 method. This is a well-established existing technique focused on single-cell CNV profiles (PMIDs: 35681161, 37730813, 37993137). Moreover, in the InferCNV tool, we integrated a random tree subcluster partition in phylogeny reconstruction with minimum p-value < 0.05 to verify the selected cut limit. The bin size for CNV calling has the highest possible resolution and specific for a data type: for snRNA-seq data, it is a single gene, while for snATAC-seq, a single peak. We have updated this information in the Methods on pg. 31.

After CNV calling for each sample separately, the phylogeny clustering results were visualized in UMAPs with k as number of clusters (subclones) in pattern from 2 to 5. Differentially expressed genes for identified subclones per sample were computed via

Wilcoxon Rank Sum test and significant MYC/MYCN/PRDM6 differential expression as well as computed per cell progenitor-like activity enrichment values were used to finalize the derived phylogeny cut limit for each case from manual inspection. The correctness of selected cut limit for the number of subclones in phylogeny was verified based on the usage of the random tree subcluster partition in phylogeny reconstruction with minimum p-value < 0.05.

To enable the precise inspection of CNV hierarchical phylogenetic trees per sample and investigate the properties of subclones, we included the results for each sample into our interactive ShinyApp on the page “Single nuclei RNA/ATAC”. An example of MB272 MYC/MYCN case screenshot of the ShinyApp is shown below (**Reviewer Figure 1.6**). As noted above, the tool is available online through the following link: kokonech.shinyapps.io/mbOncoAberrations.

Reviewer Figure 1.6. Screenshot of the interactive ShinyApp for the presentation of results in the subclonal CNV changes investigation in MB Group 3/4 tumors. Top-left : selected sample and data type as well as sample details (molecular group/subgroup, age, gender, somatic aberration, proportion of subclones, annotation selection for UMAP visualization), . Top-middle: summary of subclones formation. Top-right: CNV bulk profile. Bottom-left: UMAP of the target sample data with selected annotation (e.g. subclone). Bottom-right: CNV profiles with phylogeny and subclones annotation.

- Related to above, how were tumor cells identified in snRNA and snATAC? Text mentions marker genes but this seems insufficient. The standard approach would be to identify cells with copy changes, but this is not described.

Reply: As suggested by the reviewer, we merged the RNA-seq cohort cases and performed CNV analysis on this dataset. As shown in **Reviewer Figure 1.7** below, this analysis verified the annotation of non-tumor cells based on lack of any evident somatic changes within those cells.

Reviewer Figure 1.7. Global CNV profile of snRNA-seq profiles. Top fragment : non-tumor cells.

We included this result as **Extended Data Figure 1f** and noted in the manuscript on pg. 4:

“Non-tumor cell clusters were also verified from copy number variation (CNV) profiling of full combined snRNA-seq dataset (Extended Data Fig. 1f). “

The verification of normal cells in snATAC data was possible based on the inspection of the cells that passed quality control limit for both data types using the CNV status identified from snRNA. We stated this aspect in Methods accordingly on pg. 31.

- Related to above, one explanation for MYC or NMYC heterogeneity could be ecDNA. In other words, MYC could be an initiating event but then its copy number fluctuates across the tumor because of non-Mendelian segregation. This could even lead to subclones where MYC is lost. How do the authors rule this out? Perhaps FISH could inform this.

Reply: Potentially ecDNA could be a critical factor in the formation of the MYC heterogeneity and the presence of such events in medulloblastoma tumors was shown in a recent study (Chapman et al, Nat. Genetics, 2023). We performed MYC/MYCN FISH validations for most of the samples, as noted in Extended Data Table 1. However, standard FISH does not distinguish ecDNA. Other technical methods that can visualize ecDNA by FISH (Hung et al, Nature, 2021) have only been demonstrated in cell lines or PDX samples, and not in primary patient tissue.

Nevertheless, our data do not support a model where MYC is the initiating event but is then lost from the subclones. In particular we typically observe a formation of additional large CNV somatic change specific for a subclone. For example, the MYC subclone in the case MB248 demonstrated not only MYC amplification, but also gain in 7q chromosome fragment despite small amount of subclone cells (**Figure 2c, Extended Data Figure 6e**). Such formation of subclone-specific CNVs was observed in most of the cases. We provided the details about other CNVs formations in MYC/MYCN subclones in the corresponding **Figures 2a,c,e**, but nevertheless noted ecDNA as a potential source of variance in the Discussion on pg 18:

While MYC or MYCN amplifications could arise from the formation of circular extrachromosomal DNA (Chapman et al, Nat. Genetic, 2023), our data show that all subclones with MYC/MYCN amplification demonstrated other large specific CNV gains and losses within the oncogene-amplified subclone, suggesting these clones are not the founder clone.

- Also how were copy number altered samples integrated across snRNA and snATAC. Seurat, to the reviewer's knowledge does not account for aneuploidy. Do snRNA cells that map to snATAC clones have concordant copy profiles with those snATAC clones?

Reply: The 10X Multiomics technology focuses on the extraction of gene expression and signals from the same cells and similar CNV patterns were clearly observed from visual inspection. In order to systematically verify and compare the correctness of single-nuclei CNV profiles obtained from these data types, we compared them to bulk CNV profiles from the same tumor.

For this purpose, we first merged the CNV signals to pseudobulks per sample, both for snRNA and snATAC (**Reviewer Figure 1.8a**). Further, we normalized the mean CNV signals using the same bin resolution (500 Kbp) as available in bulk control methylation CNV profiles (**Reviewer Figure 1.8b**). To measure the correctness of single-nuclei CNV profiles, we computed the correlation between its pseudobulk and bulk CNV profile. Full match of CNV across bins assumes high positive correlation as shown in the example case (**Reviewer Figure 1.8c**). To measure correctness, we compared the resulting correlation values across all cases. We clearly observed that most of the RNA-seq pseudobulk CNV profiles demonstrated best match to the same tumor tissue bulk CNV profile (**Reviewer Figure 1.8d**). However, in 3 out of 16 cases (marked in red in **Reviewer Figure 1.8d**), we found that the highest correlation could be observed to other tumor bulk CNV cases, demonstrating potential limitations of single-nuclei RNA data. Nevertheless, in a similar inspection of snATAC-seq data, we observed the highest correlation to correct control bulk profiles for all samples, underscoring the efficacy of this data type for CNV calling (**Reviewer Figure 1.8e**).

Reviewer Figure 1.8. a) Merged pseudo-bulk CNV profile of snRNA-seq data from sample MB292 b) Methylation bulk data CNV profile from sample MB292. c) Correlation plot of CNV values across 500 Kbp bin between snRNA-seq pseudo-bulk and methylation bulk profiles from sample MB292. d) Cross-comparison of snRNA-seq CNV profiles against bulk profiles. Red boxes indicated 3 cases showing that their highest correlations did not correspond to the same unique sample. e) Cross-comparison of snATAC-seq CNV profiles against bulk profiles.

These materials are included now as **Extended Data Figure 2** and provided in the manuscript on pg. 4:

To verify the single-cell CNV calls, we calculated the correlation between pseudobulk from single-cell and bulk DNA methylation CNV profiles (Extended Data Table 1) across all cases, as shown in the example pair of comparison (Extended Data Fig. 2a-c). A full cohort cross-comparison demonstrated that most of the snRNA-seq pseudobulk CNV profiles matched well with the exception of $n=3$ false positive cases (Extended Data Fig. 2d), while CNV profiles from snATAC-seq data showed the highest correlation to the correct control bulk profile for all samples (Extended Data Fig. 2e), demonstrating the benefit of this data type for CNV calling.

- What is the blue in Figure 2a ? hard to see the copy number variation and what is signal vs. noise, and there is quite a bit of noise, maybe zooming in on the MYCN region would help. The phylogeny should be overlaid.

Reply: We updated the visualization of the corresponding figure by focusing on the phylogeny of a particular sample as well as increasing the image size and including a zoomed inset in updated **Figure 2a** and **Extended Data Fig. 3f**. As noted above, the detailed CNV phylogeny results are now available in an online shiny app for each sample.

- The ECA vs MRCA classification seems like a bespoke version of previous mutation timing approaches. The rationale of mutation timing analyses is that segments gained early in cancer evolution will have a lower density of amplified or early clonal mutations. It is unclear why the authors apply a bespoke approach to this analysis rather than using previous methods (such as Gerstung 2020 PCAWG)

Reply: Our method has similarities to, but is also substantially different from the method by Gerstung et al., 2020. In analogy to Gerstung et al., we use densities of amplified and non-amplified SNVs to time chromosomal gains relative to the most recent common ancestor (MRCA) of the tumor. Crucially, however, the approach by Gerstung et al. uses relative densities, yielding the ratio between amplified and non-amplified SNVs. By contrast, our approach uses absolute SNV counts, and takes into account the sampling error associated with these counts and the length of the chromosome /chromosomal segment affected by the gain. The advantage of our approach is that uncertainties in SNV counts are explicitly modeled, based on a negative binomial distribution that accounts for unequal distribution of SNVs along the genome. Moreover, we obtain absolute estimates of the SNV densities at both ECA and MRCA, allowing for ad-hoc inter-tumor comparison. Finally, we infer actual time at ECA and MRCA using standard population-genetics theory of mutation and selection, which is fundamentally different from the heuristic approach by Gerstung et al., 2020.

Nevertheless, when comparing our timing estimates to those obtained with MutationTimeR, we found good agreement between the two methods as documented above: on average, the mutation density at a chromosomal gain relative to the tumor's MRCA overlapped between both methods for 98.5% of the timed chromosomal gains. We show this comparison in the new **Extended Data Fig. 6g**. Moreover, we show a chromosome-wise comparison for each case in the accompanying shiny app.

We refer to this comparison on page 10:

To corroborate these observations, we contrasted our approach (Körber et al., Nature Genetics, 2023) with an alternative computational tool (Gerstung et al., Nature, 2020), which yielded similar results (Extended Data Fig. 6g).

- The bespoke approach applied by the authors categorizes segments as ECA vs MRCA, and imply that these gains coincide with the timing of a first vs. second driver, respectively. Without hard data to support this (not currently provided) it does not seem that this binary categorization is appropriate. It is more likely that gains occur asynchronously along a time continuum, each gain at a different time point, rather than at discrete moments coinciding with the acquisition of drivers.

If the authors strongly feel that their data points to a different conclusion, it should be rigorously substantiated.

Reply: We thank the reviewer for their comment. We would like to clarify that we have no prior opinion on this. Our method first determines the SNV densities per chromosomal segment and the uncertainty thereof. It then asks whether the number of amplified clonal SNVs on a particular gain is significantly smaller than expected from the SNV density at the tumor's

MRCA. All chromosomal segments with significantly smaller SNV numbers are considered to have arisen prior to the tumor's MRCA. In a second step, we compare the number of amplified clonal SNVs between all early chromosomal gains. If there is no significant difference between these, the gains likely occurred in the same time window and, accordingly, we define the early common ancestor (ECA) as the cell in which these chromosomal gains arose. We note that chromosomal gains with significantly different mutation densities are not assigned to the ECA. However, in our analysis, early chromosomal gains had indistinguishable mutation densities among all analyzed tumors, suggesting that they occurred together or very shortly after each other. To make these points clearer, we now show the number of early CNVs per tumor and the percentage of early CNVs that agree with a single time point in the **Extended Data Fig. 6f**. Moreover, we added to p.10:

The number of such early CNVs varied between 1 and 16 per tumor (mean, 5.3), with no significant difference in SNV density between early CNVs within a tumor (Extended Data Fig. 6f). Hence in tumors with early CNVs, all early CNVs likely arose in an early common ancestor (ECA) during a confined time window prior to the onset of tumor growth. In the remaining cases, where we identified no early CNVs, clonal chromosomal gains likely occurred concomitantly with, or shortly before, the onset of tumor growth.

We also amended the description in Methods to provide better explanation of the molecular timing (p. 33):

“If the mutation density of amplified clonal mutations is significantly smaller than the mutation density at MRCA, the segment is assigned to an earlier time point and else to the MRCA. MRCA.ECA.quantification() then asks whether segments assigned to earlier time points emerged in the same time window or during different time windows. We define as the null hypothesis that all CNVs emerged in an early common ancestor (ECA). To test the null hypothesis, MRCA.ECA.quantification() computes the mutation densities at the ECA from the number of clonal mutations on the amplified minor allele, $n_{b,l}$ (if $b>1$) and from the number of mutations on the amplified major allele, $n_{CN-b,l}$, as

$$\tilde{m}_{ECA} = \frac{\sum_{l, p_{adj,l} \leq 0.01} n_{b,l} + n_{CN-b,l}}{\sum_{l, p_{adj,l} \leq 0.01} g_{l,b} + g_{l,CN-b}} \quad (4)$$

where $p_{adj,l}$ is the adjusted p value of segment l belonging to the MRCA, and $g_{l,b}$ and $g_{l,CN-b}$ are the length of segment l , contributed by the amplified minor and major allele, respectively. In analogy to \tilde{m}_{MRCA} , MRCA.ECA.quantification() also estimates lower and upper 95%-confidence bounds by bootstrapping. Based on a negative binomial distribution (and an FDR of 0.01), MRCA.ECA.quantification() then tests for each early segment with a negative binomial distribution whether its mutation density indeed conforms to a joined ECA, as defined by \tilde{m}_{ECA} . Only segments with mutation densities conforming to \tilde{m}_{ECA} are assigned to the ECA, while all other segments are reported as conforming neither to the ECA nor to the MRCA. “

For example, do the authors assert that specific gains (such as 7 or 11 as suggested by the snRNA data) are the first / early driver? In that case the timing of these gains (not gains overall) should be analyzed. However, it is not clear what rules out chromosomal loss or LOH or a mutation as an early driver.

Reply: This is an excellent point. Indeed, we find that specific gains (in particular 17/17q, 7 or 8) are frequently early events that precede the onset of tumor growth. This information is contained in the upper part of the oncoprint, but we agree that a direct comparison of mutation

densities at individual gains as compared to the mutation densities at the tumor's MRCA provides more detailed insights. To account for this, we now added this information to the new **Extended Data Fig. 7b**. We also agree that we cannot rule out that mutations or LOH preceded these gains. However, in our data, only 21 out of 109 Group 3/4 medulloblastomas show evidence for a clonal SNV/indel in a known driver gene (Figure 3h), suggesting that small mutations are most likely not the dominant tumor initiating mode in these tumors. To explain our analysis better, we added to page 12:

The SNV densities at chromosomal gains were often smaller than the SNV densities at the tumor's MRCA, in particular for gains of Chromosome 4, 7, 12, 17, and LOH of Chromosome 8 and 11 (Extended Data Fig. 7b). While we cannot rule out that small mutations or chromosomal losses preceded these gains, their consistent early timing suggests that chromosomal gains or losses might be among the earliest events during Group 3/4 medulloblastoma initiation.

- It is unclear what is the power for timing copy gains in a tumor type like medulloblastoma with very few somatic SNV. Such an analysis is only really powered for tumor types with high mutation burden, such as adult tumors that evolve across decades. The power to infer timing in a very low-TMB tumor like medulloblastoma seems very low. The authors should apply previously validated approaches for mutation timing and assess the power of these approaches and/or their approach in this low-TMB context.

Reply: As correctly noted by the reviewer, the uncertainty in timing ECA and MRCA increases as the mutation count decreases. Our approach explicitly models uncertainties in SNV counts based on a negative binomial distribution, hence accounting for unequal distribution of SNVs along the genome. This is different from alternative timing methods, such as MutationTimeR, which use ratios between non-amplified and amplified SNVs. Our approach identifies an ECA if and only if there is significant evidence that mutations/chromosomal gains preceded the tumor's MRCA. Of course, we cannot (and we also do not) rule out the existence of an ECA in the remaining cases. To clarify this point, we added an explanation to the main text (on page 10). Moreover, we now show inference and uncertainties for all SNV densities at chromosomal gains, ECA and MRCA for each tumor individually in the accompanying shiny app.

We added to page 10:

The number of such early CNVs varied between 1 and 16 per tumor (mean, 5.3), with no significant difference in SNV density between early CNVs within a tumor (Extended Data Fig. 6f). Hence in tumors with early CNVs, all early CNVs likely arose in an early common ancestor (ECA) during a confined time window prior to the onset of tumor growth. In the remaining cases, where we identified no early CNVs, clonal chromosomal gains likely occurred concomitantly with, or shortly before the onset of tumor growth.

To further complement our initial analysis, we also compared our timing estimates to those obtained with MutationTimeR, and found good agreement between the two methods: on average, the mutation density at a chromosomal gain relative to the tumor's MRCA overlapped between both methods for 98.5% of the timed chromosomal gains. We show this comparison in the new Extended Data Fig. 7g. Moreover, we show a chromosome-wise comparison between our method and MutationTimeR for each case in the accompanying shiny app.

- Related to above the highlighted differences between ECA vs non ECA tumors

- Figure 3k and l analyses don't make sense. Comparing the burden of CNVs to "focal

mutations" seems like comparing apples to oranges - the biological significance is unclear. This analysis would likely give significant values for any tumor type. Also odd to use box plots for a discrete y axis (driver counts).

Reply: In Figure 3k and l, we used a simple statistic regarding the oncoprint to gain insight into the contribution of different mutation types to tumorigenesis in medulloblastoma. However, we agree with the reviewer that different molecular processes shape these mutation types and a direct comparison might be difficult to interpret. We therefore decided to remove these plots from our analysis.

- How do growth estimates from site frequency spectrum analysis correlate with Ki67 or other (eg transcriptional) measures of grade?

Reply: The reviewer raises an excellent point. To validate the tumor doubling times inferred from the site frequency spectrum, we performed transcriptomic analysis for a subset of tumors, where RNAseq was available. In total, we have tumor doubling time estimates for 35 group 3/4 medulloblastomas (for the remaining cases, data quality was not sufficient to confidently analyze subclonal SNVs, which form the data basis to model tumor growth) with sufficient data quality. Of these, we had RNAseq data available for 30 cases. Reassuringly, we observed a negative correlation between Ki67 expression and tumor doubling-time at the level of individual cases (**Reviewer Figure 1.5**). However, due to the limited sample size of 30 cases, for which both doubling-time estimates and gene expression data was available, we decided to compare tumors on a group level for a more robust analysis. This allowed us to include doubling times from all 35 tumors and gene expression data from 86 tumors, hence improving statistical power. At the group level, we found significantly longer doubling times in group 4 as compared to group 3 medulloblastomas (new **Extended Data Fig. 6k**). This finding is consistent with a higher activity of *MYC* target genes, S-phase genes and a poorer overall survival in group 3 as compared to group 4 medulloblastoma (Williamson et al., Cell Reports, 2022), which we recapitulated in our data (new **Extended Data Fig. 6i,j**). Thus, patient survival and transcription of cell cycle-related genes agree with our inferred doubling time. We have added these additional analyses to Extended Data Fig. 6 and added to the text (p. 11):

Using the bespoke model, we estimated driver mutation rates and associated selective advantages from the clonal SNV densities at ECA and MRCA measured across Group 3/4 medulloblastomas. Simultaneously, we estimated per-tumor doubling times using age and the subclonal variant allele frequencies of 35 tumors with sufficient data quality and information on age at diagnosis (Extended Data Table 5). Consistent with a higher activity of cell cycle genes, MYC target genes, and poorer overall survival (Extended Data Fig. 6i-k; Williamson et al., 2022), we estimated shorter tumor doubling times in medulloblastomas at the Group 3 pole as compared to tumors at the Group 4 pole (Extended Data Fig. 6j), confirming our modeling approach.

- Text and captions does not clearly describe the data, how it was generated or analyzed. For example, it is unclear what is the source of phylogenies in Figs 1,2, and 5, presumably snATAC but unclear?

Reply: We updated all the figure labels and legends to better describe the data. We have now included the corresponding phylogenetic trees for the corresponding CNV plots in the Figures 2 and 5 as well as for each sample in the ShinyApp as noted above.

- According to the methods, a segment is assigned to ECA if the density of amplified clonal mutations is significantly less than the m_MRCA. Before getting into details of implementation, this does not make sense - the density of amplified mutations will always be

less than the density of total mutations. Amplified clonal mutations are a subset of total clonal mutations.

Reply: Here, we respectfully disagree with the reviewer. Mutational densities measure the number of SNVs per bp. If tumor growth commences together with a chromosomal gain, the mutation densities of amplified SNVs on this chromosomal arm will be indistinguishable from the mutation density of non-amplified clonal SNVs. Only if tumor growth commences significantly later, there will be a difference in densities of amplified and non-amplified clonal SNVs. To clarify these points, we added to Methods (p. 33):

If a chromosomal gain occurs concomitantly with the onset of tumor growth, the density of amplified clonal mutations on the gained allele will, on average, be equal to the density of clonal mutations on non-amplified regions in the genome. By contrast, if a chromosomal gain occurs earlier, the density of amplified clonal mutations on the gained allele will be smaller than the density of clonal mutations on non-amplified regions in the genome. To distinguish these two cases, MRCA.ECA.quantification() tests for each gained segment whether the density of amplified clonal mutations agrees with the mutation density at MRCA based on a negative binomial distribution. If the mutation density of amplified clonal mutations is significantly smaller than the mutation density at MRCA, the segment is assigned to an earlier time point and else to the MRCA.

- The methods are unclear which is partly due to imprecise or poorly defined jargon and/or mathematical notation and left out details. For example, " the number of B alleles on this segment (line 782). A B-allele is generally non-reference allele. The authors likely mean the germline homolog with the smaller copy number, often called the minor allele. However, whatever term the authors use, it should be defined, especially if it is non-standard usage. The reader has to also guess what is gl,b and $gl,CN-b$ (equation 4). There are many other such examples.

Reply: We apologize for lack of clarity in the methods section. We have now changed wording from B alleles to "minor allele" and explained its definition in the text. Moreover, we have checked that everything is defined. We hope that the methods have now become clear.

Referee #1 (Remarks on code availability):

The code is no runnable, partly because it because it points to files paths on a private file system.

for example:

/b06x-isilon/b06x-m/mbCSF/results/humanTumor/mbSpatial/perSampleV2

Reply: We have updated the source code and created a tutorial with a sample dataset to allow re-analyses from snRNA and snATAC-seq data, available on the following link: <https://github.com/kokonech/mbOncoAberrations/wiki>.

Information on how to re-run WGS-based analyses are given in the README on <https://github.com/kokonech/mbOncoAberrations/>.

Variant calls used as input will be made publicly available on Mendeley Data upon publication, but can be accessed by the reviewers via:

<https://data.mendeley.com/preview/zrrrs3c6t6?a=c5885153-b375-4268-bebe-337803985b4d>.

Referee #2 (Remarks to the Author):

Okonechnikov and colleagues report an interesting and important study of Group 3 and 4 medulloblastoma. These enigmatic tumors include the most aggressive subtypes of the disease. As indicated by the authors, we are beginning to understand the origins of these tumors (that to date have been less clear than other disease subtypes) but we know little about how the molecular genetic alterations in these tumors drive the pathological phenotype. The authors used single-cell technologies including snRNAseq, snATAC-seq and spatial transcriptomics to dissect cell populations among medulloblastomas with known alterations in established medulloblastoma oncogenes including MYC, MYCN, and PRDM6. There are a number of important and novel findings from this study that will be of broad interest. First, the study shows that large-scale chromosomal aberrations are early tumor initiating events, while single-gene oncogenic events are late events and typically sub-clonal. This is important because single gene events are often considered as therapeutic targets, but this may not be the case in these tumors. Understanding the timing of these alterations is key to distinguishing truncal and subclonal events. The authors show also that MYC alterations are clonal in certain instances of disease progression and may drive tumor development and treatment resistance. A particularly attractive aspect of the study is the use of spatial transcriptomics to define the topographical distribution of subclones that reveals clear segregation of the cells. Finally, by integrating the genetic findings with a population genetics model, the authors estimate that medulloblastoma initiation in cerebellar unipolar brush cell-lineages starts from the first gestational trimester. This report is likely to be of broad interest to the community and the data in general support the conclusions drawn. There are some questions that, if addressed, might further strengthen the manuscript.

We thank the reviewer for their overall very positive feedback and very useful suggestions to further improve our manuscript.

- The authors discuss (and present in Fig. 2) the differentiation, proliferation and progenitor-like activity states of cells using snDNAseq. These data are interesting, but the presentation is confusing. What precisely are the authors trying to say here? Since proliferation and differentiation subclones are seen in both the MYCN amplified and non-amplified subclones, then what is the relevance of MYCN to this phenotype? These clearly are different compartments, but what are these differences? No real discussion of this seems to be presented. There may be a decrease in the MYCN expression within the differentiation compartment but, in the context of a large set of genes that will be changing, what does this mean? Since the non-MYCN amplified subclone also displays differentiation, then what is the relevance of MYCN? Can this be better described and related to other changes in the transcriptomes.

Reply: We are thankful to the reviewer to raise this critical point. Our aim to describe proliferative and differentiating compartments within a tumor was to identify association of these characteristics with the respective driver oncogene. We also addressed the long-standing question: do MYC/MYCN-amplified tumors differentiate? We indeed identified differentiation within the MYC/MYCN compartment that was anti-correlated to oncogene expression. Moreover, we observed the enrichment of progenitor-like activity within MYC/MYCN subclones. We now note this more clearly in the manuscript to resolve any confusion:

As cells differentiated, expression of the oncogene itself showed lower expression within the MYCN-amplified subclone (Pearson cor. = -0.23, $p < 2.2e-16$, Extended Data Fig. 2c), demonstrating that MYCN has a connection to an undifferentiated state (He et al, Front Oncol., 2013).

Based on reviewer's suggestion, we have now also identified differentially expressed gene signatures that drive the identity of the *MYC/MYCN* compartments as well as oncogene-negative subclones. In particular, we systematically identified subclone-specific differentially expressed genes (DEGs) that are supported at least by n=2 cases (**Extended Data Table 2**). We found a clear enrichment of neuroinflammation activity in *MYCN*-subclones, while rRNA and ribosome biogenesis were strongly enriched in *MYC*-specific subclones, known to be associated with these somatic factors. *PRDM6* subclones demonstrated association with forebrain development. Interestingly, non-*MYCN* and non-*MYC* subclones were similar in terms of group specific DEGs (overlap ~36%). In general, non-*MYCN*, non-*MYC* and non-*PRDM6* subclone DEGs showed enrichment in neuronal-system development. We further checked more precisely the enrichment of cerebellum normal cell types within subclones without *MYC/MYCN/PRDM6* activation. We found that these subclones are clearly enriched in the development of unipolar brush cells, the proposed cell-of-origin for Group 3/4 tumors ($p < 1e-06$), while in *MYC/MYC* subclones this property was not observed ($p > 0.05$). We noted this aspect in the manuscript on pg. 5:

By inspecting differentially expressed genes specific for each subclone (Extended Data Table 2) we also identified a unique property of non-MYCN subclones with stronger enrichment of markers of unipolar brush cells ($p < 1.14e-06$), the cell-of-origin of Group 3/4 medulloblastoma (Hendrikse et al., Nature, 2022; Smith et al, Nature, 2022), while MYCN subclone-associated genes did not show this property ($p > 0.05$). This observation was also confirmed using subclone-specific genes associated with cis-regulatory elements derived from the integration of snATAC-seq data (Extended Data Table 3).

The authors use UMAPs for the various proliferation and differentiation states of cells but then (e.g., in Fig.2e) individual scale-dot heat graphs. Relating these two is not straightforward. It is worth noting that given the lack of apparent variance in the expression scores of C1 and C2 in the UMAP, these are not that informative.

Reply: To avoid confusions in the interpretation, we redesigned the figures and removed the differentiation signal from the main UMAPs. Instead, we now provide them as additional plots in the supplementary figures. Please see **Reviewer Figure 2.1a-c** below as an example for a *MYCN*-amplified case. Importantly, we also created an interactive ShinyApp web application that provides all results on a per sample basis and includes colored cell annotations as shown in the exemplary screenshot (**Reviewer Figure 2.1d**). The ShinyApp is available at kokonech.shinyapps.io/mbOncoAberrations.

Reviewer Figure 2.1. a) *snRNA-seq* UMAP of single MYCN sample MB183. Gray boxes, proliferating cells. d) MYCN expression in C1 and C2 clones. b,c) Per cell GSVA enrichment of proliferation (b) and differentiation (c) markers within UMAP of MB183 MYCN-amplified sample. d) Screenshot of the interactive ShinyApp for the presentation of results in the subclonal and global CNV changes investigation in MB Group 3/4 tumors.

The finding that large clonal CNVs are likely founding events is also interesting. But little discussion is made of what genes might be the focus of such events. If these large CNVs are driver events, then what genes are upregulated in their expression as a consequence? and how, if at all, do these genes relate to the progenitor, differentiation and other signatures studied.

Reply: This is indeed an important aspect of the manuscript. To further investigate the corresponding common putative driver CNV regions, we aimed to identify genes specific for medulloblastoma Group 3/4 located in the regions of interest compared with other central nervous system tumors, using our large bulk gene expression dataset, encompassing 43 central nervous system tumor classes in 2482 samples (Okonechnikov et al, Neurooncology, 2023). Differential expression analysis identified 394 genes overexpressed and located within commonly gain regions of Group 3/4 as well as 261 genes inactive and located in commonly deleted regions specific for Group 3/4 accordingly. Interestingly, the gained genes demonstrated enrichment in differentiation (p-val: 0.0018), thus supporting our observation independent of subclone type. Also, most of these genes overlap with other existing known somatic drivers e.g. derived from the COSMIC database. We included the list of identified genes as an additional **Extended Data Table 6b**.

While the authors note MYC-amplified clones demonstrated strong enrichment of progenitor-like activity compared to non-MYC-amplified compartments and that differentially expressed genes specific to MYC-amplified subclones were enriched in known MYC target genes, it would be helpful to know if these are topographically enriched (or not) on chromosomal locations subject to CNV.

Reply: We inspected whether differentially expressed genes specific for MYC-amplified subclones are enriched in any chromosomal locations via chi-squared test. For this purpose, we used as a reference the gene lists annotations from the chromosome and karyotype band tracks provided by MSiGDB (Subramanian et al, Proc Natl Acad Sci, 2005) . However, in none out of 6 MYC cases any statistically significant enrichment was observed in a CNV location specific for a MYC subclone. We noted this in the manuscript accordingly on pg. 7:

The MYC subclone-specific genes were not found enriched in any corresponding subclone-specific CNVs.

The data presented in Fig.3 and ext data Fig.4 are very interesting. To date, our understanding of the origins of medulloblastoma have largely come from extrapolating observations in mouse models. Therefore, timing disease onset in humans is important. As a side note, the authors may want to use nomenclature other than subgroup I-VIII in Fig.3h given that subgroup is also applied to Group 3 and 4 medulloblastoma. The authors note that they make the assumption that upon malignant transformation the tumor adopts exponential growth until age of diagnosis. This is not unreasonable, but it is also not inline with what is observed in mouse brain tumour models. In a number of these models, pinpointing transformation is difficult. Susceptible tissues initially undergo hyperplastic expansion and may remain in this state until progression to malignant state occurs. Some recognition of this and how it may affect assumptions is worth consideration. Does the overlap in timing of origin support a common lineage for Group 3 and 4 tumors?

Reply: We thank the reviewer for their constructive input on our modeling approach. In our model, we assume that malignant transformation occurs in two steps, where one (or several) initial mutations drive a pre-malignant lesion. The pre-malignant clone transforms upon acquiring additional driver mutations, spawning the malignant clone that eventually constitutes the tumor. In our model, we assume exponential expansion of the final tumor mass only after malignant transformation has occurred, i.e., when both driver mutations have been acquired. In that regard, we attempt to recapitulate tumor initiation in two stages, where an initial hyperplastic phase precedes exponential expansion of the tumor. We agree, however, that our modeling approach likely oversimplifies the true biological situation.

To explain our modeling approach better, we have now added to pg 11:

Acquisition of the first driver defines a pre-malignant state, arising prior to a tumor's MRCA. While, in principle, the first driver can be any type of mutation, small driver mutations are overall rare in Group 3/4 medulloblastoma (Northcott et al., Nature, 2017), and thus are likely not the major driving force of early tumor evolution. Hence, we focused on cases where early CNVs defined an ECA, and associated the time point at which the first driver mutation was acquired with the mutation density in the ECA. We assumed that the first driver mutation spawns a pre-malignant clone, in which the second driver mutation is acquired. Hence, we associated the mutation density in the MRCA with the acquisition of the second driver and the onset of tumor growth. Finally, upon malignant transformation of the tumor's MRCA, we assumed exponential growth to a tumor size of 10^9 cells (corresponding to a few cubic centimeters) at the age of diagnosis (see Methods for details).

Moreover, to put our results in context with observations from mouse models, we added to the Discussion (pg. 17):

This finding is in line with the hypothesis that tetraploidization is a frequent early event in medulloblastoma and is associated with survival rates and high risk of relapse. Using mutational clocks and mathematical modeling, we find that medulloblastoma initiation is likely a multi-step process. Specifically, our model suggests that early CNVs, acquired in fetal development, drive a pre-malignant clone, in which malignant transformation occurs within the first decade of life. While this process is similar to observations in mouse models, where an initial hyperplastic state precedes medulloblastoma growth (Pei et al., Cancer Cell, 2012; Gibson et al., Nature, 2010; Oliver et al., Development, 2005), our modeling approach has some limitations. We here assume that UBCs, the likely cell of origin of Group 3/4 medulloblastomas (Hendrikse et al., Nature, 2022; Smith et al, Nature, 2022), divide at a fairly constant rate and that tumor growth can be approximated with exponential growth. Both assumptions may oversimplify the true biology, which may influence our timing estimates. Reassuringly, however, a disease origin that dates back to fetal development or the first year of life, agrees with the detection of UBC progenitors in human brain samples from this time span (Okonechnikov, Neurooncology, 2023).

The reviewer also asks whether the overlap in timing of origin supports a common lineage for Group 3 and Group 4 tumors. This question is difficult to address from genome data alone. We do infer a similar time span for tumor initiation in both groups (Fig. 3b), but cannot rule out that initiation occurs in different lineages.

Finally, we would clarify that the subgroup in Fig. 3h do indeed correspond to the Group 3/4 medulloblastoma subtypes.

Similar to my comment above, the authors classified gains of Chromosome 1q, 4, 7, 12, 17/17q and 18, as well as LOH on 5q, 8, 10/10q, 11 and 17p as putative drivers of Group 3 and 4 medulloblastoma initiation. However, no discussion is made of what genes on these CNVs might be clonally selecting this driver event. This is an important issue that warrants some discussion.

Reply: As noted above, we prepared a more detailed supplementary table listing the corresponding genes associated with common CNVs and investigated their properties. For this purpose, we focused our analysis on gain or loss of chromosomal segments associated with chromosomes 1, 7, 10 and 17, which are among the most frequent CNV events in Group 3/4 MB (Northcott et al, Nature, 2017). Using bulk transcriptomic data from 405 Group 3/4 patients with available CNV metadata (Cavalli et al, Cancer Cell, 2017), we identified genes up (or down) regulated upon gain (or loss) of a chromosomal segment. We identified around ~10% of the differentially up (or down) regulated genes, which were located on the gained (or lost) chromosomal segment. For example, comparing 57 patients with chr1q gain (while chr1p being neutral) and 310 control cases (both chr1p and 1q neutral), we identified that 142 genes out of 988 upregulated genes are located on chr1q. Among these upregulated genes, we identified *MUC1*, *PBX1* and *SLC45A3* as genes associated with cancer based on COSMIC database.

All details are included in the **Extended Data Table 6a** accordingly, while both tables focusing on the common gains and losses are noted in the manuscript on pg. 12:

To search for putative drivers located on these regions, we analyzed the expression of genes lying in commonly gained or lost regions. For this purpose we contrasted gene expression between tumors with and without particular CNVs in a bulk RNA-seq cohort of Group 3/4 medulloblastoma (Cavalli et al, Cancer Cell, 2017; Extended Data Table 6a) and inspected differentially expressed genes specific for Group 3/4 medulloblastoma

in global CNS tumor cohort (Okonechnikov et al, Neurooncology, 2023; Extended Data Table 6b). Among these identified genes, up to 10% have been described as known somatic drivers, thus associated with medulloblastoma evolution.

The spatial data presented in the manuscript are also very interesting; but other than a brief mention of proliferation e.g., Ki67 or differentiation there is a lack of clarity around what genes were selected for study and precisely why. The patterns of clones are also interesting, but the authors chose to restrict this analysis simply a description of clonal compartmentalisation. There are other important issues at play here. For example, how do proliferation and differentiation compartments relate to other microanatomical structure such as vessels that might provide a niche?

Reply: The initial list of the genes chosen was based on predefined properties: tumor-specific genes (60%) and normal cell markers (40%). The tumor-specific genes were selected based on (i) known target markers of Group 3/4 tumors including *MYC*, *MYCN*, *SNCAIP* and *PRDM6*; (ii) proliferation, differentiation and cell cycle activity markers; and (iii) cell-of-origin-associated markers. The normal cell type markers were selected based on knowledge about normal cell types of cerebellum that could also be present in the tumor, such as astroglial, meningeal, immune, etc. For each cell type, at least 2 markers were selected. All the genes that we selected were also verified on available single cell RNA-data as well as bulk profiles. We provide these details in **Extended Data Table 7** listing the genes as well as describe them in the Methods section.

Importantly, the spatial protocol that we applied also has a technical limitation: the size of the tissue fragments (maximum square fragment length 2 mm) might obscure the presence of blood vessels. Nevertheless, we investigated this possibility based on a more precise annotation of normal cell types. In particular, we distinguished astroglial, meningeal and immune cell types in the spatial data. Potentially cells that we refer to as “meningeal”, also in association to other cell types such as endothelial and mural, could reflect the formation of vessels. We used *IGFBP7* (visualized per sample in Supplementary Figure 5) and *COL1A2* (shown in Reviewer Figure 2.2a,e) as the main marker genes. In some large image fragments, these markers reflected the possible formation of vessels, such as the *MYC*-amplified case MB292. After a detailed annotation of normal cell types, we observed an enrichment of meningeal cells in specific spatial regions (**Reviewer Figure 2.2a-c**). We further performed measurements of contact strength between proliferating/differentiating subclones and non-tumor cell types for this *MYC*-amplified case and observed that the meningeal cells remain mostly isolated from the tumor cells, but remain in contact with immune cells (**Reviewer Figure 2.2d**). This same pattern was observed in another *MYCN* case with an extensive amount of non-tumor cells containing potential vessels (**Reviewer Figure 2.2e-h**).

Reviewer Figure 2.2. a) COL1A2 spatial expression of sample MB292. b) Spatial visualization of subclones and non-tumor cell types of sample MB292 c) Spatial data UMAP of sample MB292. d) Proximity of each compartment to each other of sample in MB292. e) COL1A2 spatial expression of sample MB165. f) Spatial visualization of subclones and non-tumor cell types of sample MB165 g) Spatial data UMAP of sample MB165. h) Proximity of each compartment to each other of sample in MB165.

However, due to the noted technical limitation, we observed the presence of such vessels only in a small subset of samples in our cohort. We have now stated the limitations of the technique and suggested potential future investigations in the Discussion accordingly, on pg. 18:

Single-cell spatial data from Group 3/4 tumors allowed us to inspect the composition of the subclones across tumor tissue fragments; however, due to the fixed number of genes ($n=100$) and limited image size (max 2 mm), some other spatial properties, such as formation of blood vessels, were not covered in our study. The application of novel spatial techniques that overcome such limitations will be an important research direction in the future.

Referee #3 (Remarks to the Author):

Okonechnikov, Joshi, Körber and colleagues present a manuscript focused on oncogene-amplified Group 3/4 medulloblastoma. Medulloblastoma is one of the more common types of childhood brain tumors, which are overall rare but pose significant clinical challenges. About one third of children with medulloblastoma will die from their disease. The authors present results on three sets of experiments: First, they describes the use of single cell RNA and ATAC-seq datasets of a relatively large cohort of 20 patient tumor samples. The authors infer copy number variations to define genetic subclones and putative phylogenetic trees. Amplifications of medulloblastoma oncogenes MYC, MYCN, and PRDM6 are often subclonal and occur late in tumor evolution. The authors conclude that oncogene amplifications drive disease progression and therapy resistance, but are likely not involved in the early steps of tumorigenesis. The second part describes the use of deep whole genome sequencing data from 183 medulloblastoma samples to utilize somatic variants as a molecular clock to define the time of onset for different medulloblastoma subgroups. The authors find that the earliest common ancestor of individual tumors likely formed between the first trimester of pregnancy and the first year of life. Finally, the authors apply spatial transcriptomics to a cohort of 17 patient samples to map the spatial relationship of molecular subclones defined in the first set of experiments and describe two major localization patterns in which subclones are either interspersed or segregated. Finally, rare mosaic medulloblastoma tumors that show amplifications of both MYC and MYCN in separate subclones are reported.

Overall, I find this study to be interesting as it offers a fresh look on medulloblastoma genome evolution from early initiating events to tumor relapse. The study is entirely based on data generated from patient samples, which in itself might be a valuable resource to the scientific community. The most intriguing part of the study are the molecular clock analyses which define a putative time of tumor onset. This is a clever approach that uses the density of somatic passenger mutations in genomic regions that subsequently undergo broad copy-number changes (which define the first oncogenic event/early common ancestor). The authors recently published this approach in a study focused on neuroblastoma (Körber et al, Nature Genetics 2023). The developmental origins of Group 3/4 medulloblastoma have recently been identified by correlative analyses to cell types in the developing human cerebellum, and an estimation of the time of tumor onset adds an important layer of information. That said, I have some concerns about the quality of some of the other analyses and the lack of technical validation as outlined in the specific comments below. The study is of interest to the field of medulloblastoma research and perhaps other pediatric cancers, but offers few insights for researchers outside this area of research.

We thank the reviewer for their important feedback and overall very positive evaluation of our manuscript.

1 - Single cell-derived CNV profiles appear very noisy and it is difficult to visually discern the variations that are being described. Very little information is provided in the methods on how inference of CNVs is performed. It is stated that five single cells are combined into a meta-cell based on their similarity (in the transcriptome space?) prior to inferCNV analysis to “improve the specificity”. This approach is unorthodox and requires clarification and critical assessment.

Reply: We are thankful to the reviewer for this critical note. We have now provided a detailed explanation for the CNV analysis using snRNA-seq and snATAC-seq data in the Methods. In particular, we now clearly state the adjusted settings of InferCNV method for the phylogeny reconstruction using hierarchical clustering, namely, the ward.D2 method with predefined cut-off limits for the number of clusters in order to select MYC/MYCN/PRDM6 subclones. We also describe this phylogeny reconstruction verification further in detail.

The metacells method was applied as described and tested in a previous study (Blanco-Carmona et al, Cell Reports Medicine, 2023); the relevant reference has been now included. To generate the meta-cells: first, the data is clustered using an unsupervised method as implemented in Seurat. Then from each cluster, n=5 cells are randomly selected to merge and form the meta-cell. The meta-cell size is based on the suggested amount of 10X protocol noted in the study above. The Meta-cell approach was used to average out the noise associated with single-cell data, especially the snATAC-seq data as the reviewer has correctly pointed out. To highlight the improvement in visualization using the meta-cell approach, a comparative example with and without the meta-cell approach is shown below. (**Reviewer Figure 3.1a-b**).

Reviewer Figure 3.1. a-n) Single nuclei CNV profiles from snRNA-seq data of MB183 sample using standard cell counts as input (a) and meta-cells derived (b) as input.

Importantly, the main analysis workflows including CNV calling are also provided now in our shared open source code repository with a tutorial describing all steps including meta-cells applied on a specific sample (<https://github.com/kokonech/mbOncoAberrations/wiki>).

2 - Bulk copy-number profiles (WGS or methylation array-based) should be placed adjacent to the single cell-derived profiles to allow for a better interpretation of potential artifacts of the approach. Are the same patients included in Figure 3h? Do clonal and subclonal CNVs match between the WGS and single cell datasets? Also it is generally difficult to appreciate the amplification status of said oncogenes. Other means of illustration should be explored. Limitations of the approach (inferring CNVs from RNA expression or open chromatin) also need to be more clearly discussed.

Reply: We fully agree with the reviewer that it is critical to demonstrate bulk CNV profiles for the verification of results obtained from single-cell multiomics data. We had initially performed manual inspection per sample by comparing the CNV profile obtained from snRNA-seq/snATAC-seq to that obtained from bulk WGS/methylation array data. We have now shared these comparative analyses in our accompanying interactive ShinyApp, which includes bulk methylation CNV profiles, UMAPs with detailed annotation and full single-cell CNV profiles obtained from both methods for each sample on the page “Single Nuclei RNA/ATAC”. An example of this visualization is shown in the **Reviewer Figure 3.2** below. The application is available via corresponding link: kokonech.shinyapps.io/mbOncoAberrations.

In **Figure 3h**, we provide an additional annotation stating which samples are present within the single-cell dataset accordingly. All the details for bulk WGS analysis including WGS CNV profiles are also available in the corresponding ShinyApp on the page “Whole Genome Sequencing”.

Reviewer Figure 3.2. a) Screenshot of the interactive ShinyApp for the presentation of results in the subclonal and global CNV changes investigation in MB Group 3/4 tumors.

As the reviewer suggested, we have now also included analyses to obtain a metric for comparing CNV profiles obtained from bulk versus single-cell multiomics data. In this analysis, we first merged the CNV profile obtained from single-cell multiomics data into a pseudobulk profile per sample (**Reviewer Figure 3.3a**). Further, we normalized the mean CNV signals for this pseudobulk profile to the same bin resolution (500 Kbp) as available in the bulk control DNA methylation CNV profiles (**Reviewer Figure 3.3b**). To measure the similarity between the two profiles, pseudobulk and control reference, we computed the correlation between these profiles using bin for all the chromosomes except X, Y and MT. Full match of CNV across bins assumes high positive correlation as shown in the example visualization (**Reviewer Figure 3.3c**). To measure correctness, we performed full cross-comparison for all cases. We clearly observed that most of the RNA-seq pseudobulk CNV profiles demonstrated best match to the same tumor tissue bulk CNV profile (**Reviewer Figure 3.3d**). However, in 3 out of 16 cases (highlighted in red in Figure) we found that highest correlation could be observed to other tumor sample bulk CNV profiles, demonstrating potential limitations of single nuclei RNA data, as the reviewer correctly pointed out. However, using a similar approach for snATAC-seq based CNV profile, we observed the highest correlation to correct control bulk profiles for all samples, demonstrating the benefit of using snATAC-seq based CNV calling (**Reviewer Figure 3.3e**). These materials are now included as Extended Data figures into the manuscript (**Extended Data Figure 2**) and noted accordingly in the text on pg. 4:

*To verify the single cell CNV calls, we calculated the correlation between pseudobulk from single-cell and bulk DNA methylation CNV profiles (**Extended Data Table 1**) across all cases, as shown in **Extended Data Fig. 2a-c**. A full cohort cross-comparison demonstrated that most of the snRNA-seq pseudobulk CNV profiles matched well with the exception of $n=3$ false positive cases (**Extended Data Fig. 2d**), while CNV profiles from snATAC-seq data showed the highest correlation to the correct control bulk profile for all samples (**Extended Data Fig. 2e**), demonstrating the benefit of using this data type for CNV calling.*

Reviewer Figure 3.3. a) Merged pseudo-bulk CNV profile of snRNA-seq data from sample MB292 b) Methylation bulk data CNV profile from sample MB292. c) Correlation plot of CNV values across 500 Kbp bin between snRNA-seq pseudobulk and methylation bulk profiles from sample MB292. d) Cross-comparison of snRNA-seq CNV profiles against bulk profiles. In red marked 3 cases where highest correlation does not correspond to the same sample. e) Cross-comparison of snATAC-seq CNV profiles against bulk profiles.

Additionally we now integrated an orthogonal experimental method for verification of subclones: parallel RNA and DNA profiling from ResolveOme protocol applied on 4 MYC- and 2 MYCN-amplified cases. The CNV profiling of DNA data from this method clearly verified the presence of CNV structure that we detected from snRNA/ATAC-seq data. This result is shown in **Extended Data Figure 5** and noted in the manuscript on the page 7:

“To further verify the presence of subclones, we performed single-cell whole-genome DNA-sequencing and RNA-sequencing from the same cells, in a subset of MYCN- and MYC-amplified cases (Extended Data Figure 5a). In all tested samples, the presence of the corresponding specific subclones, and their CNV profile matched between the snMultiomic data based on single-cell RNA projection (Extended Data Fig. 5b) and WGS CNV analysis (Extended Data Fig. 5c-e).”

We also included an additional visualization to demonstrate *MYC/MYCN* amplifications for our example target cases. For this purpose, we magnified the target region (chr 2 for *MYCN* and chr8 for *MYC*) and compared the signal between subclones in the **Figure 2a** and **Extended Data Figure 3c**.

Finally, to account for uncertainty with respect to causality in integration of bulk CNV profiling for tumor evolution investigation, we have changed the title of the previous section “Large-scale copy number variation drivers early tumor growth” to “Large-scale copy number variation acquired early during tumor evolution”.

3 - It is unclear how phylogenetic trees were derived (the methods mention hierarchical clustering). More details need to be provided, and examples need to be shown. The presented results look sometimes arbitrary. For example, the first tumor presented in Figure 2a (MB183) has almost equal fractions of C1 and C2. Figure 1b shows 27% for C1. In the second tumor (MB177), I do not see on which basis C1 and C2 were separated, there seems to be a more obvious split. How robust and generalizable is the approach? How large (frequency/number of cells) does a subclone need to be in order to be detected? Are the same subclones detected if the data is randomly split into subsets? What is the minimum size of a CNV in order to be confidently distinguished between subclones?

Reply: We now provide detailed information on the CNV calling and associated integrated results for each sample in the interactive ShinyApp. Because differences could be present between snRNA and snATAC in the subclone proportion due to technical variance in the amount of cells satisfying quality control limits, we carefully inspected all the visualizations and updated corresponding proportions for the target subclones. These updated values are provided in the ShinyApp for each data type separately, while in the manuscript **Figures 2b,f,j**, we provide the mean proportion between snRNA-seq and snATAC-seq results. As an example, we included precise CNV visualization of sample MB183 snRNA (**Extended Data Figure 3a**) and snATAC (**Figure 2a**) data in the manuscript.

For the selection of subclones, we performed manual tree cut limit k selection, based on the inspection of *MYC/MYCN* locations, as well as verification in the UMAP based on the obtained gene expression/epigenetic signal for *MYC*, *MYCN* and *PRDM6*. We did not use any strict limitations for the size of the CNV to distinguish subclones. However, to verify that our selected cluster formation is valid, we also inspected whether selected k passes filtering control after performing the random tree subcluster partition in phylogeny reconstruction within InferCNV toolkit with minimum p-value 0.05 cut limit. These details are now provided in the Methods at pg. 31 as well.

Moreover, as noted above, in 6 control cases the subclones identified from snRNA/ATAC data types were fully verified from the DNA/RNA ResolveOme data. These results are shown in the **Extended Data Figure 5**.

4 - The UMAP clusterings provide strong evidence for the existence of major subclones in the presented samples. Based on these clusterings it appears that the number of subclones might be even higher than estimated based on inferred CNVs (e.g., in Figure 2g/k)? Comparing the differentiation state to tumor subclones is interesting, however a different scaling of values/colors would be appropriate.

Reply: Indeed, in some cases we could observe a larger number of subclones in comparison to our selection. A clear example is the *MYC/MYCN* MB272 case, where the *MYC* subclone demonstrates a split into two different subclones with amplifications in chromosomes 5 and 6 (as shown in **Reviewer Figure 3.2**). However, as noted above, we selected our cut-off limit

for the subclones in phylogenetic trees based on oncogene amplification *MYC/MYCN/PRDM6*.

To avoid mixed visualization, we removed the usage of color of differentiation in the corresponding main figures (**Fig. 2d / Reviewer Figure 3.4a**) but also provided additional UMAP figures as supplementary materials where proliferation and differentiation signal is demonstrated (**Extended Data Figure 3b-c, Reviewer Figure 3.4b-c**). Importantly, this visualization is available for each sample in our interactive ShinyApp on the page “Single nuclei RNA/ATAC” .

Reviewer Figure 3.5 a) *snRNA-seq UMAP of single MYCN sample MB183. Gray boxes, proliferating cells. d) MYCN expression in C1 and C2 clones. b,c,) Per cell GSEA enrichment of proliferation (b) and differentiation (c) markers within UMAP of MB183 MYCN-amplified sample.*

5 - Are the oncogenes in question located on double minutes/ecDNA? The authors need to consider the possibility that double minutes are eliminated in cellular subsets because they are unequally distributed over daughter cells during cell division. This applies especially to the inferred phylogenies and the spatial analyses presented in Figure 4.

Reply: We agree that ecDNA could be a critical factor in the formation of the *MYC* heterogeneity. Moreover, the presence of such events in medulloblastoma tumors was shown in the corresponding recent study (Chapman et al, Nat. Genetic, 2023). Therefore, we performed *MYC/MYCN* FISH validations for most of our cases as noted in Extended Data Table 1; however, standard FISH does not allow precise resolution to distinguish ecDNA and more other technical methods that can visualize ecDNA by FISH (Hung et al, Nature, 2021) have only been demonstrated in cell lines or PDX samples, and not primary patient tissue.

Nevertheless, we observed a clear effect supporting the propagation of subclones as we hypothesize: the formation of other large CNV somatic changes in the oncogene subclone. For example, in the *MYCN* subclone in the case MB183 demonstrated not only *MYCN* amplification, but also gain in chromosome 4 (**Figure 2a**). Such oncogene-subclone-specific CNVs was observed in most of the cases. We provided the details about other CNV formations in *MYC/MYCN* subclones in the corresponding **Figures 2a,c,e**, but nevertheless noted ecDNA as a potential source of variance in the Discussion on pg 18:

While MYC or MYCN amplifications could arise from the formation of circular extrachromosomal DNA³⁵, our data show that all subclones with MYC/MYCN amplification demonstrated other large specific CNV gains and losses within the oncogene-amplified subclone, suggesting these clones are not the founder clone.

6 - The authors suggest that cellular migration may lead to interspersed cellular localization patterns of subclones, such as shown in Figure 4c. These results are based on RNA scope analysis of 100 target genes provided in Table 4. How were target genes selected, and how do they enable detection of genetic subclones? Or do patterns merely reflect the expression of MYCN? Have orthogonal assays for technical validation been considered? Please also state more clearly which instrument/technology has been used.

Reply: The list of the genes (n=100) for the spatial Resolve Bioscience protocol that we applied was based on specific properties. Initially, the gene selection was split into 2 blocks: tumor-specific genes (60%) and normal cell markers (40%). The tumor-specific genes had several blocks of selection: (i) known target markers of Group 3/4 tumors including *MYC*, *MYCN*, *SNCAIP* and *PRDM6*; (ii) proliferation, differentiation and cell cycle activity markers, such as *MKI67*; (iii) cells-of-origin-associated markers, such as *EOMES* for UBCs. The normal cell type markers were selected based on knowledge about normal cell types of cerebellum, which could be also present in the tumor tissue blocks such as astroglial, meningeal, immune etc. For each cell type at least 2 markers were selected (see **Extended Data Table 7**).

All the genes that we selected were also verified on single cell RNA-data as well as bulk profiles. We provided these details in the supplementary table listing the genes as well as extended the description in the Methods section.

For the projection of subclones, we used the computational method TransferAnchors from Seurat R package as noted in Methods (*Hao et al, Cell, 2021*). We also verified this projection by visually inspecting corresponding markers such as *MYC/MYCN/PRDM6* themselves to check the subclones' correctness. We have clear examples with strict borders such as MB249 for *PRDM6* and MB272 *MYC/MYCN* that confirmed this approach. The main orthogonal technical validation that we used was FISH applied on *MYC* and *MYCN* samples. We include this information into the **Extended Data Table 1**, listing FISH status for these cases.

Referee #1 (Remarks to the Author):

The authors have adequately addressed my concerns, including technical issues. I congratulate them on an interesting dataset and story.

Reply: We thank the Reviewer for evaluating our work and for their positive feedback.

Referee #2 (Remarks to the Author):

Okonechnikov and colleagues have made a thorough and careful attempt to answer the questions raised by the reviewers. I have no further issues that would improve the quality or support the conclusions of their manuscript significantly.

Reply: We thank the Reviewer for their positive feedback and confirmation of our work.

Referee #3 (Remarks to the Author):

Okonechnikov and colleagues provide a revised manuscript describing their study of the subclonal genomic architecture of medulloblastoma. Their response has clarified some of my previous questions regarding the experimental and computational methods, which has allowed me to better judge the quality of the analyses. While I believe the study is interesting, I still have major concerns. I am not convinced that (i) the main data types used in the study (10x Genomics single cell RNA-seq and ATAC-seq) are suitable for inferring CNV profiles of sufficient quality, that (ii) some of the computational approaches used to analyze these datasets are appropriate, and that (iii) the critical assessment and validation of results is sufficient.

Reply: We thank the Reviewer for their additional feedback; however, we believe that some of their persisting concerns may have stemmed from misunderstanding our methodology and results. We apologize for not being clearer in the previous rebuttal, but we offer additional clarifications below. We agree with the Reviewer that copy number inference from 10x single-cell data requires orthogonal validation, which we previously addressed by performing a systematic statistical comparison between the CNV profiles obtained from the scRNAseq/scATACseq data to those derived from bulk-DNA methylation array data. For the vast majority of the samples (**82%**), the CNV profile obtained from the scRNAseq data showed the highest similarity to the bulk-derived CNV profile of the same tumor sample. For scATACseq data, this was the case for all of the cases (**100%**). We show these results in **Extended Data Figure 2**.

We agree, however, that due to the noisy nature of scRNA/ATACseq, some CNVs may be wrongly inferred in some cases. To account for this limitation, we considered only CNV events that match CNVs seen in the bulk methylation data. We stated this in the manuscript *Methods* section (p.4, l.148).

Moreover, to critically assess and orthogonally validate our results, we generated additional scDNA/RNAseq data from a set of available cases. While (as expected, based on the limitations

discussed above) the phylogenies derived from scDNAseq data (**Extended Data Fig. 5**) might not completely match the results obtained from snRNA/ATACseq data analyses (Fig. 2), the expected MYCN/MYCN subclone observation was clearly orthogonally confirmed based on an additional independent method as shown in **Extended Data Figure 5b** for all selected cases (n=5). This result was based on the projection of gene expression profiles to ResolveOme RNA profiles using 10X scRNA data, as was described in the Methods. We suppose that the few discrepancies in inferred CNV profiles between DNA and RNA/ATAC are most likely due to technical limitations of the ResolveOme protocol – low coverage, small amount of cells (n=96). Importantly, we now compared the corresponding RNA/DNA subclone profiles to 10X snRNA profiles systematically based on CNV profiles as we further state in the reply to point 3-4 below. To avoid confusions, we also now clarify this issue in the Discussion section accordingly:

“Even though single-cell CNV profiling from single-cell RNA/ATAC data may have a certain false positive rate due to technical limitations, we used bulk CNV profiles to focus only on confident gains and losses, as well as verified subclone formation using single-cell DNA sequencing.”

1 - The meta-cell approach, which is central to many of the analyses of the study, remains poorly justified and tested. A single sentence in the methods is not sufficient to describe the approach, and a single citation does not make it an “established method”. Moreover, the cited publication does not study genetic subclones within the same tumor but uses the approach to separate all malignant from non-malignant cells. I understand that the code is available on GitHub, however my main issue is with the underlying assumptions and lack of critical assessment. My concern is that artifacts, which are inevitable when inferring CNV profiles from single cell RNA-seq and ATAC-seq datasets, are amplified using this approach, which in turn leads to wrongful definition of subclones.

Reply: We appreciate the Reviewer’s critique of a meta-cell-based approach. However, we respectfully disagree with the statement that the meta-cell approach “is central to many of the analyses of the study”. The meta-cell method was only applied to increase the computational speed/decrease resources usage and to improve the visualization of our results, as described in our previous reply. When comparing the results obtained with and without a meta-cell approach, we observe the same subclonal hierarchy results for MYC/MYCN/PRDM6 tumors, despite increased visual noise in the CNV. We further demonstrate in **Reviewer Figure 1** the analysis of four samples (MB183 MYCN, MB272 MYC/MYCN, MB89 MYC and MB249 PRDM6, also shown in **Figures 2a, 5a; Extended Data 3f, 4a** in the revised manuscript) where the same phylogeny of CNV profiles was identified with and without meta-cell approach for each case.

In summary, the raw CNV results are visually noisier compared to the results derived with a meta-cell-based approach, but the calls remain identical, thus showing the advantage of the meta-cell approach primarily for visualizing the single cell-derived CNV profiles.

Reviewer figure 1. Copy number profiles of snATAC-seq without (left side) and with (right side) meta-cells adjustment from MYCN sample MB183 (a,b), MYC-MYCN sample MB272 (c,d), MYC sample MB89 (e,f) and SCNAIP-PRDM6 sample MB249 (g,h). Signal color codes for both approaches are included below by sample MB2489.

2 - The sample that is showcased in Reviewer Figure 3.2 (sample MB272) is a good example of why I question if the data is suitable for inferring CNV profiles that are of sufficient quality for reliably defining subclones and phylogenies. The RNA-based analysis shows a loss of chromosome 10 (as annotated in the phylogenetic tree), which is not observed in the ATAC-based analysis (Figure 5a). The RNA-based analysis shows a gain of chromosome 19, whereas the ATAC-based analysis shows a loss of the same chromosome. There are many such examples in other samples. This puts the validity of the approach into question, especially when calling CNVs in smaller genomic regions or in subsets of cells. The novel analysis provided in Reviewer Figure 3.3 and in EDF 2 also indicates that inferred CNV profiles are

associated with a high level of technical noise. In comparison, the newly added single cell DNA-based dataset is of higher quality, however in many cases CNVs and phylogenies do not match well with previous results and new results are not properly integrated into the manuscript.

Reply: As discussed above, we agree that single-cell derived CNV profiles could suffer from noisy input data. In particular, the reviewer stressed that for the MB272 MYC/MYCN case “RNA-based analysis shows a loss of chromosome 10 (as annotated in the phylogenetic tree), which is not observed in the ATAC-based analysis”. The CNV profiles obtained from both bulk methylation data (**Reviewer Figure 2**, also available at interactive ShinyApp <https://kokonech.shinyapps.io/mbOncoAberrations/>) and snATACseq data (**Reviewer figure 1d**) of this case shows that only a part of 10q is lost, while the snRNA-seq CNV profile would imply a full loss of chr10. When again comparing to methylation bulk-data, the conclusion would be that the resolution of snATACseq CNV inference is superior to snRNAseq data based inference. Further, the reviewer states that “The RNA-based analysis shows a gain of chromosome 19, whereas the ATAC-based analysis shows a loss of the same chromosome.” Even though snRNA and snATAC-seq-derived CNV profiles show variance in chr19 CNV, this locus does not have any strong evidence in the control bulk CNV profile. Therefore, we did not include it as a specific subclone property in the phylogenetic reconstruction properties as shown in **Figure 2f**.

Reviewer figure 2. Bulk methylation CNV profile of case MB272

We would like to highlight that to systematically address such potential ambiguities, we performed a global computational comparison between sc- and bulk-derived CNVs in **Extended Data Figure 2**, showing overall very good agreement between the results obtained with scRNAseq or scATACseq and bulk-methylation data (numbers given above).

In our manuscript and response to the Reviewers we emphasize the use of single-cell derived CNV profiles to investigate clonal evolution in tumors, highlighting both advantages and

disadvantages of this approach. We strongly believe that the novel sub-clonal information gained from our single-cell approach provides fundamental insights into oncogenes that drive tumor evolution, and thus are an important advancement of our understanding of tumorigenesis in pediatric solid tumors.

3/4 - The approach to define genetic subclones and infer phylogenies as described in the revised manuscript is overly simplistic and relies heavily on manual inspection. This is an issue if CNV profiles are very noisy and subclone definitions are not obvious. How comparable are results obtained for the same sample from RNA and ATAC datasets? How do results compare to the newly generated DNA-based dataset? A systematic assessment is required.

Reply: Our approach of phylogenetic reconstruction is an established method based on hierarchical clustering of CNV profiles that has been used in several published sc-data studies by a number of outstanding and independent groups (PMIDs: 35681161, 37730813, 37993137). Moreover, we additionally used bulk methylation derived CNV profiles as controls to avoid false positive conclusions. As noted above, this global systematic comparison to bulk CNV profiles (**Extended Data Figure 2**) clearly demonstrated a high concordance between the CNV profiles derived from bulk methylation data and snRNA-seq data (82% concordance), with snATAC-seq data performing best (100% concordance). This result demonstrates the higher accuracy of snATAC-seq data for inferring single cell CNV profiles.

Reviewer figure 3. a) Cross-comparison of snRNA-seq CNV profiles against ResolveOme DNA profiles. b) Cross-comparison of snATAC-seq CNV profiles against ResolveOme DNA profiles profiles.

Finally, we would like to emphasise that we performed additional systematic assessment of the subclonal architectures derived from 10x scRNA/ATACseq-data and the newly added single-cell DNA sequencing data. For this purpose, we focused on the comparison of derived MYC/MYCN subclones between platforms: the subclone CNV profiles were combined into pseudobulk for 10X RNA/ATAC and ResolveOme DNA for each tumor and cross-compared to the other case with subclones verified via correlation of CNV profiles. These results are summarized in **Reviewer Figure 3**. In general, we observed a clear match for most of the

subclones with concordance measures of 81% for RNA and 90% for ATAC in subclone similarity when considering scDNA-seq as the reference. We also observed some variation between the CNV profiles derived from single-cell RNA/ATAC and single-cell DNA data, and, as noted above, it is potentially due to a lower cellular coverage of the latter (N=96 cells for scDNA-seq vs >8000 cells for 10x platform).

5 - For this example (MB183) I cannot make out the co-occurring gain of chromosome 4 in the RNA-based analysis or the ATAC-based analysis. The described phylogeny appears to be more in line with the new DNA-based analysis. In any case, the question if described oncogenes are located on ecDNA remains important and other technical methods should be explored.

Reply: Here we would like to contend that, visually, the chromosome 4 gain in the MYCN subclone of MB183 (C1) can be identified by the enrichment of the corresponding color in the snATACseq data. Although this gain is not readily visible in the snRNA-seq data, it is apparent in the snATAC-seq data, as shown on the left side of **Figure 2a/Reviewer Figure 1b** above.

We acknowledge, however, that this result may be less perceptible due to limitations inherent in color-based visualization. To address this, we present an alternative visualization method in **Reviewer figure 4a**, specifically focusing on the CNV profile of subclones on chromosome 4 via pseudobulk analysis, similar to the approach used for the MYCN fragment inspection on the right side of **Figure 2a**. This analysis confirms the gain of chromosome 4 in the MYCN subclone, supported by a statistically significant difference via t-test (p-val < 2.2e-16) as visualized in the boxplot in **Reviewer figure 4b**.

Reviewer figure 4. a) Inset of chromosome 4 region of copy numbers derived from snATAC-seq data in MYCN-amplified sample MB183 between subclones C1-MYCN and C2. Red, chromosome loss. Green, chromosome gain. b) Boxplot of CNV signal between C1 MYCN and C2 subclone. Dotted line: mean CNV signal across all chromosomes.

6 - My issue is with the finding of the interspersed cellular localization pattern of genetic subclones, as a contrast to the separated pattern. No reliable evidence is provided that the interspersed localization pattern (for example in MB165, Figure 4b-e) indeed represents different genetic subclones. Given the current evidence, it cannot be ruled out that cells of the same genetic subclone express varying levels of MYCN or “differentiation markers” that result in different genetic subclone annotations using the TransferAnchors approach.

Reply: We fully agree with this point of the Reviewer. Indeed, we rely on a specific spatial protocol and derive this result using a computational method. While we demonstrated the applied projection method to be correct for control cases with spatial subclonal variance, such as PRDM6 or MYC/MYCN cases, we have not performed orthogonal experiments, such as FISH + IHC in this case. To address this, we have now extended the description of the limitations of the spatial technology we used in the Discussion (l. 596, p.18), (novel text highlighted in bold font):

*“Single-cell spatial data from Group 3/4 tumors allowed us to inspect the composition of the subclones across tumor tissue fragments; however, due to the fixed number of genes (n=100) and limited image size (max 2 mm), some other spatial properties, such as formation of blood vessels, were not covered in our study. **Additionally, the computational methods applied for projection of subclonal CNV patterns into spatial data may be biased based on the technical properties of the protocol.** Therefore, the application of novel spatial techniques that overcome such limitations will be an important research direction in the future.”*

Referee #1 (Remarks to the Author):

Okonechnikov and colleagues have revised their manuscript in response to comments from Reviewer 3 regarding the accuracy of CNV reconstruction from snRNA and snATAC. After re-review of the manuscript alongside these critiques and the authors response, it seems that the key conclusion of the study (namely single cell data supporting focal amplifications of MYC and oncogenes as late or subclonal events in Group 3/4 MBs) still stand, despite some valid technical caveats raised by the reviewer.

Reply: We are thankful to the reviewer for confirmation that our main conclusions are valid.

Additional comments

- Figure 5b panel legend appears to missing colors (i.e. for C1, C2, C3)

Reply: We have corrected the corresponding figure to include the colors.

- Scrutinizing the single cell data some more, the criteria for delineating the "proliferation=yes box" drawn in Fig 5b and other figure panels (e.g. Figure 2c, 2g) are not well described. It seems that the signature was computed per cell but the boxes were somewhat arbitrarily hand drawn (e.g. comparing the MB183 UMAP for the actual proliferation signature data plotted in Extended Data Fig 3b with the "box" version in Figure 2c). And using MB183 as an example, it seems that not all the cells included in the proliferation box in Fig 2c are lighting up for the signature in EDF 3b. Furthermore, there is a cluster of cells that are not boxed on the upper right side of EDF3b that are not boxed. While this does not seem central to the story, the approach for labeling the UMAPs with this box should be better described in the corresponding captions and/or text.

Reply: The boxes representing proliferating cells in the sample UMAPs were assigned manually based on two aspects:

- 1) Proliferation enrichment (also demonstrated in an additional figure for each sample as noted by reviewer)
- 2) The presence of a unique cluster of cells identified using Seurat and enriched with this signal.

This second aspect had an impact on drawing the boxes around the corresponding cell groups. Indeed, some other cells demonstrated proliferation enrichment, but did not constitute a second cluster. To clarify this to the readers, we now updated each legend accordingly (Figure 2c,g,j; Figure 5b). Below is example for the Figure 2c:

"Figure 2. ... c) snRNA-seq UMAP of single MYCN sample MB183. Gray boxes, proliferating cells cluster with strong proliferation enrichment. Blue, C1 clone. Orange, C2 clone."

We have updated the Methods to make this more clear as well:

"The enrichment of proliferation, differentiation, and progenitor-like activity of medulloblastoma-specific markers per cell was performed using single sample function from

the GSVA R package using two independent reference datasets. Cell clusters enriched with proliferation signals were selected and marked based on maximum GSVA signal enrichment from manual inspection per sample.”

Importantly, we provide detailed clustering information for each sample at our github repository as described in the wiki:

<https://github.com/kokonech/mbOncoAberrations/wiki#test-input-data>

- Extended Data Figure 6c is missing a legend

Reply: We are thankful for noting this issue – we have updated the legend accordingly.

Referee #3 (Remarks to the Author):

Okonechnikov and colleagues have provided a rebuttal to my second round of comments that includes several additional figures and have made further clarifications within the manuscript. While some concerns regarding the inference of subclones from single-cell RNA and ATAC-seq profiles remain (see below), I believe that these can be addressed and that main findings of the study are sufficiently validated using orthogonal approaches.

Reply: We are thankful to the reviewer for the thoughtful comments and for confirmation that our main findings are sufficiently validated using orthogonal approaches.

My issue with the meta-cell approach for CNV analysis is that it assumes that cells within the same Seurat cluster (which is based on single-cell RNA or ATAC-seq profiles) share a similar CNV profile and hence can be combined into meta-cells. While this might be the case, it is also possible that cells from different subclones fall into the same cluster and are erroneously combined, thereby blurring out their signal. Looking at the results, I do not think that this is an issue that led to wrongful conclusions in the end. However, this is an unconventional approach that should be clearly laid out and justified to the reader (e.g., are major subclones likely to form separate clusters, how many RNA/ATAC clusters are detected per sample). Unless I have misunderstood the approach, in which case it needs to be explained more clearly, this is an important part of the analysis that is not “only applied to increase the computational speed and to improve the visualization”, as stated in the rebuttal. The definition of subclones and trees are dependent on it.

Reply: We agree with the reviewer that meta-cell method could be potentially biased due to the pre-selection of cells based on their clustering. To better explain how we mitigate this bias, we now explicitly state in the manuscript that we inspected the stability of the approach by comparing standard single-cell CNV calling and initial clustering on a subset of samples. We measured the similarity of the clustering approach to the subclones based on purity estimation and further explained how we used marker gene expression and signal enrichment for the clustering cut limit adjustment. The following text is now included in the Methods:

”Initially, CNV analysis was performed on a subset of samples (n=2 for each oncogene) using the InferCNV tool (PMID: 24925914) on the raw gene expression matrix with droplet protocol

adjusted parameters (average read counts cutoff 0.25, smooth method runmeans, denoise active) and hierarchical clustering via ward.D2 method to derive the clonal phylogeny. The main subclones obtained from this phylogeny demonstrated a close match to the initial Seurat clustering results (mean purity evaluation metrics across samples: 0.921). To further improve the visualization and increase computational efficiency, the CNV analysis was performed on the full cohort by transferring single cells into meta-cells, based on an established method (PMID: 37883975). For this purpose we computed the sum of gene expression counts across n=5 cells combined within the clusters derived from Seurat processing. The meta-cell InferCNV calling was performed for each sample separately with read counts cutoff 0.5 and the phylogeny clustering results were visualized in UMAPs with k as number of clusters varying from 2 to 5. Differentially expressed genes for identified subclones per sample were computed via a Wilcoxon Rank Sum test. Significant MYC/MYCN/PRDM6 differential expression and progenitor-like activity enrichment values were computed per cell to finalize the derived phylogeny cut limit for each case after manual inspection. The selected cut limit for the number of subclones in the phylogeny was verified by using random tree subcluster partition in the phylogeny reconstruction with a minimum p-value < 0.05.”

In addition, we also share and describe the comparison between InferCNV and Seurat clustering for the verification of meta-cell approach usage as well as include the precise RNA/ATAC clustering information per sample, as an additional point in the corresponding data analysis source code repository documentation of our manuscript:
<https://github.com/kokonech/mbOncoAberrations/wiki#meta-cells-usage-verification>

Moreover, we created a separate bioinformatics tool with detailed documentation to demonstrate the method we used for CNV calling of the ATAC-seq data. This documentation includes the meta-cell approach and a “how-to” on performing the analysis on our shared data: <https://github.com/kokonech/atacInferCnv>

The added single-cell DNA sequencing analysis is an important validation of initial findings that highlights the validity and limitations of the RNA/ATAC-based approach. I agree that for most analyzed samples the inferred subclones match between both analyses. For MB272, the case that is characterized by an amplification of both MYC and MYCN, it appears that the amplification of MYCN on chromosome 2p is not detected in the DNA-based analysis or in the microarray-based CNV analysis of bulk DNA. Was a different piece of tumor tissue used for these analyses? The authors should clarify.

Reply: The comparison of single-cell DNA vs single-cell RNA/ATAC CNV profiles is included now in the manuscript as **Extended Data Figure 5b-c**.

As noted by reviewer, the single-cell DNA-sequencing data of MB272 does not contain a MYCN-amplified population. Indeed, different parts of the tissue were used for bulk methylation, single-cell RNA/ATAC, spatial transcriptomics, and single-cell DNA-sequencing, providing a plausible explanation for the lack of signal. We have stated this limitation in the manuscript accordingly:

“This subgroup II sample was originally characterized as MYC-amplified only based on bulk methylation profiling (Extended Data Figure 9a), while the MYCN-amplified subclones were

observed in the single-cell multiome and spatial transcriptomics analysis. This discrepancy most likely results from examining different fragments of the tumor tissue for each analysis.”